# Genetic risk predicts adolescent mood pathology via sexual differentiation of brain function and physiological aging

Raluca Petrican [1] ✉, Alex Fornito [2], Christopher Murgatroyd [3], Emma Boyland[1] & Charlotte A. Hardman[1]

Recent evidence challenged the traditional, categorical approach to sex differences, indicating that each human brain comprises a mosaic of features, some of which are more common among males, others, among females, whereas the remaining are equally common between sexes. Thus, a focus on regional sexual differentiation of brain function, instead of holistic sex-based categorization, could be more useful for understanding psychiatric conditions, such as mood and behavioural disorders, to which males and females are differentially vulnerable. To probe this untested hypothesis, we estimate sexual differentiation within each brain in a longitudinal ($N = 199$) and cross-sectional ($N = 277$) sample of male and female adolescents. Greater feminization of association networks, involved in higher-order cognition, compared to sensory networks, at ages 9-10 correlates with earlier puberty and greater immune/metabolic dysregulation at ages 11-12, particularly among girls. Greater masculinization of association networks relates to later puberty and reduced immune/metabolic dysregulation, especially among boys. The brain and physiological profiles sequentially mediate the relationship between genetic risk and rising mood/behavioural symptoms. These links are replicated in the cross-sectional sample and shown to hold across sexes. Our study emphasizes the importance of integrating assessments of regional sexual differentiation and physiology in personalizing psychiatric intervention in adolescence.

The high degree of neuroplasticity associated with adolescence renders it a key life stage for the onset of psychiatric disorders and the emergence of sex differences in mental health trajectories[1,2]. Interactions between social environmental factors and interconnected neurophysiological processes spanning resolutions that range from the molecular (e.g., gene expression[3,4]) to the systemic (e.g., the maturation of large-scale brain networks[5]; inflammation[6,7]) are thought to underpin the rising differential susceptibility of males and females to mood disorders during this life stage[8,9]. The emergence of sex-

dependent risk for psychopathology broadly coincides with the onset of puberty[9]. Accordingly, pubertal hormones have been implicated not only in the development of typical sex differences in behavior and neurophysiology[10–15], but also in the emergence of those relevant to internalizing disorders[12,16–18]. Specifically, with advancing sexual maturation, female adolescents show increasingly stronger neuroimmune responses to stress relative to male youths. These findings dovetail with evidence that inflammation and metabolic dysregulation, likely indicative of greater physiological wear-and-tear[19,20], make a

[1]Institute of Population Health, Department of Psychology, University of Liverpool, Bedford Street South, Liverpool L69 7ZA, UK. [2]The Turner Institute for Brain and Mental Health, School of Psychological Sciences and Monash Biomedical Imaging, Monash University, Melbourne, VIC, Australia. [3]Department of Life Sciences, Manchester Metropolitan University, Manchester, UK. ✉e-mail: raluca.petrican@liverpool.ac.uk

particularly strong contribution to the pathology of depression and anxiety disorders among female adolescents and young adults[21,22].

Extant research on sex differences in risk for mood disorders and their link to immune and metabolic dysregulation has traditionally focused on differences between biological males and females. However, recent evidence suggests that categorizing individuals based on their biological sex ignores considerable within-group variability, since the human brain may be better described along a continuum of sexual differentiation (for an example, see ref. 23). Thus, continuous assessments of the degree to which an individual's brain is sexually differentiated (i.e., the degree to which it shows brain features expressed more strongly among healthy male, rather than female, adults, or vice versa) may offer a more sensitive approach to personalizing psychiatric detection and intervention paradigms.

To examine this possibility, we leveraged two multimodal public datasets, the Adolescent Brain and Cognitive Development (ABCD) and the Human Connectome Project-Development (HCP-D) studies. Our investigation was guided by evidence that adolescent neurodevelopment and mental health are influenced by the interaction between the immune system and the hypothalamic-pituitary-gonadal (HPG) axis[21,24–30]. We therefore examined how sexual differentiation of brain function may act in conjunction with reproductive maturation and physiological aging (i.e., immune/metabolic dysregulation) processes to impact vulnerability to disorders showing sex-biased prevalence[10,11,31] (i.e., internalizing vs externalizing psychopathology). To this end, we first took advantage of cross-sectional and longitudinal data in the ABCD study to test whether neural sexual differentiation, reproductive maturation, and physiological wear-and-tear are

primarily markers of concurrent or future change in psychiatric symptoms, and probe the extent to which any such relationships vary by sex. Subsequently, we examined whether the link between genetic risk and psychopathology could be sequentially explained via neural sexual differentiation and reproductive maturation/physiological wear-and-tear (i.e., immune/metabolic dysregulation). Put differently, we tested whether genetic risk for psychiatric disorders with sex-biased prevalence is linked to the emergence of brain features differentially expressed among male vs female individuals, which, in turn, predict sex-specific physiological maturation processes. This line of inquiry complemented prior investigations on the role of reproductive maturation in driving neural sexual differentiation, immune responses to stress and sex-dependent risk for mood disorders[9,12,16–18]. Finally, we leveraged the larger cross-sectional HCP-D sample available for our present analyses to examine whether any of the effects detected in the ABCD sample are apparent in an independent sample. Figure 1 depicts the conceptual framework of the present study.

Our analytical strategy, shown in Fig. 2, was as follows. First, in the ABCD sample, we investigated whether pubertal development (i.e., adrenarche and gonadarche, measured separately via self-reports and hormonal assays) and immune/metabolic dysregulation (estimated with the PhenoAge algorithm[20]) at ages 11-12 would show sex-dependent relationships with contemporaneous variations in psychiatric disorder symptoms (measured through parental ratings on the Child Behavior Checklist [CBCL][32]) and/or changes in psychiatric symptoms from ages 11-12 to 12-13 (Fig. 2: Analysis 1). Second, we used functional magnetic resonance imaging (fMRI) to characterize regionally specific patterns of sexual differentiation in brain function

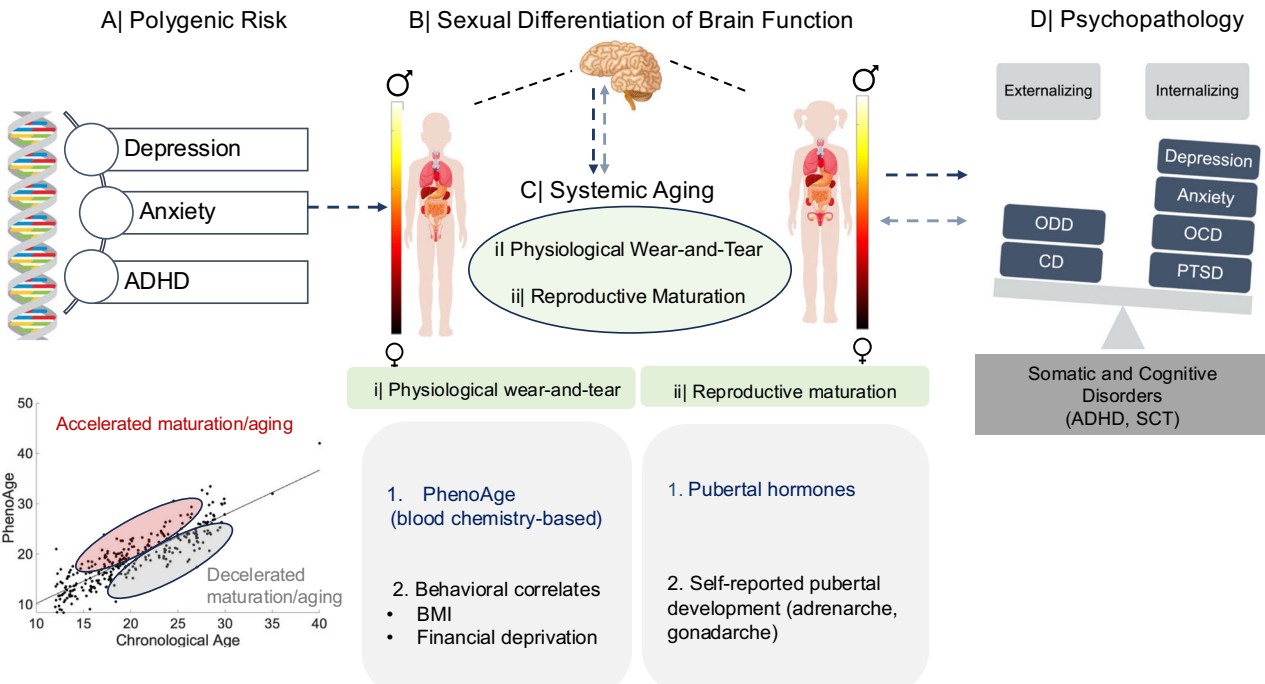

Fig. 1 | Schematic representation of our conceptual framework. In the ABCD sample, we set out to test the causal sequence underpinning the relationship between polygenic risk (panel A) and differential vulnerability to internalizing vs externalizing spectrum disorders (panel D) via neurobiological maturation processes: sexual differentiation of brain function (panel B), and systemic aging (panel C), specifically, physiological wear-and-tear (i.e., immune and metabolic dysregulation, assessed via the PhenoAge algorithm, panel C-i) and reproductive maturation (specifically, pubertal hormone levels and self-reported adrenarche vs gonadarche, panel C-ii). The PhenoAge algorithm (panel C−i) uses an exponential function to predict mortality from a set of biomarkers in a reference group. An individual's PhenoAge biological age prediction corresponds to the chronological age at which their mortality risk would be normal in the reference group. A PhenoAge estimate higher/lower than an individual's chronological age indicates advanced/delayed physiological aging, respectively. The longitudinal analyses from the ABCD sample (as indicated through the dark blue arrows) were supplemented by an examination of the cross-sectional relationships among the same variables (apart from genetic risk, PhenoAge, and pubertal hormones) in the HCP-D sample (as indicated through the light blue arrows). ADHD attention deficit hyperactivity disorder, BMI body mass index, CD conduct disorder, OCD obsessive-compulsive disorder, ODD Oppositional Defiant Disorder, PTSD post-traumatic stress disorder, SCT slow cognitive tempo.

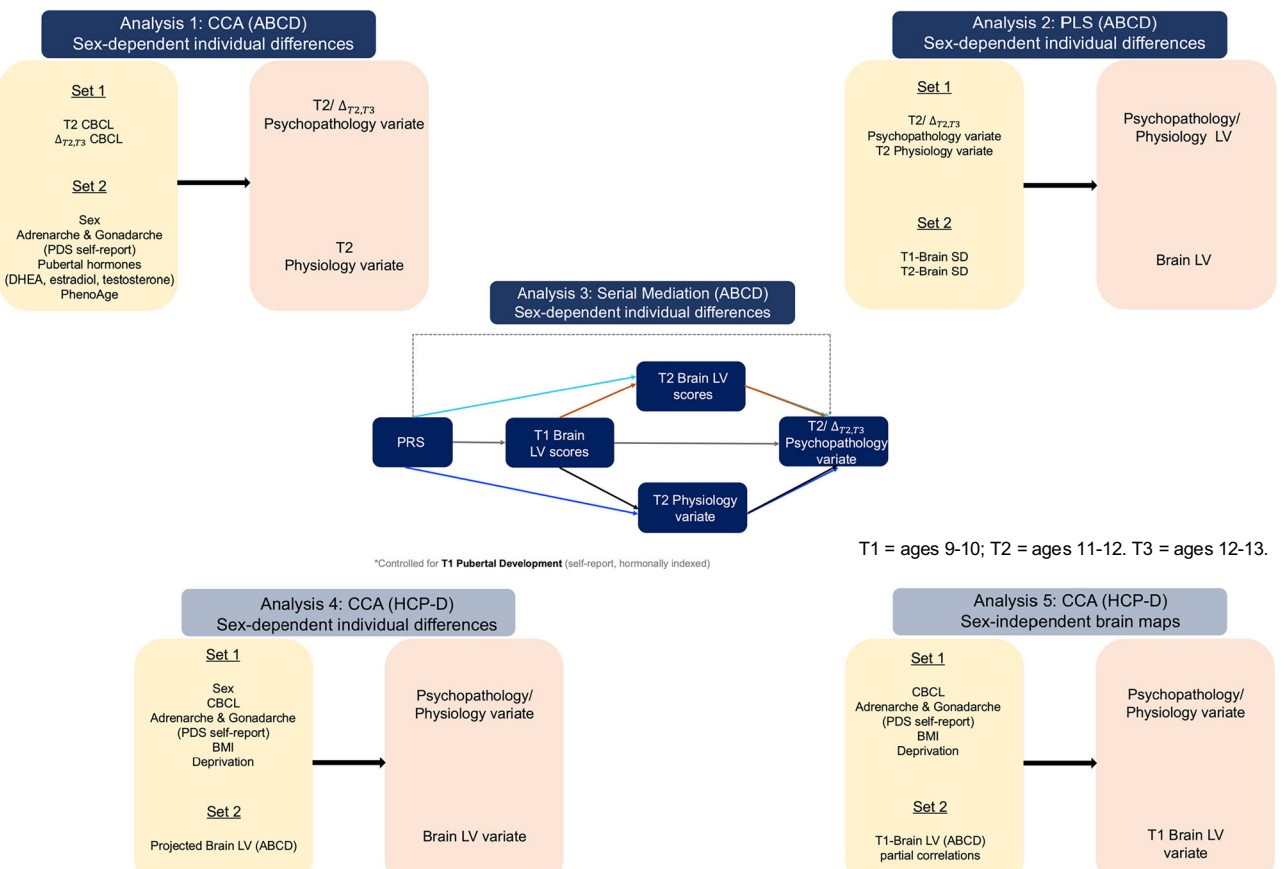

**Fig. 2 | Outline of our workflow.** Analysis 1: In the ABCD sample, CCA related psychiatric disorder symptoms assessed with CBCL scores at Time 2 and change in CBCL scores from Time 2 to Time 3 to sex-dependent patterns of pubertal development (i.e., self-reported and hormonally derived adrenarche/gonadarche) and physiological aging (i.e., PhenoAge) assessed at Time 2. Analysis 2: A PLS correlation analysis related the psychopathology and physiology CCA variates from Analysis 1 to the ROI-specific sexual differentiation indices estimated in reference to the intrinsic functional coupling profiles observed in the HCP-Young Adult sample. Analysis 3: A serial mediation analysis probed the temporal sequence in which the sexual differentiation of brain function LV (from Analysis 2) and the physiology CCA variate (from Analysis 1) mediated the impact of genetic risk markers (PRS) on the psychopathology CCA variate (from Analysis 1). Black arrows indicate our indirect path of interest, whose specificity was tested by including alternate mediation paths depicted through colored arrows. All the contributing variables were residualized by self-reported and hormonally related adrenarche/gonadarche at Time 1. Analysis 4: In a subset of participants from the HCP-D covering late childhood to late adolescence, a CCA probed the cross-sectional replicability of the longitudinal links among neural sexual differentiation (i.e., the brain LV from Analysis 2 which was projected onto the HCP-D sample), psychopathology and physiology, as established in the ABCD sample. Analysis 5: A CCA tested the overlap between the neural sexual differentiation brain map identified in the ABCD sample and neural sexual differentiation maps, characterized in the HCP-D sample and tracking, in a sex-independent manner, psychopathology, correlates of PhenoAge and self-reported pubertal development. HCP Human Connectome Project, CCA canonical correlation analysis, PLS partial least squares correlation analysis, BMI body mass index, CBCL Child Behavior Checklist, DHEA Dehydroepiandrosterone, LV latent variable, PDS Pubertal Development Scale, PRS polygenic risk score, SD sexual differentiation, T1 Time 1, T2 Time 2, T3 Time 3.

at ages 9-10 and 11-12 for each individual adolescent in the ABCD sample. Subsequently, in the ABCD sample, we tested the relevance of neural sexual differentiation for concurrent and future pubertal development/physiological aging and associated patterns of psychiatric susceptibility (as identified at point 1) (Fig. 2: Analysis 2). To contextualize our findings within the broader literature, we also probed the extent to which the regionally specific patterns of neural sexual differentiation thus identified tracked the sensorimotor-association (S-A) axis defined by prior work as a key organizing principle of neurodevelopment (i.e., association areas mature later than sensorimotor areas)[33,34] and of corresponding sex differences in functional brain network organization (i.e., association network regions are most effective at classifying youths based on sex)[35]. Third, in the ABCD sample, we probed whether neural sexual differentiation in late childhood (ages 9-10) and reproductive maturation/physiological wear-and-tear in early adolescence (ages 11-12) sequentially mediate the impact of genetic risk markers on liability to psychiatric disorders (Fig. 2: Analysis 3). Fourth, in a subset of participants from the HCP-D covering late childhood to late adolescence (ages 9–17), we sought to replicate the links between psychiatric risk, sexual differentiation of brain function, and physiology (i.e., self-reported pubertal development [adrenarche, gonadarche] and behavioral correlates of PhenoAge), as observed in the ABCD sample (see point 2) (Fig. 2: Analysis 4). Finally, in the same HCP-D participants, we examined whether the patterns of neural sexual differentiation identified in the ABCD sample can predict, in a sex-independent manner, variations in disorders that show sex-biased prevalence (i.e., internalizing vs externalizing disorders) and related physiology-relevant patterns (as detailed at point 5) (Fig. 2: Analysis 5). This analysis probes whether sex-biased prevalence of psychiatric disorders in adolescence is associated with relative neural sexual differentiation alone, or whether this association depends on its interactions with other sex-specific biological processes.

## Results

All the reported permutation- and bootstrap-based tests are based on 100,000 samples, with the exception of the mediational analysis (Analysis 3), which featured 50,000 bootstrap samples.

## Analysis 1: Advanced phenoage and pubertal maturation show sex differential associations with anxiety, depression and behavioral disorders (ABCD)

We first sought to establish whether, in early adolescence (ages 11-12), pubertal maturation and physiological aging would show sex-dependent associations with concurrent and/or future risk for specific psychiatric disorders (assessed through parental ratings on the CBCL). Self-ratings on the Pubertal Development Scale [PDS][36] and hormonal assays were used to estimate progression through the two component processes of puberty which are linked to the emergence of secondary sex characteristics (i.e., adrenarche), as well as reproductive maturation and development of sex-specific characteristics (i.e., gonadarche)[37,38]. Hormonal markers of adrenarche and gonadarche were computed separately for boys and girls. In girls, average standardized DHEA and testosterone levels indexed adrenarche[37], whereas standardized estradiol levels gauged gonadarche (cf.[37,38]). In boys, standardized DHEA levels assessed adrenarche, while standardized testosterone levels were used as an indicator for gonadarche (for a review of supporting findings, see ref. [38]).

Physiological wear-and-tear was estimated with the PhenoAge algorithm[39] which uses an exponential growth formula to predict mortality from blood chemistry measures of metabolic and immune system functioning in a reference group. The PhenoAge algorithm yields a biological age prediction which corresponds to the chronological age at which an individual's mortality risk would be normal in a reference group. A more advanced PhenoAge (relative to chronological age) indicates greater physiological wear-and-tear and, thus, greater than expected (by chronological age) risk for disease and mortality[39,40]. Using the blood chemistry data available in the ABCD sample, we trained and validated the PhenoAge algorithm in two young adult (ages 20–40) samples from the National Health and Nutrition Examination Survey (NHANES) (https://wwwn.cdc.gov/nchs/nhanes/Default.aspx) (NHANES III: $N = 6084$; 2892 males; NHANES IV: $N = 14{,}782$; 7421 males). We focused on young adults for whom, where applicable, deaths were related to intrinsic causes (e.g., cardiac, cancers, kidney/lung-related, respiratory infections, diabetes) rather than accidents, because we reasoned that they would best reflect physiological wear and tear processes that are distinguishable from those linked to typical aging and, thus, most likely to be observed among the adolescents in the ABCD cohort. In both NHANES samples, more advanced PhenoAge than expected based on chronological age, was observed among poorer individuals (Spearman's *rho* (5611) = 0.16, 95% CI = [0.14; 19], $p = 7.71 \times 10^{-34}$ [NHANES III] and *rho*(14780) = 0.16, 95% CI = [0.14; 0.18], $p = 2.5 \times 10^{-85}$ [NHANES IV]), was linked to higher body mass index (BMI) (Spearman's *rho* (6075) = 0.22, 95% CI = [0.20; 24], $p = 4.81 \times 10^{-67}$ [NHANES III] and *rho*(14780) = 0.28, 95% CI = [0.27; 30], $p = 1.41 \times 10^{-265}$ [NHANES IV]) and poorer overall health (Spearman's *rho* (6082) = 0.15, 95% CI = [0.13; 18], $p = 2 \times 10^{-32}$ [NHANES III] and *rho*(12406) = 0.20, 95% CI = [0.18; 22], $p = 1.8 \times 10^{-113}$ [NHANES IV]).

We replicated these associations in the larger ABCD sample of biologically unrelated youths with available blood chemistry, BMI, financial deprivation, and medical history data at the two- or three-year follow-up ($N = 922$). Specifically, in these youths, a cross-validated canonical correlation analysis (CCA) linked more advanced PhenoAge (adjusted for chronological age and sex) to parental reports of financial deprivation, higher BMI and more serious medical problems, as indicated by the number of emergency room visits and unplanned medical visits, particularly those related to very severe headaches, episodes of high fever, asthma, bronchitis, allergies and diabetes. Importantly, the association between PhenoAge and the physical health/deprivation variate was replicated in our target sample of 199 ABCD participants (see Methods for detailed analyses and Fig. S1 for a representation of the pertinent CCA results).

After establishing that PhenoAge is a robust indicator of physiological wear-and-tear in early adolescence, we proceeded to investigate its relevance, together with pubertal development, to concurrent and subsequent psychopathology. To this end, we conducted a CCA in our target ABCD sample ($N = 199$). This analysis identified maximally correlated latent factors (i.e., variates) linking the 18 CBCL scores (i.e., 9 scores from the two-year follow-up [ages 11–12] and 9 change scores estimated as the difference between the corresponding standardized scores at the two- and three-year follow-ups: ages 11–12 to 12–13; hereafter called the T2/$\Delta_{T2,T3}$ psychopathology CCA variate [variate 1]) to sex assigned at birth (corroborated through single nucleotide polymorphism (SNP) analysis and/or menstrual history), PhenoAge, pubertal hormone levels, and self-reported adrenarche/gonadarche assessed at T2 (hereafter called the T2 physiology CCA variate [variate 2]). In this analysis, the sex-dependence of the psychopathology-physiology relationship is indicated by a robust loading of the sex variable (coded "0" for male, "1" for female) on variate 2. A preferential association between the physiology variate and concurrent or future change in psychiatric symptoms is implied by a robust loading of the T2 or $\Delta_{T2,T3}$ CBCL change scores on variate 1. A robust loading of a T2 CBCL score, but not of the corresponding $\Delta_{T2,T3}$ CBCL change score on variate 1 would indicate a link between the physiology variate and consistently lower or higher (depending on the canonical loading sign) disorder-specific scores from the ages of 11–12 to the ages of 12–13.

Discovery CCAs unveiled a statistically significant physiology-psychopathology variate pair which was successfully cross-validated in analyses adjusted for age, race, testing site, handedness, adoption status (coded "1" for the 3 adoptees and "0" for the remaining sample), ambiguous biological sex (coded "1" for the 2 participants with ambiguous sex and "0" for the remaining sample) ($r_{CV} = 0.18$, permutation-based $p = 0.019$, 95% CI = [0.02; 0.32], Fig. 3C). Thus, female adolescents characterized by more advanced PhenoAge and pubertal development, particularly, gonadarche, showed persistently higher somatic disorder scores and increasing depression, anxiety, OCD and PTSD scores from ages 11-12 to 12-13 (Fig. 3A, B). Conversely, consistently lower somatic disorder scores but rising conduct and oppositional defiant disorder scores were observed among male adolescents characterized by a younger PhenoAge and delayed pubertal development (Fig. 3A, B).

To shed further light on the results described above, we evaluated whether sex, PhenoAge and the pubertal development variables, particularly gonadarche, make independent contributions to the physiology variate. To this end, we conducted a regression analysis predicting the z-scored physiology variate from sex, z-scored PhenoAge, z-scored pubertal development variables, and all the control variables entered in the cross-validation of the Analysis 1 results. This analysis is akin to estimating the standardized canonical coefficients, which reflect the unique contribution of each observed variable to its corresponding variate[41]. The results confirmed that the physiology variate reflects the unique, additive contributions of sex ($b = 0.88$, $t(171) = 17.44$, $p = 8.64 \times 10^{-40}$, 95% CI [0.79; 0.99]), PhenoAge ($b = 0.55$, $t(171) = 19.56$, $p = 1.74 \times 10^{-45}$, 95% CI [0.50; 0.61], and gonadarche, based on hormone levels ($b = 0.14$, $t(171) = 4.39$, $p = 2.02 \times 10^{-5}$, 95% CI [0.08; 0.21]) and self-reports ($b = 0.33$, $t(171) = 10.57$, $p = 2.10 \times 10^{-20}$, 95% CI [0.27; 0.39]). Hormonally indexed and self-reported adrenarche did not show robust unique associations with the physiology variate in the multiple linear regression analysis. The results remain unchanged if the z-scored physiology variate is regressed only on its corresponding observed variables. These findings indicate that the physiology variate does not merely reflect earlier reproductive maturation and more advanced PhenoAge in girls relative to boys.

## Analysis 2: Sexual differentiation of regional functional coupling (fc) predicts advanced phenoage and pubertal maturation (ABCD)

Having established the sex differential and primarily longitudinal relationship between physiology and psychopathology in the ABCD

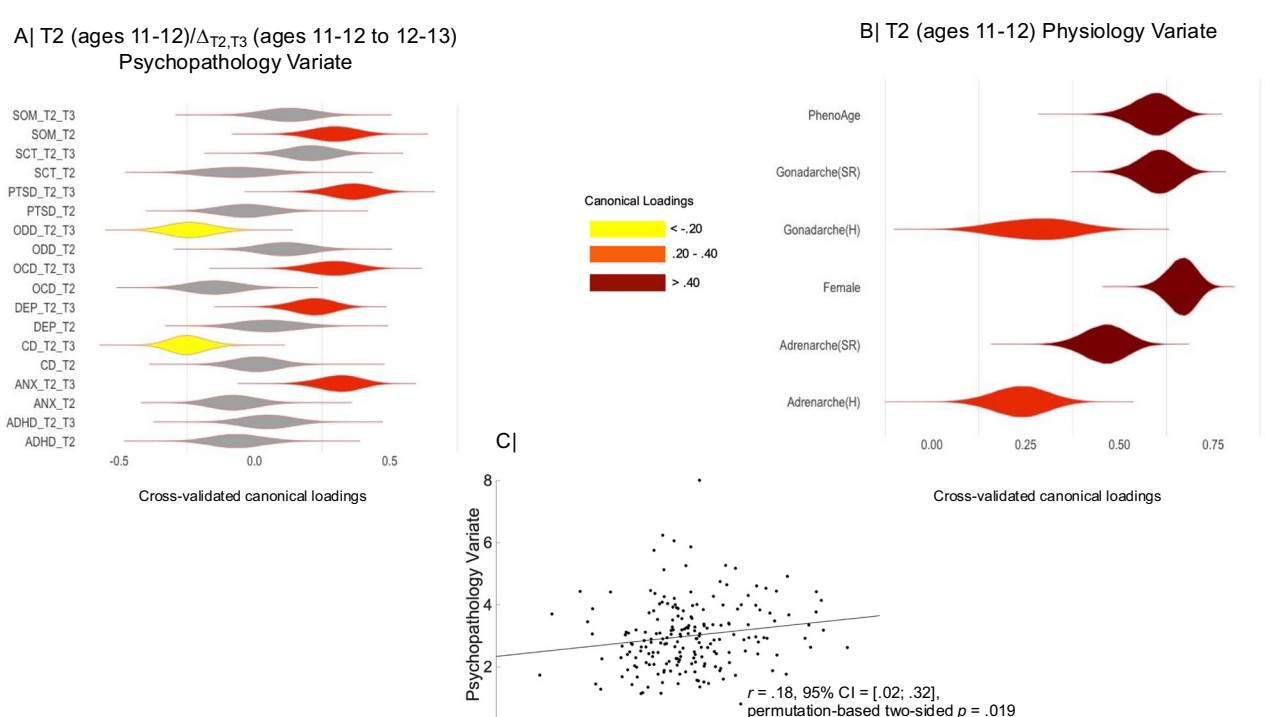

Analysis 1: Sex, Physiological Age, Pubertal Development, Pubertal Hormones, and Internalizing vs Externalizing Psychopathology

**Fig. 3 | Disorder-specific two-year follow-up ("_T2) and two- to three-year fol-low-up change ("_T2_T3") scores linked by CCA to sex, PhenoAge, as well as self-reported and hormonally based indices of adrenarche and gonadarche in the ABCD sample.** Panel **A** The horizontal graph shows the partial correlation coeffi-cients describing the relationship between the observed disorder-specific scores and the predicted value of their corresponding canonical variate across all test CCA folds from the cross-validation procedure. Panel **B** The horizontal graph shows the partial correlation coefficients describing the relationship between the observed values of the sex (female = 1, male = 0), PhenoAge, as well as self-reported and hormonally based indices of adrenarche and gonadarche with the predicted value of their corresponding canonical variate across all test CCA folds from the cross-

validation procedure. The shaded areas with non-zero color coding (see "Loadings" legend) in panels **A** and **B** correspond to robust correlations based on 99% CIs from the cross-validation procedure. The violin plots in panels **A** and **B** depict the dis-tribution of partial correlation coefficients across the 100,000 bootstrap samples from the cross-validation procedure. Panel **C** contains the scatter plot describing the linear relationship between the two CCA variates pictured in panels **A** and **B**. CCA = canonical correlation analysis. ADHD = attention deficit hyperactivity dis-order. Anx anxiety, CD conduct disorder, Dep depression, OCD obsessive-compulsive disorder. ODD oppositional defiant disorder, PTSD post-traumatic stress disorder, SCT slow cognitive tempo, SOM somatic disorder, SR self-report, H hormonally based.

sample, we next sought to shed light on its relevance to patterns of neural sexual differentiation. We thus entered the T2/$\Delta_{T2,T3}$ psycho-pathology and T2 physiology CCA variates from analysis 1 into a partial least squares (PLS) correlation (Analysis 2) to examine their associa-tions with ROI-specific measures of sexual differentiation in FC, as identified at T1 (ages 9-10) and T2 (ages 11-12). Sexual differentiation in the ABCD sample was characterized in reference to the functional connectivity (FC) patterns observed among the young healthy adult participants (ages 22–30) in the Human Connectome Project (HCP). These individuals have reached reproductive maturity and are thus expected to show maximal differentiation of brain function related to their biological sex (as assigned at birth [self-reported] and confirmed through menstruation history[12,16–18,42,43]). ROI-to-ROI FC estimates based on two widely used functional atlases[44,45] were extracted inde-pendently from the ABCD and HCP samples, respectively.

The discovery PLS analysis uncovered a single brain-behavior latent variable (LV) pair ($p = 2 \times 10^{-5}$, shared variance of 65.35%) relating T1 and T2 regional sexual differentiation to T2 physiology and T2/$\Delta_{T2,T3}$ psychopathology scores (Fig. 4A; see Fig. S2 for the replication of these results with the Gordon atlas, and Fig. S3, S4, for the replica-tion of these results using more stringent motion controls). Cross-validation tests, which controlled for participant motion (i.e., average frame-wise displacement[46]), age, race, testing site, handedness,

adoption status, ambiguous biological sex, suggested that the brain-behavior LV relationship is robust (permutation-based $p = 0.001$, shared variance of 65.30%) and mostly captures the link between neural sexual differentiation at ages 9-10 and physiology at ages 11-12 ($r_{CV} = 0.19$, 95% CI = [0.05; 0.34]; see Fig. 4B). Of note, the association between concurrent neural sexual differentiation and physiology at ages 11-12 failed to reach conventional reliability levels ($r_{CV} = 0.05$, 95% CI = [−0.08; 0.20]; see Fig. 4B). Testifying to its specificity, the corre-lation between T1 neural sexual differentiation and T2 Physiology was left virtually unchanged after additionally controlling for the available T1 Physiology measures (i.e., pubertal hormones, self-reported adre-narche and gonadarche) ($r_{CV} = 0.18$, 95% CI = [0.04; 0.34]).

Cross-validated ROI loadings on the brain LV tracked the cano-nical S-A axis (Spearman's $rho = −0.47$, $p_{spin} = 10^{-5}$, 95% CI = [−0.38; −0.55]), along which typical neurodevelopmental processes, including those associated with sex differences in functional brain architecture, have been shown to unfold[33–35]. Thus, greater feminization along the S-A axis (i.e., greater feminization of association, relative to sensor-imotor, regions) at T1 was observed among female adolescents showing more advanced PhenoAge and pubertal development (parti-cularly, gonadarche) at T2 (Fig. 4C–F). Complementarily, greater masculinization along the S-A axis (i.e., greater masculinization of association, relative to sensorimotor, regions) typified male

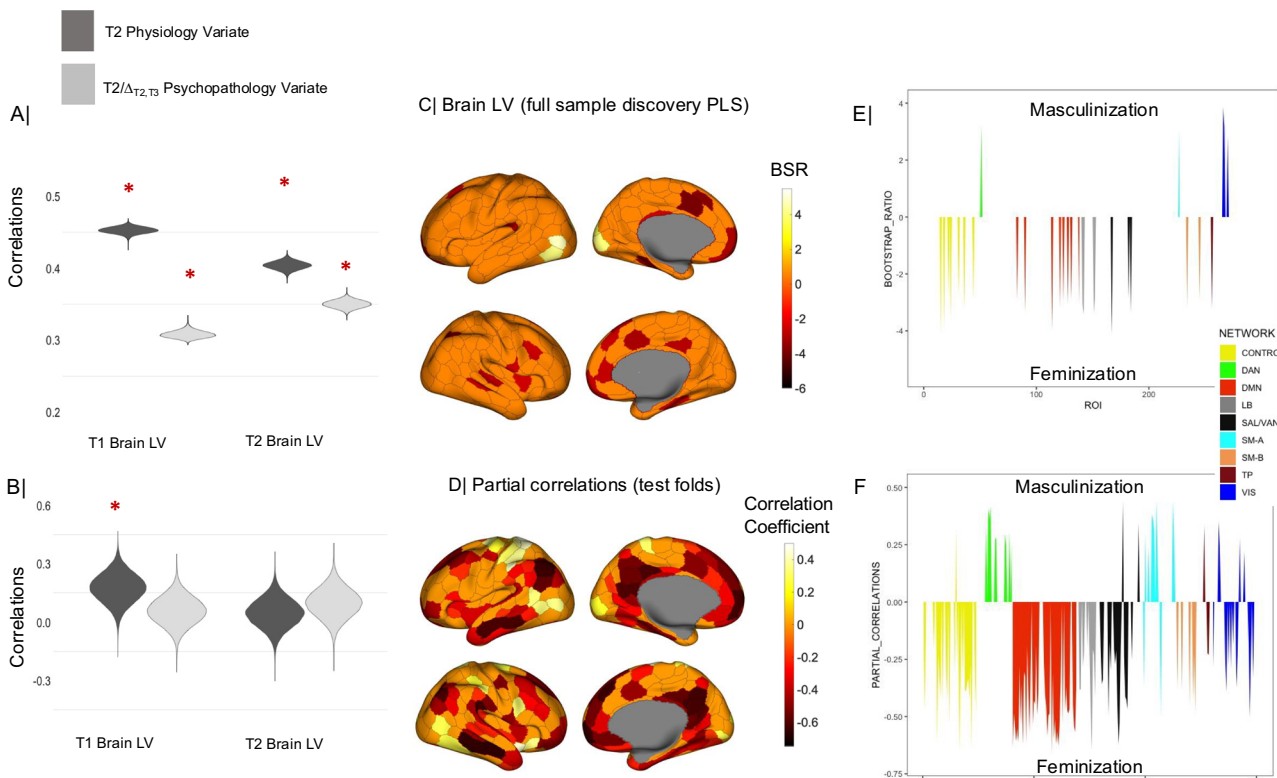

Fig. 4 | The brain LV from the behavioral-PLS analysis linking the psychopathology and physiology variates from CCA 1 (cf. Figure 3) to neural sexual differentiation in the ABCD sample. Panel A shows the correlations of the two CCA 1 variates with the Time 1 and Time 2 brain LV scores in the discovery PLS analysis. Panel B shows the correlations of the two CCA 1 variates with the predicted Time 1 and Time 2 brain LV scores (based on the 10-fold cross-validation procedure). A red asterisk indicates a robust correlation between the respective CCA 1 variate and the discovery (panel A) or predicted (panel B) brain LV scores across all participants. The violin plots in panels A and B depict the distribution of these correlation coefficients across the 100,000 bootstrap samples from the discovery (panel A) or cross-validated (panel B) PLS analysis. Panel C depicts the ROI-specific weights/loadings on the brain LV identified with the discovery PLS analysis with a bootstrap ratio greater than 2.75 in absolute value (equivalent to a 99% CI). Panel D depicts the Schaefer ROIs robustly correlated (based on cross-validated 99% confidence intervals) with the predicted value of the brain LV from the cross-

validation procedure. These are partial correlations controlling for the confounders listed under "Control variables". To facilitate interpretation, panels E and F present Schaefer network-based distributions of PLS weights (panel E) or partial correlations (panel F) summarizing the ROI-specific results from panels (C) and (D), respectively. The subnetworks from the Schaefer 17-network atlas (e.g., Control A/B/C) have been combined into one to increase comparability with the Gordon atlas. In panels C–F, positive values indicate masculinization, whereas negative values indicate feminization of the connectivity patterns associated with a specific ROI. As stated in the main text, sexual differentiation was estimated in reference to resting state connectivity data from the Human Connectome Project. BSR = bootstrap ratio. PLS = partial least squares. LV= latent variable. T1 Time 1, T2 Time 2, Schaefer networks: TP temporo-parietal, SAL-VAN salience/ventral attention, LB limbic, DMN default mode, DAN dorsal attention, SM-A somatomotor-A, SM-B somatomotor-B. VIS visual.

adolescents characterized by a younger PhenoAge and delayed pubertal development at T2.

## Analysis 3: Feminization of FC, PhenoAge, and Pubertal Maturation Sequentially Mediate the Link between Genetic Risk for Anxiety and Anxiety Symptoms

Our first two analyses provided evidence that physiology at ages 11-12 years shows longitudinal associations with earlier (but not concurrent) neural sexual differentiation (Fig. 4B) and increases in psychopathology, rather than concurrent psychiatric symptoms (Fig. 3A). Building on these findings, we next examined whether sexual differentiation in brain function and physiology sequentially mediate the relationship between genetic risk and psychopathology in the ABCD sample. To this end, we specified the serial mediation model depicted in Fig. 2 (Analysis 3) and tested with PROCESS 4.2, which is an ordinary least squares (OLS) and logistic regression path analysis modeling tool based on observable variables[47] (SI 7.3). In line with existing guidelines for probing causal chains[48], the serial mediators were sampled from different time points. Based on the results from our first two analyses, we were primarily

interested in whether the T1 neural sexual differentiation LV from Analysis 2 (mediator 1) and the sex-dependent physiology CCA variate from Analysis 1 (mediator 2) would constitute a viable causal chain linking polygenic risk scores (PRS) for anxiety and, potentially, depression to rising internalizing symptoms (cf Fig. 3A). In testing this sequence, we simultaneously accounted for alternate models involving serial (T1 and T2 neural sexual differentiation from Analysis 2) and parallel (T2 Physiology from Analysis 1 and T2 neural sexual differentiation from Analysis 2) mediation, respectively (see Fig. 2-Analysis 3). The absence of blood chemistry data at T1 prevented us from computing PhenoAge and, thus, estimating the alternate serial and parallel mediation models involving T1 Physiology. However, prior to mediation analysis, all variables were adjusted for T1 pubertal hormone levels and self-reported adrenarche/gonadarche based on the PDS (in addition to participant motion, age, race, testing site, handedness, adoption status, ambiguous biological sex). Additionally, inclusion of the PRS-T2 Physiology-Psychopathology path allowed us to estimate indirect effects of physiology that are independent of T1/T2 neural sexual differentiation and therefore uniquely related to T1 PhenoAge.

Analysis 3: Sequential Effect of Genetic Risk on Increasing Psychopathology via Neural Sexual Differentiation and Physiological Aging

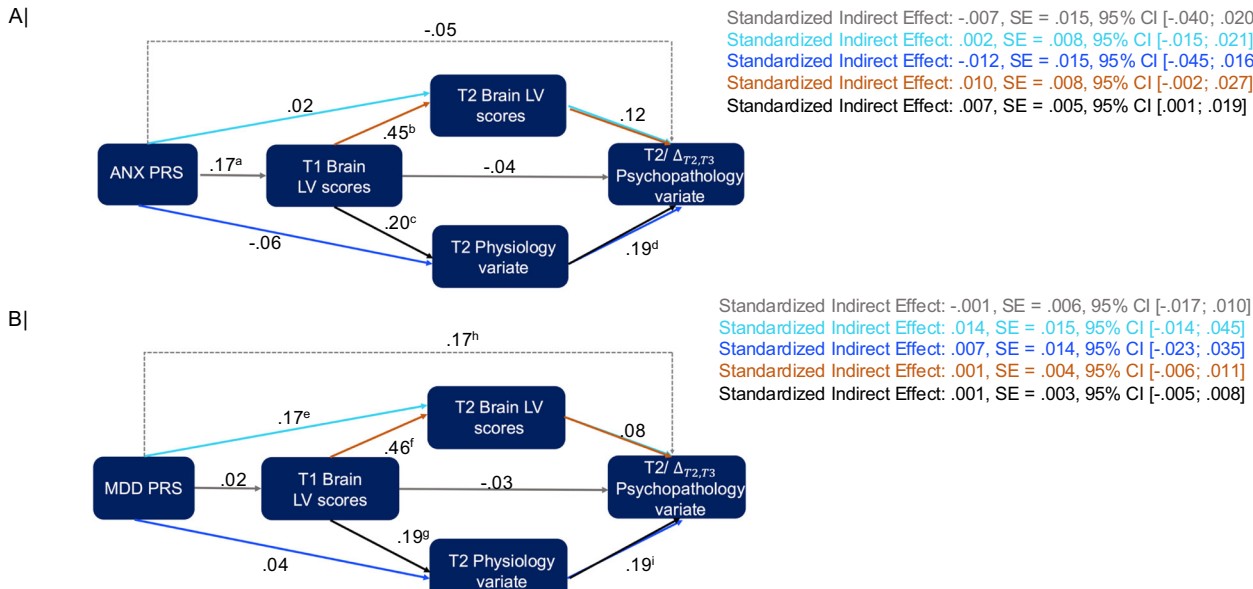

**Fig. 5 | Serial mediational model testing the roles of Time 1 neural sexual differentiation (cf Fig. 4) and the Time 2 CCA Physiology variate (cf. Fig. 3B), in mediating the impact of genetic risk on psychopathology (cf Fig 3A) in the ABCD sample.** The mediation model anchored in the ANX PRS is depicted in panel A, whereas panel B shows the mediation model anchored in the MDD PRS. In both panels, the indirect effect path of interest is indicated by black arrows, whereas the alternate paths, included as controls, are indicated by colored arrows. ANX = anxiety disorder. LV = latent variable. MDD = major depressive disorder. PRS = polygenic risk score. T1 = Time 1. T2 = Time 2. T3 = Time 3. 95% confidence intervals for the indirect effects were estimated using percentile bootstrap with 50,000 bootstrap samples. All *p*-values of the *t*-statistic associated with each path are two-tailed. [a]$p = 0.008$ [degrees-of-freedom = 197]. [b]$p = 10^{-8}$ [degrees-of-freedom = 196]. [c]$p = 0.004$ [degrees-of-freedom = 196]. [d]$p = 0.013$ [degrees-of-freedom = 194]. [e]$p = 0.008$ [degrees-of-freedom = 196]. [f]$p = 10^{-8}$ [degrees-of-freedom = 196]. [g]$p = 0.006$ [degrees-of-freedom = 196]. [h]$p = 0.030$ [degrees-of-freedom = 194]. [i]$p = 0.018$ [degrees-of-freedom = 194].

PRSs for anxiety, depression, and ADHD were computed using summary statistics from large and independent disorder-focused genome-wide association studies (GWASs), featuring case/patient-control comparisons (i.e., MDD[49], anxiety disorders[50], ADHD[51]). We did not include PRSs for ODD or CD because we could not locate relevant case-control GWASs. Based on the results from Analyses 1 and 2, robust indirect effects via neural sexual differentiation and physiology were expected for mediational models anchored in the anxiety PRS and, potentially, the depression PRS, whereas models anchored in the ADHD PRS were included for specificity analysis since ADHD did not have a robust loading on the psychopathology variate in Analysis 1 (Fig. 3A).

As expected, we observed a robust indirect effect of the anxiety PRS on the $T2/\Delta_{T2,T3}$ psychopathology variate via the cross-validated T1 brain LV (mediator 1) and the cross-validated T2 physiology CCA variate (mediator 2) (total standardized indirect effect: 0.007, SE = 0.005, 95%CI = [0.001; 0.019]) (Fig. 5A). No additional indirect effects were detected, which implies that any association between T1 Pheno-Age and later psychopathology, independent of T1 pubertal development (for which we controlled), would most likely be explained by its overlap with T1 neural sexual differentiation (for a replication of these results using different significance thresholds for the contributing SNPs and with Gordon atlas, see Figs. S5, S8).

The mediational analysis anchored in the MDD PRS yielded no robust indirect effects, but instead yielded a reliable direct effect (Fig. 5B,S6). This pattern of results implies that genetic risk for MDD predicts worsening depression and anxiety disorder-related symptoms from ages 11-12 to ages 12-13. However, this link between the MDD PRS and affective problems is not mediated by the neural sexual differentiation and physiology patterns herein identified. Finally, we confirmed the specificity of the sequence anchored in the anxiety PRS by re-running the serial mediation analysis with ADHD PRS as the predictor, which revealed no significant indirect effect (Fig. S7).

## Analysis 4: The link between regional FC feminization, advanced phenoage, pubertal maturation, and anxiety disorders replicates in an independent cross-sectional sample

In our final set of analyses, we tested whether the sex-dependent links between brain function, psychiatric risk, and physiological maturation, as established in the ABCD sample (Analyses 1–3) would extend cross-sectionally in the larger HCP-D sample. In Analysis 4, we therefore used CCA to probe whether individual differences in sexual differentiation of regional FC profiles (as defined using the PLS-extracted latent variable from Analysis 2) (variate 1) relate to scores on the CBCL scales, pubertal development, and markers of physiological aging (BMI, exposure to poverty) in a sex-dependent manner (variate 2). Of note, BMI, which had been robustly linked to PhenoAge in the ABCD sample, was also independently correlated with more advanced adrenarche/gonadarche in the ABCD (Spearman's *rho* = 0.29, 95% CI = [0.15; 0.40], permutation-based $p = 8 \times 10^{-5}$ [adrenarche] and Spearman's *rho* = 0.33, 95% CI = [0.15; 0.41], permutation-based $p = 10^{-5}$ [gonadarche]), as well as the HCP-D (Spearman's *rho* = 0.19, 95% CI = [0.06; 0.30], permutation-based $p = 0.002$ [adrenarche] and Spearman's *rho* = 0.32, 95% CI = [0.20; 0.42], permutation-based $p = 10^{-5}$ [gonadarche]) (correlations adjusted for sex and chronological age, similar to the CCA cross-validation analyses described below). Conversely, financial deprivation, which had been linked to PhenoAge in the ABCD sample

Analysis 4: Feminization of Brain Function, Psychopathology, Pubertal Development and Physiological Aging Markers: Sexually Dimorphic Associations (HCP-D)

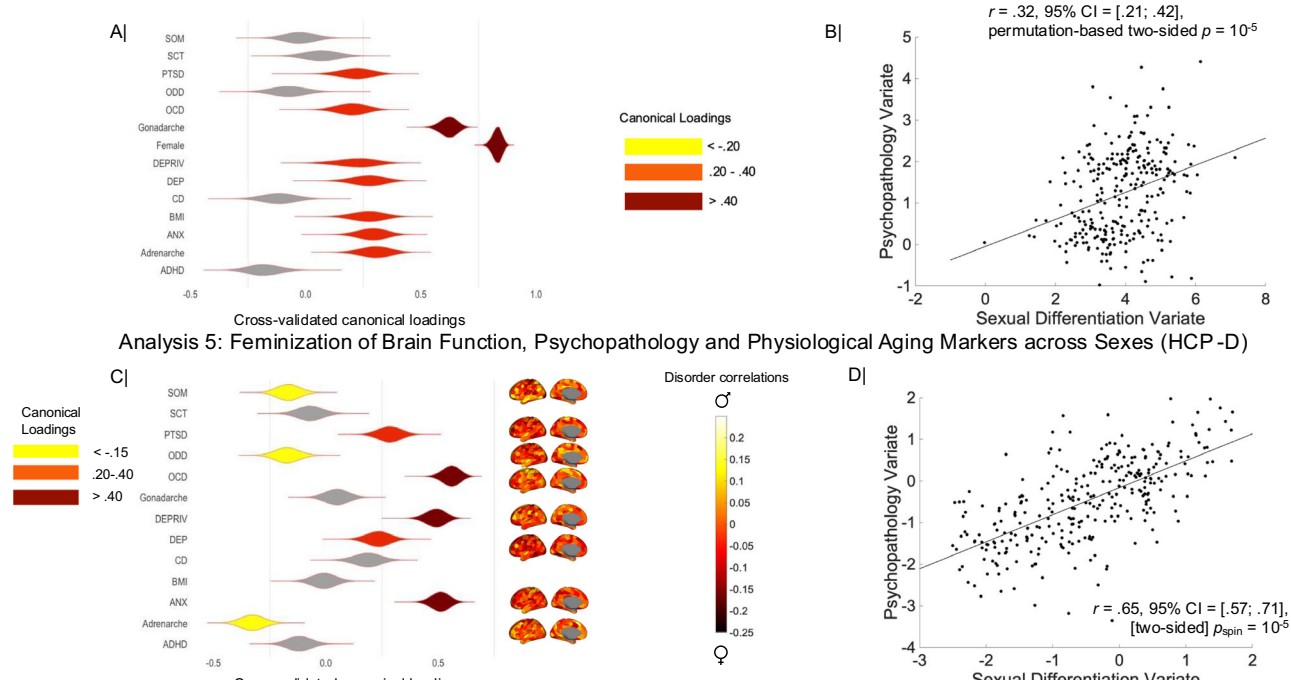

Analysis 5: Feminization of Brain Function, Psychopathology and Physiological Aging Markers across Sexes (HCP-D)

**Fig. 6 | The sexual differentiation brain LV from Analysis 2 linked by CCA to sex, individual differences in psychiatric disorder risk and maturation-relevant factors (panels A, B), as well as psychiatric disorder- and maturation-relevant neural sexual differentiation patterns in the HCP-D sample (panels C, D).** The horizontal graph in panel (**A**) contains the correlation coefficients describing the relationship between the observed disorder- and maturation-relevant scores and the predicted value of their corresponding canonical variate across all test CCAs. Panel (**B**) contains the scatter plot describing the linear relationship between the sexual differentiation brain LV (identified in Analysis 2 and projected onto the HCP-D sample) and the predicted values of the CCA variate from panel (**A**). The horizontal graph in panel (**C**) contains the correlation coefficients describing the relationship between the observed disorder- and maturation-relevant sexual differentiation scores and the predicted value of their corresponding canonical variate across all test CCAs. The shaded areas on the brain images reflect the strength of the partial correlation between the sexual differentiation scores and the disorder-

or maturation-relevant scores (see the Method for the confounders controlled for in the partial correlation). Panel (**D**) contains the scatter plot describing the linear relationship between the predicted values of the CCA variate from panel (**C**) and the unthresholded sexual differentiation brain map estimated in the ABCD sample (Analysis 2). In panels (**A**) and (**C**), the shaded areas correspond to robust correlations observed in cross-validated CCAs based on both the Schaefer and Gordon atlas data (based on the bootstrapping-derived 99% confidence intervals). In the same panels, the violin plots depict the distribution of partial correlation coefficients across the 100,000 bootstrap samples from the corresponding cross-validation procedures. ADHD attention deficit hyperactivity disorder. ANX anxiety, CCA canonical correlation analysis, CD conduct disorder, DEP depression, LV latent variable, OCD obsessive-compulsive disorder, ODD oppositional defiant disorder, PLS partial least squares, PTSD post-traumatic stress disorder, SCT slow cognitive tempo, SOM somatic disorder, DEPRIV financial deprivation, BMI body mass index.

was not robustly correlated with adrenarche/gonadarche in the HCP-D (Spearman's $rho = -0.12$, 95% CI = [−0.37; 0.05], permutation-based $p = 0.05$ [adrenarche] and Spearman's $rho = -0.04$, 95% CI = [−0.12; 0.10], permutation-based $p = 0.49$ [gonadarche]). In the ABCD sample, after controlling for PhenoAge, financial deprivation showed a modest correlation with adrenarche (Spearman's $rho = 0.16$, 95% CI = [0.01; 0.26], permutation-based $p = 0.028$), but not gonadarche (Spearman's $rho = 0.13$, 95% CI = [−0.12; 0.20], permutation-based $p = 0.074$). Hence, while BMI is a likely correlate of global aging processes, financial deprivation seems to be more specifically associated with the immune/metabolic dysregulation indexed by PhenoAge.

As in Analysis 1, sex-dependence was indicated by a robust loading of the sex variable on variate 2. Unavailability of pubertal hormone and blood chemistry data relevant to PhenoAge prevented us from including these variables in the HCP-D analyses. However, the main purpose of the cross-sectional HCP-D analyses was to probe the viability of neural sexual differentiation as a potential psychiatric vulnerability marker linked to observable individual characteristics such as BMI, and environmental variables, such as exposure to poverty.

For each HCP-D participant, we thus computed scores on the neural sexual differentiation LV linked to physiology and psychopathology in

the ABCD sample. The HCP-D discovery CCAs, linking these brain LV scores to the 9 DSM-oriented CBCL scores, sex assigned at birth, self-reported adrenarche/gonadarche, BMI, and financial deprivation unveiled a sole cross-validated variate pair ($r_{CV} = 0.32$, 95% CI = [0.21; 0.42], permutation-based $p = 10^{-5}$, Fig. 6B, adjusted for age, handedness, testing site, participant motion, and adoption status). In line with the longitudinal ABCD results, the CCA mode identified in the HCP-D sample indicated that greater feminization along the S-A axis was primarily detected among female youth showing more advanced pubertal development, particularly, gonadarche, higher BMI, greater exposure to financial deprivation, and more severe depression and anxiety disorder symptoms (Fig. 6A; for a replication of these effects with the Gordon atlas, see Fig. S9A, B).

**Analysis 5: FC feminization along the S-A axis distinguishes exposure to poverty and anxiety from behavioral, somatic and attentional disorders, regardless of biological sex**
Finally, in Analysis 5, we tested whether the ABCD profile of neural sexual differentiation could predict psychiatric risk not only in a sex-dependent manner (cf. Analysis 4), but also independently of the effects of sex. This analysis thus tests the utility of a continuous

approach to conceptualizing sex differences in brain function by shedding light on whether individual variability in relative brain feminization (or masculinization) can predict the same psychiatric symptoms across sexes.

Our final analysis linked the cross-validated sexual differentiation brain LV from the ABCD (Analysis 2, Fig. 4D) to the partial correlation maps quantifying the relationship between neural sexual differentiation and the DSM-oriented CBCL scores, pubertal development (i.e., adrenarche, gonadarche), and markers of physiological aging as identified in the ABCD (BMI, financial deprivation) (adjusted for sex, age, handedness, testing site, participant motion and adoption status). To account for similarity in neural sexual differentiation based on anatomical proximity, the correlation between the two cross-validated CCA variates was probed for significance using spatial null maps generated with a so-called spin permutation test[52–55]: (https://github.com/frantisekvasa/rotate_parcellation/commit/bb8b0ef10980f162793cc180cef371e83655c505).

The cross-validated CCA mode emerging from Analysis 5 ($r_{CV} = 0.65$, 95% CI = [0.57; 0.71], $p_{spin} = 10^{-5}$, Fig. 6D), indicated that, regardless of biological sex, greater feminization along the S-A axis correlated positively with exposure to poverty and mood problems, particularly, anxiety, whereas greater masculinization related to more advanced adrenarche, as well as increased behavioral and somatic disorder scores (Fig. 6C; for a replication of these effects with the Gordon atlas, see Fig. S9C, D).

## Discussion

Our analysis sheds light on how sexual differentiation of brain function mediates the relationship between genetic risk, physiology, and rising psychiatric symptoms in late childhood/early adolescence. Prior evidence implicated pubertal hormones and metabolic dysregulation in structural neurodevelopment[56,57] and mental health, particularly internalizing symptoms among female youth[21,22,27,28,58–62]. Here, we demonstrate that sexual differentiation along a previously identified sensorimotor-to-association (S-A) functional hierarchy[63], an alleged key organizing principle of neurodevelopment and sex differences in brain organization[33–35], shows distinct relationships with physiological age, reproductive maturation and affective vs behavioral symptoms among male vs female adolescents. Our results thus imply that regionally specific patterns of relative sexual differentiation could help elucidate the contribution of sensorimotor systems to anxiety disorders[64], and of transmodal systems to both anxiety[65], particularly threat learning and generalization[66], and metabolic dysregulation (e.g., insulin resistance[67]; gut microbiota composition and diversity[68]).

Leveraging the longitudinal ABCD dataset, we also characterized the temporal sequence underlying the sex differential links among neurodevelopment, physiology, and rising psychiatric symptoms. Specifically, we related reduced genetic risk for anxiety disorders to greater masculinization along the S-A axis at T1, as well as slower reproductive maturation and younger PhenoAge at T2 among boys. This event chain culminated in rising oppositional defiant and conduct disorder symptoms between T2 and T3. Conversely, we found that greater feminization along the S-A axis at T1 mediated the impact of genetic risk on faster pubertal development and more advanced PhenoAge at T2 among girls, a sequence leading to persistently greater somatic complaints, as well as rising anxiety and depression symptoms between T2 and T3. Together, these findings dovetail with cross-species reports of sex differences in the neurobiological circuits underpinning aggression[69] and internalizing disorders[70], as well as evidence on the robust neural signature of externalizing, but not internalizing, disorders in males[71]. The absence of robust indirect effects for the models anchored in the depression PRS (Fig. 5B) and ADHD PRS (Fig. S7) speak to the specificity of the mediational sequence linking genetic risk for anxiety disorders to worsening internalizing symptoms in early adolescence (Figs. 3A, 5A).

The sex differential event chain linking genetic risk to rising affective symptoms parallels evidence that sex differences in MDD prevalence stem from functional differences in the expression of MDD-relevant alleles, particularly in transmodal areas such as the hippocampus[72]. In our case (Fig. 5A), sex differences in anxiety symptoms may reflect functional differences in the expression of anxiety-relevant alleles, which yield distinct inter-relationships among brain, peripheral physiology, and pubertal development in males vs females. More broadly, our results echo cross-species findings that activity changes in threat-relevant neural circuits precede the immune/metabolic alterations predictive of subsequent anxiety symptoms[73,74]. Our mediational model, which included controls for alternate indirect effect paths (Fig. 5A), provides suggestive evidence on the interdependence of the neural sexual differentiation and later physiological/pubertal development in mediating the relationship between genetic risk for anxiety and rising internalizing symptoms in early adolescence. Given the correlational design of our study, our present results cannot shed light on underlying causal mechanisms. Cross-species experimental manipulations are needed to elucidate this matter.

The longitudinal ABCD results were replicated cross-sectionally in the HCP-D sample, where feminization along the S-A axis was observed among faster-developing females, with greater exposure to poverty, at risk for faster physiological aging (higher BMI) and showing more severe mood (particularly, anxiety) symptoms. Considering these findings in light of the Analysis 3 outputs, there emerges the possibility that neural sexual differentiation may constitute a marker of concurrent psychiatric vulnerability (cf. Analyses 4 and 5) which may also identify individuals most likely to show a trajectory of worsening depression and anxiety disorder-related symptoms. This line of inquiry is beyond the scope of the present analysis, but certainly warranting further study.

Underscoring the importance of a continuous approach to neural sexual differentiation[23], we further found that, regardless of biological sex, greater feminization along the S-A axis correlated with prior exposure to poverty, anxiety and, to a lesser extent, depression symptoms, whereas greater masculinization was related to more advanced adrenarche, as well as more severe somatic and behavioral problems. The results of Analyses 4 and 5 fit well with the broader literature on the associations among physiological aging, deprivation, BMI (e.g., via maladaptive stress coping mechanisms, such as emotional eating[75,76]), and health[77]. They further dovetail with evidence on the bidirectional relationship between BMI and internalizing symptoms that emerges in middle childhood[78], as well as the stronger link between BMI and deprivation observed in girls (relative to boys)[79].

It is worth noting that substantial efforts have been channeled towards characterizing the prototypical female vs male brain and developing sex-based classification algorithms using structural or functional neural features[80–83]. A complementary strand of research views human brains as comprising a mosaic of characteristics, some of which are more common in females, some in males, and some occurring equally between the sexes[84]. While more closely aligned with the latter approach, our work diverges from both lines of research. Specifically, our focus is on determining whether the relative feminization/masculinization of functional connectivity patterns observed in the same brain is independently related to sex and aging-relevant variables (i.e., PhenoAge, pubertal development). Put differently, we sought to elucidate whether differential susceptibility to internalizing vs externalizing disorders can be explained by patterns of within-brain sexual differentiation, which are independently correlated with sex and aging processes.

### Limitations and future directions

Our analysis featured sex-independent genetic risk markers. Future studies incorporating sex-specific markers, which are reportedly better at predicting complex traits (e.g., immune/metabolic dysregulation[85], psychiatric disorders[86]) may further illuminate the effects herein

documented. We focused on global PhenoAge estimates based on organ-level measures of immune and metabolic functioning. Examination of specific inflammatory markers, derived from different tissues, cell types (e.g., mitochondria[87,88], microglia[29]) and focusing on different life stages (cf.[89]) could improve the monitoring of stability and change in mood symptoms. Furthermore, development of personalized, sex- and age-specific, multimodal systemic aging markers, combining physiological (e.g., peripheral and central inflammation[90]) and epigenetic (e.g., DNA methylation)[91] measures could help fine-tune predictions of functional outcomes following exposure to different types of stressors across the life course[92]. Relatedly, understanding the mechanisms underpinning physiological wear-and-tear among youth raised in adverse environments, including sex differences in the intergenerational transmission of adversity sequelae (e.g., via parental gut microbiota[93] or maternal placenta in the final stages of pregnancy[94]) and the potential moderating effect of psychosocial factors, such as parenting[95] are important goals for future investigations. Indeed, Analysis 5 suggested that our identified patterns of neural sexual differentiation covary with financial deprivation across both male and female individuals. Consequently, the sex-dependent associations among brain function, physiology, and psychopathology described in this report may partly reflect sex differences in adversity exposure, including intergenerational transmission of adversity sequelae (cf.[96,97]), a possibility that would certainly warrant further study. Lastly, our three samples rated their sex in a binary manner, and only one participant in each of the two adolescent cohorts indicated that their gender is different from the sex assigned at birth (no information on gender– as distinct from biological sex– was collected in the HCP). Therefore, some of the sex differences herein reported[98] may stem from gendered experiences, including those resulting from gender differences in societal expectations that may extend to potential biases in psychiatric diagnosis. Future studies featuring samples that allow for meaningful dissociations between sex and gender[99] and which probe the interactive impact of multiple social identities[100] would provide further insights into the effects herein documented.

In sum, we characterized regionally specific patterns of sexual differentiation in brain function along a canonical unimodal-to-transmodal functional hierarchy. We demonstrated that individual differences in such differentiation mediate the association between genetic risk and rising internalizing vs externalizing symptoms in adolescence, via their effect on physiological aging. The identified patterns of FC sexual differentiation also tracked the cross-sectional severity of internalizing symptoms in a sex-dependent and sex-independent manner in an unrelated sample. By characterizing cross-sectional and longitudinal relationships between sexual differentiation of the S-A axis, sex differential patterns of pubertal and physiological aging, and psychiatric vulnerability, this study sheds light on a potential path towards more personalized interventions targeting affective and behavioral pathology across sexes.

## Methods

### Participants

Our samples were selected from three publicly available datasets. All participants contributed data on all scrutinised variables. The participant selection process for the two developmental samples is described in Fig. 7. The study was approved by the local ethics committee of each dataset, and written informed consent was obtained from each participant or their authorized legal representative/guardian (the latter for ABCD and HCP-D youth). For the ABCD study, the research ethics boards were Children's Hospital Los Angeles, Florida International University, Laureate Institute for Brain Research, Medical University of South Carolina, Oregon Health & Science University, SRI International, University of California San Diego, University of California Los Angeles, University of Colorado Boulder, University of Florida, University of Maryland at Baltimore, University of Michigan, University of

Minnesota, University of Pittsburgh, University of Rochester, University of Utah, University of Vermont, University of Wisconsin-Milwaukee, Virginia Commonwealth University, Washington University in St. Louis and Yale university. For the HCP-D study, the ethics boards were Harvard University, University of California Los Angeles, University of Minnesota, and Washington University in St. Louis. For the HCP study, the ethics board was Washington University in St. Louis. Participants in all three samples were compensated for time and travel (e.g., parking, mileage, cab to and from the study site) expenses. The children in the HCP-D study also received gift cards for their participation.

### Target developmental groups

**Adolescent Brain and Cognitive Development (ABCD).** The sample included biological parents and offspring who participated in the ongoing Adolescent Brain Cognitive Development (ABCD) study (for a detailed sample description, see ref. 101). The present research uses baseline, two-, and three-year follow-up data downloaded in May 2024 as part of the ABCD Study Curated Annual Release 5.1. For the resting state data, we downloaded the package recommended for use by the ABCD study team (as flagged on the NDA site).

Our sample included 199 adolescents (88 biological females), the majority of whom were predominantly right-handed ($N = 152$) and identified by their parents as being White (82.4%), although other racial backgrounds were also represented (3% Native American Indian, 1% Asian Indian, 13.6% Black, 1% Chinese, 2% Filipino/Filipina, 0.5% Japanese, 0.5% Korean, 0.5% Pacific Islander, 0.5% Vietnamese, 6.5% other or not reported racial background). The parents could select multiple racial backgrounds for their child. Biological sex was based on parental reports of sex assigned at birth, which were corroborated with menstruation history data (youth and parental reports) from the two-year follow-up and SNP analysis (yielding the SNPSEX variable) using the PLINK command "plink –bfile data –check-sex". A dummy-coded variable was created to account for the two youths for which sex assigned at birth was not corroborated through menstruation history or SNP analysis. Participants were aged 9-10 years at baseline ($M_{base} = 118.74$ months, $SD_{base} = 7.81$ months; $M_{2-yr} = 143.10$ months, $SD_{2-yr} = 8.05$ months; $M_{3-yr} = 154.65$ months, $SD_{3-yr} = 8.01$ months).

**Human connectome project-development (HCP-D).** We included data from 277 HCP-D participants (145 females, 90.3% predominantly right-handed, ranging in age from 8 to 17 years ($M = 171.78$ months, $SD = 22.53$ months). Sex was determined from parental reports of sex assigned at birth and corroborated through menstruation history for 108 of the 137 female participants aged 12 and older. Based on parental reports of the participants' racial background, the sample was predominantly White (66.8%), although other racial backgrounds were also represented (0.4% American Indian/Alaska Native, 4.7% Asian, 8.3% Black, 18.4% mixed race, 1.4% other or not reported racial background). A detailed description of the HCP-D sample is available in ref. 102.

### Adult reference group

**Human connectome project (HCP).** Sexual differentiation in brain function was characterized in reference to the FC patterns observed among the young healthy adult participants in the HCP who were expected to show peak levels of neural sexual differentiation related to reproductive maturity[12,16–18]. Sex assignments were based on self-reported sex assigned at birth which was also corroborated through self-reported menstruation history (cf[42,43]).

We only included individuals aged 22–30 years (22–25 years: $N = 141$, 63 females; 26–30 years: $N = 195$, 96 females) in light of evidence that neurocognitive performance starts showing subtle signs of decline following the age of 30[95,96]. The sample, who was predominantly right-handed ($N = 300$), had the following racial composition: 7.7% Asian, 13.1% Black, 3.3 % more than one race, 73.5% White and 2.4% not reported.

## STROBE Flow Diagram

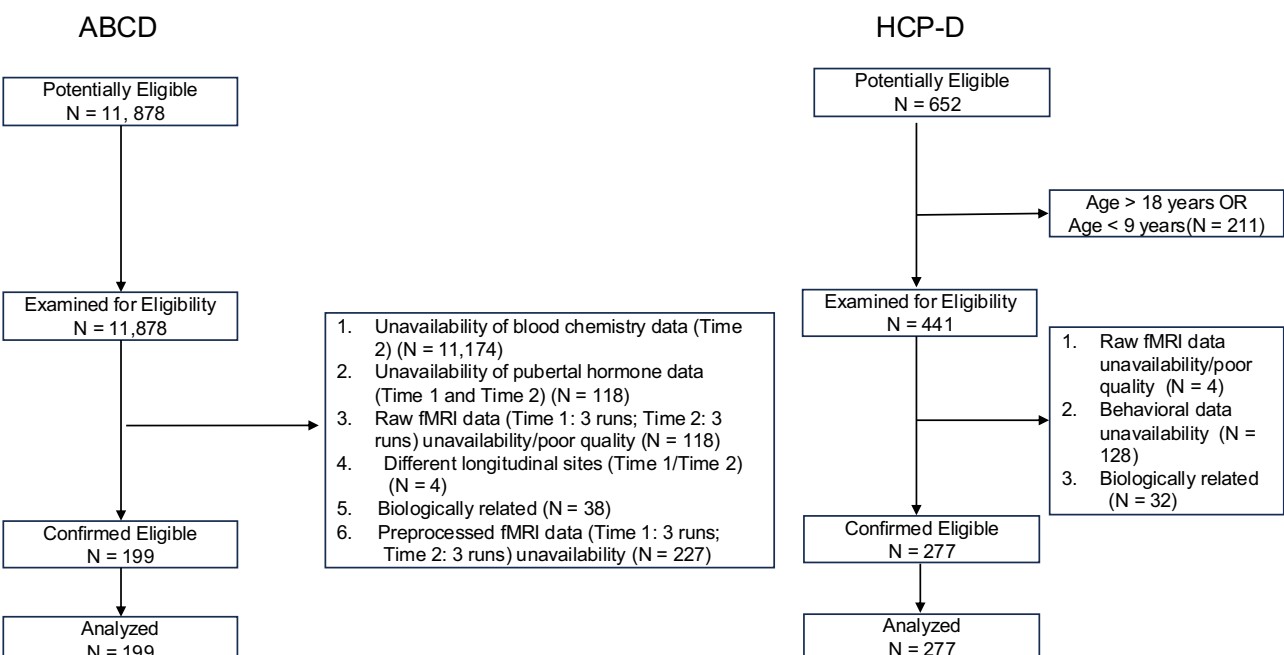

**Fig. 7 | STROBE flowchart describing sample selection for the ABCD and HCP-D cohorts.** For the resting state fMRI data from the ABCD study, "good quality" was defined based on the "_pc_score" (corresponding to each of the 3 runs analyzed) and the "iqc_rsfmri_ok_ser " variable in the "mri_y_qc_raw_rsfmr.csv" file from the data release 5.1 as participants who had at least 3 complete protocol compliant time series who passed quality control at both Time 1/ages 9–10 and Time 2/ages 11–12.

## Pubertal development

**ABCD.** In line with prior work[37,103], pubertal maturation was assessed via self-ratings on the Pubertal Development Scale (PDS[36]) and hormonal assays, specifically, dehydroepiandrosterone (DHEA), testosterone and, for female youth only, estradiol. Based on self-reports and hormonal measures, we independently estimated progression through the two component processes of puberty, adrenarche and gonadarche. Adrenarche is the earlier onset process, which starts around ages 5–7 and is characterized by release of adrenal hormones, which, in turn, support the development of secondary sex characteristics, such as pubic hair growth and acne, in both male and female youth (for review of relevant findings, see refs. 37,38). Maturation of the hypothalamic-pituitary-gonadal (HPG) axis marks the onset of gonadarche, in which the release of sex hormones (i.e., estradiol in girls, testosterone in boys) leads to reproductive maturation and the development of sex-specific characteristics[38]. Following prior studies[37,38,104], in the PDS, adrenarche was estimated via self-reported skin changes and body hair growth for both boys and girls, whereas gonadarche was estimated via self-reported growth spurt, breast development and menarche in girls, and self-reported growth spurt, deepening of the voice, and facial hair growth in boys. At Time 1 only (ages 9-10), 9 youths failed to complete any of the adrenarche-relevant items, and 5 youths failed to complete any of the gonadarche-relevant items on the PDS. For these participants, at Time 1 only, we used the corresponding parental ratings.

Pubertal hormone levels were extracted from saliva samples, which had been collected using the passive drool method. Upon collection, saliva samples were stored at −20 to −80 °C before being shipped on dry ice to Salimetrics, which evaluated the quality of all the samples and extracted pubertal hormone levels using replicate analyses (for details, see https://salimetrics.com/ and reference[105]). For each pubertal hormone, we analysed its mean level across all the

replicate analyses. Hormonal markers of adrenarche and gonadarche were estimated separately for boys and girls. In boys, z-scored DHEA levels were used as a marker of adrenarche, whereas in girls, adrenarche was estimated as the average of the standardized (i.e., z-scored) DHEA and testosterone levels, because existing evidence links testosterone more strongly to adrenarche, rather than gonadarche, among female youth[37]. Gonadal hormone scores were computed based on hormones that had been directly implicated in sex-specific reproductive maturation processes. Thus, z-scored estradiol levels indexed female gonadarche, whereas z-scored testosterone levels indexed male gonadarche (for a review of supporting findings, see refs. 37,38).

Using the information available in the "ph_y_sal_horm.csv" file from the ABCD data release 5.1, hormonal indices of adrenarche and gonadarche were residualized separately within each sex for saliva quality-related concerns (i.e., low quantity, discoloration, excessive bubbles, potential contamination, other concerns), activity levels and caffeine quantity in the 12 h preceding saliva collection and the delay between waking time and saliva collection (cf.[105]). The saliva quality variable was dummy-coded as "1" ("no concerns") or "0" ("concerns present") since only 4 youths at T1 and 2 youths at T2 showed more than one saliva quality-related issue. Across the full sample, concerns for the saliva samples were noted for 20 youths at T1 and 12 youths at T2. For female youths, the hormonal indices were further residualized by self-reported age at menarche since, in samples as young as ours who had recently experienced menarche, age at menarche relates to regularity of the menstrual cycle and associated hormonal fluctuations[106], days since the start of the last menstrual cycle, and whether the menstrual cycles are regular (based on self- and parental reports as a subjective complement to the cycle regularity information gauged via self-reported age at menarche) (cf.[105]). None of the female participants reported taking birth control pills. The hormonal indices of adrenarche and gonadarche, residualized and standardized (i.e., z-

**Table 1 | Summary Statistics for the Pubertal Development Measures Collected at Time 1 and Time 2**

| | ABCD (N = 199) | |
|---|---|---|
| Variable | Time 1 (Ages 9-10) Female M (SD, range)/Male M (SD, range) | Time 2 (Ages 11-12) Female M (SD, range)/Male M (SD, range) |
| DHEA*(pg/mL) | 77.20 (43.14, 5.09-188.09)/61.08 (41.62, 8.00-215.65) | 115.53 (67.74, 13.50-370.10)/84.44 (69.32, 3.12-314.47) |
| Testosterone* (pg/mL) Estradiol* (pg/mL) Adrenarche (self-report) | 37.26 (16.06, 5.22-73.00)/36.97 (23.58, 8.41-224.03) 1.01 (.45, .13-2.53) 1.61 (.62, 1-3.50)/1.60 (.61, 1-4) | 51.71 (21.31, 14.68-112.35)/54.63 (33.51, 11.73-168.58) 1.15 (.63, .34-5.29) 2.37 (.73, 1-4)/1.97 (.72, 1-4) |
| Gonadarche (self-report) Adrenarche (parental report) Gonadarche (parental report) | 1.62 (.55, 1-3)/1.66 (.53, 1-4) 1.56 (.60, 1-3)/1.36 (.53, 1-3) 1.78 (.39, 1-3)/1.58 (.42, 1-3) | 2.26 (.75, 1-4)/1.83 (.52, 1-3) 2.35 (.76, 1-4)/2.04 (.77, 1-4) 2.47 (.68, 1-3.67)/1.84 (.61, 1-4) |

M mean, SD standard deviation, DHEA Dehydroepiandrosterone. *These are the raw values before residualization by the confounds listed in the text.

scored) separately within each sex, were combined across the full sample and entered in the reported analyses. Table 1 contains summary statistics for hormonal indices, as well as youth and parental ratings of pubertal development. At Time 2, partial correlation analyses controlling for sex revealed a robust association between self-reported adrenarche and the adrenal hormonal index (DHEA[boys]/DHEA/testosterone [girls], Spearman's *rho* of *r* of 0.27, 95% CI = [0.12; 0.39], permutation-based $p = 1.5 \times 10^{-4}$), as well as the gonadal hormonal index (testosterone [boys]/estradiol [girls], Spearman's rho of *r* of 0.35, 95%CI = [0.21; 0.47], permutation-based $p = 10^{-5}$). At the same time point, a robust correlation (adjusted for sex) was detected between self-reported gonadarche and the gonadal hormone index (Spearman's rho of *r* of 0.29, 95% CI = [0.13; 0.39], permutation-based $p = 10^{-4}$) as well as the adrenal hormone index (Spearman's rho of *r* of 0.24, 95% CI = [0.10; 0.35], permutation-based $p = 0.001$). In contrast, at Time 1, while self-reported adrenarche was robustly correlated with both the adrenal hormone index (Spearman's rho of 0.17, 95% CI = [0.01; 0.29], permutation-based $p = 0.015$) and the gonadal hormone index (Spearman's rho of 0.19, 95% CI = [0.03; 0.30], permutation-based $p = 0.009$), self-reported gonadarche showed a statistically significant correlation only with the adrenal hormone index (Spearman's rho of 0.19, 95% CI = [0.02; 0.30], permutation-based $p = 0.006$).

**HCP-D**. Hormonal measures of pubertal development are unavailable in the latest existing data release (2.0) from the HCP-D. Hence, to index reproductive maturation, we relied on self-reports of adrenarche (Female youth: M = 2.87, SD = 0.70; Male youth: M = 2.47, SD = 0.67; score range: 1–4) and gonadarche (Female youth: M = 3.09, SD = 0.87; Male youth: M = 2.19, SD = 0.72; score range: 1–4) which were strongly correlated (Spearman's rho of *r* of 0.71, 95% CI = [0.60; 0.77], permutation-based $p = 10^{-5}$ [female youth] and Spearman's rho of *r* of 0.54, 95% CI = [0.37; 0.66], permutation-based $p = 10^{-5}$ [male youth]) (parental reports were not available for the full sample).

**Physiological aging**
**ABCD**. Physiological wear-and-tear was quantified with the BioAge R (https://github.com/dayoonkwon/BioAge) implementation of the PhenoAge algorithm[39]. This algorithm has been validated as a predictor of mortality, morbidity, and healthspan in younger and older adults[19,20,39,107–109]. The PhenoAge algorithm uses an exponential growth formula to predict mortality from blood chemistry measures of metabolic and immune system functioning in a reference group. An individual's PhenoAge biological age prediction corresponds to the chronological age at which their mortality risk would be normal in the reference group. Following existing guidelines in the literature[20,39,40], the PhenoAge algorithm was applied to a training set of young adults (N = 6084; aged 20–40 years; 2892 males; tested in 1991), who had data on all the biomarkers of interest (see below) and had been selected from the 18,825 participants (aged 20–90 years) in the National Health

and Nutrition Examination Survey (NHANES) III (https://wwwn.cdc.gov/nchs/nhanes/Default.aspx) included in the BioAgeR package. The predictions of the PhenoAge algorithm were validated in a test set of young adult (N = 14,782; aged 20–40 years; 7421 males; tested from 1999 to 2017) participants in the National Health and Nutrition Examination Survey (NHANES) IV who are also available in the BioAge R package. We focused on young adult subsamples for whom, where applicable, death was related to intrinsic causes (e.g., cardiac, cerebrovascular, pulmonary, kidney-linked, diabetes-linked), rather than accidents because we reasoned that they would best reflect physiological wear and tear processes that are distinguishable from those linked to typical aging and, thus, most likely to be observed among the adolescents in the ABCD cohort.

Based on availability in the ABCD sample, the biomarker set for PhenoAge included 8 blood chemistry variables (i.e., mean cell volume [mcv], red blood cell count [rbc], red cell distribution width [rdw], white blood cell count [wbc], lymphocyte percent, monocyte percent, glycohemoglobin [hba1c], and total cholesterol). These biomarkers reflect metabolic and immune function, and had been used before to quantify physiological aging across the adult lifespan[19,20,39,107,108,110]. In the NHANES III and IV samples, we verified that the original and our modified biomarker set yielded nearly identical age predictions with PhenoAge (*r*s of 0.92 and 0.93, respectively). Moreover, in both NHANES samples, more advanced PhenoAge than expected based on chronological age, which tended to be observed among poorer individuals (Spearman's *rho* (5611) = 0.16, 95% CI = [0.14; 19], $p = 7.71 \times 10^{-34}$ [NHANES III] and $rho(14780) = 0.16$, 95% CI = [0.14; 0.18], $p = 2.5 \times 10^{-85}$ [NHANES IV]), was linked to higher body mass index (BMI) (Spearman's *rho* (6075) = 0.22, 95% CI = [0.20; 24], $p = 4.81 \times 10^{-67}$ [NHANES III] and $rho(14780) = 0.28$, 95% CI = [0.27; 30], $p = 1.41 \times 10^{-265}$ [NHANES IV]) and poorer overall health (Spearman's *rho* (6082) = 0.15, 95% CI = [0.13; 18], $p = 2 \times 10^{-32}$ [NHANES III] and $rho(12406) = 0.20$, 95% CI = [0.18; 22], $p = 1.8 \times 10^{-113}$ [NHANES IV]).

**Validation of PhenoAge in the ABCD sample**. To show that PhenoAge is similarly predictive of physiological wear and tear in early adolescence and young adulthood, we used the largest ABCD sample of biologically unrelated youths with available data on blood chemistry, BMI, financial deprivation at the two- or three-year follow-up (N = 922). In this subsample, we ran a CCA with 10-fold cross-validation linking PhenoAge (based on the model estimated in the NHANES III sample) to concurrent BMI, parental reports of financial deprivation (i.e., difficulty providing for basic needs assessed with the 7-item scale developed by[111]) in the preceding year and parental reports of medical history covering the study period (i.e., from the 1-year follow-up reports that extended back to the start of the study [i.e., when participants were aged 9-10] to 3-year follow-up reports). We included medical history items gauging serious physical health problems, such as number of emergency room visits and unplanned medical visits related to high

fever, very bad headaches, head injuries, having been knocked unconscious, asthma, allergies, bronchitis, cancer, cerebral palsy, diabetes, seizures, hearing problems, kidney disease, lead poisoning, muscular dystrophy, multiple sclerosis, vision problems, heart problems, and anemia. The cross-validated CCA mode emerging from this analysis (Spearman's *rho* = 0.35, 95% CI = [0.29; 0.41], permutation-based $p = 10^{-5}$, adjusting for sex and chronological age) linked more advanced PhenoAge to higher BMI, greater financial deprivation, a higher number of emergency room visits and unplanned medical visits, particularly those related to high fever, very bad headaches, asthma, allergies, bronchitis and diabetes (see Fig. S1A for a representation of the physical health CCA variate). Because not all participants had medical history data at all 3 time points, we re-ran the CCA using the 916 youths who had complete medical history data from the year preceding the collection of the blood chemistry data for Pheno-Age (i.e., medical history data from the 1-year follow-up for those with PhenoAge data at the 2-year follow-up and medical history data from the 2-year follow-up for those with PhenoAge data at the 3-year follow-up). The findings from the first CCA on physical health were replicated (see Fig. S1D for a representation of the physical health CCA variate). Finally, we projected the two physical health CCA variates (Fig. S1A, B, D, E) onto our target sample of 199 ABCD participants and replicated their association with PhenoAge observed in the larger samples of 922 and 916 participants, respectively (Spearman's *rho*s of 0.28, 95% CI = [0.14; 0.41], and 0.29, 95% CI = [0.14; 0.41], respectively, both *p*s = 0.0001, see Fig. S1C, F).

**HCP-D.** Blood biomarkers are not included in the latest existing data release (2.0) from the HCP-D. Therefore, capitalizing on their robust correlations with PhenoAge in the ABCD sample, we used estimates of financial deprivation (assessed identically as in the ABCD study[111]) and BMI as markers of (likely) greater systemic wear-and-tear.

### Psychological Functioning

In both developmental samples, psychological functioning was assessed through parental responses on the Child Behavior Checklist (CBCL[32]). The analyses featured scores on all the DSM-oriented scales that are part of the default CBCL output and quantify variations in Depression (DEP), Anxiety (ANX), Attention Deficit Hyperactivity Disorder (ADHD), Somatic Disorder (SD), Oppositional Defiant Disorder (ODD), Conduct Disorder, (CD) Sluggish Cognitive Tempo (SCT), Obsessive-Compulsive Disorder (OCD) and Post-traumatic Stress Disorder (PTSD) (see Table 2 for summary statistics). To provide a more fine-grained description of brain-psychopathology relationships, we focused on disorder-specific, rather the composite Internalizing vs Externalizing scores which are also part of the CBCL output. Parental responses collected concurrently with brain and physiology-relevant measures (ABCD: T2/two-year follow-up; HCP-D: one time point) assessed each adolescent's standing on mental health relative to the remaining sample. In the longitudinal ABCD dataset we also assessed rise/decline in a participant's standing on each DSM-oriented scale from the two- to the three-year follow-up (i.e., $\Delta_{T2, T3}$). The scales showed acceptable to very good reliability (ABCD: Cronbach's alphas from 0.56 to 0.83 across both the two- and three-year follow-up assessments; HCP-D: Cronbach's alphas from 0.51 to 0.74).

### Sexual differentiation in brain function

**Resting state scans.** Four resting state fMRI scans (eyes open with passive crosshair viewing), each lasting approximately 15 min (HCP), 6 min (HCP-D) and 5 min (ABCD), respectively, were collected from all three samples (ABCD: baseline/2-year follow-up; HCP/HCP-D: 1 time-point). For the ABCD participants, due to data availability, we used only 3 scans from each time point (6 resting state scans in total) for a total of approximately 15 min at each timepoint. Analyses were conducted on

the same number of volumes/similar duration data for all three samples at each available time point (see below for details).

### fMRI data acquisition

**ABCD.** Participants were scanned across 21 US sites, with a protocol harmonized for Siemens Prisma, Philips, and GE 3 T scanners. Scanner type was controlled for in all analyses by using site id as a covariate to account for magnet and sociodemographic differences among sites. The fMRI data were acquired with a multiband EPI sequence (TR = 800 ms, TE = 30 ms, flip angle = 52°, FOV = 216 × 216 mm, 60 slices of 2.4 × 2.4 mm in-plane resolution, 2.4 mm thick, multiband acceleration factor of 6).

**HCP-D.** Scanning was performed across 4 US sites on Siemens Prisma 3 T scanners (32-channel coil; for details, see ref. 112). The fMRI data were acquired with a multi-band gradient-recalled (GRE) EPI sequence (TR = 800 ms, TE = 37 ms, flip angle = 52°, FOV = 208 mm, 104 × 90 matrix, 72 oblique axial slices, 2 mm isotropic voxels, multiband acceleration factor of 8).

**HCP.** Participants were scanned with a customized Siemens 3 T Connectome Skyra scanner housed at Washington University in St. Louis. Functional images were acquired with a multiband EPI sequence (TR = 720 ms, TE = 33.1 ms, flip angle = 52°, FOV = 208 mm, 104 × 90 matrix, 72 slices of 2 × 2 mm in-plane resolution, 2 mm thick, no gap; multiband acceleration factor of 8). Two runs were acquired with a left-to-right (LR) and the other two with a right-to-left (RL) phase encoding sequence[113].

### fMRI data preprocessing

**ABCD.** Analyses were conducted on minimally preprocessed resting state fMRI data, which was available as part of the ABCD Study Curated Annual Release 5.0. Using a Multi-Modal Processing Stream[114], which primarily combines MATLAB, Freesurfer[115], FSL[116] and AFNI[117] functions, these data had been corrected for head motion, spatial and gradient distortions, bias field removal, and the cleaned functional images had been co-registered to the participant's T1-weighted structural image. Using FSL and MATLAB, we further applied the following steps: (1) elimination of initial volumes (8 volumes [Siemens, Philips], 5 volumes [GE DV25, 16 volumes [GE DV26] to allow the MR signal to reach steady state equilibrium, (2) linear regression-based removal of the mean time courses of cerebral white matter (WM), gray matter (GM), cerebrospinal fluid (CSF), as well as the quadratic trends and 24 motion terms (i.e., the six motion parameters, their first derivatives, and squares) from the time course of each parcel. Prior to being regressed, the motion terms had been filtered to eliminate signals within the respiratory effect range, a step that had been shown to lead to more effective removal of head motion-related artifacts (i.e., 0.31-0.43 Hz, cf.[118]). To verify the effectiveness of the denoising pipeline, we estimated QC-FC correlations (i.e., correlations between participant-level framewise displacement and each ROI-to-ROI functional connectivity index, see "ROI definition and correlations" below). The distribution of the QC-FC correlations and the scatterplots describing the distance-dependence of the motion artifacts are included in Fig. S10. The observed metrics for the QC-FC correlations paralleled those of the best performing pipelines described in ref. 119 (see also[120]). To remove any potential lingering artifacts, all group-level analyses controlled for average framewise displacement per participant (see section titled "Control variables" which also contains details on additional motion control analyses).

**HCP-D and HCP.** We analyzed data already preprocessed by the respective study teams using the HCP Preprocessing Pipelines, specifically, the Generic fMRI Volume and Surface Processing Pipelines,

**Table 2 | Descriptive Statistics for the CBCL Subscales Analyzed in the Two Developmental Samples**

| DSM Disorder | ABCD (N = 199) | | HCP-D (N = 277) |
|---|---|---|---|
| | Time 2 (Ages 11–12) M (SD) | Time 3 (Ages 12–13) M (SD) | M (SD) |
| Depression | 0.11 (0.16) | 0.13 (0.18) | 0.10 (0.14) |
| Anxiety | 0.18 (0.23) | 0.21 (0.27) | 0.17 (0.21) |
| Somatic | 0.14 (0.19) | 0.14 (0.19) | 0.12 (0.17) |
| ADHD | 0.30 (0.38) | 0.30 (0.37) | 0.20 (0.27) |
| ODD | 0.25 (0.35) | 0.25 (0.32) | 0.30 (0.35) |
| CD | 0.05 (0.11) | 0.05(0.12) | 0.06 (0.10) |
| SCT | 0.13 (0.23) | 0.13 (0.23) | 0.16 (0.26) |
| OCD | 0.15 (0.20) | 0.18 (0.23) | 0.15 (0.17) |
| PTSD | 0.17 (0.20) | 0.20 (0.24) | 0.18 (0.19) |

*M* mean, *SD* standard deviation, *ADHD* attention deficit hyperactivity disorder, *CD* conduct disorder, *OCD* obsessive compulsive disorder, *ODD* oppositional defiant disorder, *PTSD* posttraumatic stress disorder. *SCT* slow cognitive tempo.

multi-run independent component analysis (ICA) FIX denoising and multimodal surface matching registration (cf.[121,122]).

### ROI definition and correlations

To characterize similarity between adolescent and adult male vs female FC patterns, we estimated pairwise Pearson correlations between regional time series extracted from the Schaefer 300 parcel/ 17-network functional atlas[44] for each HCP, ABCD and HCP-D participant, respectively. The correlation analyses were conducted in Matlab (version 2024a) and the resulting coefficients were Fisher's z-transformed. In each sample, the correlations were run separately for each run using a duration of approximately 5 min (417 volumes for HCP; 375 volumes for ABCD/HCP-D). The resulting ROI-to-ROI (300 × 300) matrices, each estimated over a roughly 5-min interval for all three samples, were averaged across the 3 (ABCD) or 4 (HCP/ HCP-D) runs available at each time point.

### Gradient of sexual differentiation in brain function

In the HCP sample, we regressed each FC estimate against sex (coded for 1 for male, −1 for female), age in years, income, years of education, employment status, race (coded 0 for White, 1 for non-White), handedness (coded 0 for predominantly right-handed, 1 for not predominantly right-handed), psychopathology (i.e., scores on the DSM-oriented scales of depression, anxiety, ADHD, antisocial personality disorder, somatic disorder, avoidant personality disorder from the Adult Self-Report [ASR][32]), average framewise displacement during the resting state scans as a proxy for individual differences in head motion together with scores on fluid and crystallized/verbal intelligence (i.e., Progressive Matrices and Picture Vocabulary Test, respectively[123]). This analysis resulted in a t-statistic quantifying the degree to which individual differences in each FC estimate are associated with biological sex, such that positive values represent stronger FC in males and negative values represent stronger FC in females. The absolute value of the t-statistic quantifies the magnitude of sexual differentiation for each FC index and, as such, it implicitly accounts for individual differences in sexual differentiation in the young adult group. A low absolute t-value denotes an FC index that shows greater between-individual, rather than between-sex, variability, whereas a high absolute t-value denotes an FC index that is consistently higher (positive t-value) or lower (negative t-value) across most males relative to females. The resulting matrix of t-values thus represents a reference of the degree to which FC estimates are sexually differentiated in healthy young adults.

We then computed, for each adolescent in the ABCD and HCP-D cohorts, the Spearman correlation between each row of their own FC

matrix (i.e., each regional FC profile) and the corresponding row of the template sexual differentiation t-statistic matrix estimated in the HCP sample. A positive correlation coefficient indicates that the regional FC profile identified in the ABCD or HCP-D participant resembles a more masculine pattern, as identified in HCP individuals, whereas a negative coefficient indicates a more feminine profile.

### Sensorimotor-Association (S-A) axis

To shed light on whether patterns of sexual differentiation in brain function align with the unimodal-to-transmodal hierarchy identified in prior research[63], we downloaded the corresponding gradient map characterized in[63] from neuromaps[124]. Using the "cifti-parcellate" (MEAN function) from the Connectome Workbench, we extracted ROI-specific gradient coefficients for the Schaefer and Gordon atlases. Subsequently, for each of the two functional atlases, we conducted a Spearman's rank correlation between the cross-validated ROI loadings extracted with PLS from the ABCD sample in Analysis 2 (for details, see Fig. 2-Analysis 2 and the Analysis 2 section in the Results) and the S-A axis ranks.

To account for correlated sexual differentiation and S-A gradient coefficients based on anatomical proximity[125] we used Vasa's "rotate_parcellation" function in Matlab (https://github.com/frantisekvasa/rotate_parcellation/commit/bb8b0ef10980f162793cc180cef371e83655c505) in order to generate 100,000 spatially constrained permutations of the Schaefer and Gordon brain latent variables (LV), as identified in the behavioral PLS analyses for the ABCD sample. These spatially constrained permuted functional brain LVs were used to assess the significance of the correlation between the Schaefer or Gordon brain LV and the S-A axis coefficients.

### Psychiatric disorder-related maps of sexual differentiation in brain function in HCP-D

To characterize patterns of neural sexual differentiation tracking variations in psychiatric disorder symptoms and physiological maturation in HCP-D, we computed partial Spearman's correlation coefficients between the ROI-specific sexual differentiation index described above and three sets of measures: (1) scores on reproductive maturation (i.e., youth ratings of adrenarche and gonadarche, respectively); (2) markers of PhenoAge, as identified in the ABCD (i.e., BMI, financial deprivation); and (3) each of the nine DSM-oriented CBCL scales (identical to the ABCD sample). These partial correlations controlled for chronological age, handedness, race, sex, testing site and average modality-specific scan-to-scan displacement (see "Control variables" below). These analyses resulted in 13 different brain maps (i.e., 9 related to CBCL scales, 2 related to aging and 2 related to pubertal development [i.e., adrenarche and gonadarche]). In these maps, positive and negative coefficients respectively indicate that either a more masculine or feminine FC pattern within a given ROI is related to (1) more advanced reproductive maturation; (2) higher BMI or exposure to poverty; and/ or (3) higher CBCL-related symptom severity.

### Polygenic Risk Scores (PRSs) for Psychiatric Disorders Linked to Sexual Differentiation in Brain Function

To shed light on whether patterns of sexual differentiation in brain function mediate the link between genetic risk and psychopathology, we used summary statistics from large disorder-focused GWASs, featuring case-control comparisons, in order to compute polygenic risk scores (PRS) for three neuropsychiatric conditions assessed in the ABCD sample via parental responses on the CBCL, specifically, MDD[49], anxiety disorders[50], and ADHD[51]. We did not include PRSs for ODD and CD because we could not locate relevant case-control GWASs. Similar to the ABCD sample herein used, the GWAS samples contributing to the PRSs were predominantly, but not exclusively, White. However, race was used as a covariate in all our analyses (see "Control variables" below).

The disorder-specific PRSs were each computed as the weighted sum of risk alleles based on the summary statistics of large GWASs focused on each disorder, which had been made available by the original authors via the "Public Results" tab on the FUMA website (https://fuma.ctglab.nl/browse[126], MDD, ADHD) or on the Psychiatric Genetics Consortium webpage (https://pgc.unc.edu/for-researchers/download-results/) (anxiety). For MDD, we used the top 10k most informative variants, based on approximately 76k patients and 230k, which have been made publicly available by ref. 49. These variants were obtained by clumping the corresponding GWAS statistics with the following parameters p1 = p2 = 1, window size <500 kb, and $r^2 > 0.1$.

To compute PRSs for each disorder based on the *.genotype ABCD data, we used the PLINK genetic analysis toolset[127] with SNPs significant at GWAS level $p \leq 10^{-5}$ since this was the lowest $p$-value for which all scrutinised disorders had contributing SNPs. However, in supplemental analyses, we confirmed that all the effects hold when using more lenient $p$-thresholds, including the MHC region (see Figs. S5–8).

Prior to PRS computation, the following preprocessing steps were implemented: (1) genes with a minor allele frequency (MAF) < 0.05, insertion/deletion and ambiguous single nucleotide polymorphisms (SNPs) (i.e., A/T and G/C pairs) were excluded; (2) highly correlated SNPs ($r^2 > 0.10$) within a 500 kb window were eliminated. In line with extant PRS-related practices and related investigations (e.g.[128]), the major histocompability complex (MHC) region has been excluded (but in Supplemental Analyses we verified that its inclusion does not impact the reported effects). The SNPs which survived the preprocessing and had an associated GWAS level $p \leq 10^{-5}$ contributed to the computation of the disorder-specific PRSs (MDD PRS: $N = 7$ SNPs; ANX PRS: $N = 3$ SNPs; ADHD PRS: $N = 12$ SNPs).

## Statistical analysis

We examined associations among physiological age, pubertal development, psychopathology, and sexual differentiation in brain function by combining canonical correlation analysis (CCA) with partial least squares (PLS), two multivariate techniques that explain commonalities in two sets of variables by creating linear combinations from the variables in each set[41,129]. All CCA and PLS models were estimated using a 10-fold cross-validation procedure. 99% confidence intervals (CI) for each correlation coefficient (CCA) and loading/salience (PLS) were obtained via the 'bootci' function in Matlab (with default settings and 100,000 bootstrap samples).

## Canonical correlation analysis (CCA)

To probe the relationship of psychopathology with sex, physiological age and pubertal development in the ABCD (see Fig. 2, Analysis 1), as well as the relevance of sexual differentiation in brain function, as estimated in the ABCD sample to psychopathology and factors relevant to physiological aging in the HCP-D (see Fig. 2, Analyses 4 and 5), we conducted a series of CCAs with cross-validation (cf.[41]). CCA was selected for these analyses because of its greater sensitivity to cross-set relationships relative to PLS[130]. CCA was implemented in Matlab using the 'canoncorr' module. A 10-fold cross-validation procedure tested the performance of our CCA-derived models. Discovery analyses were conducted on nine folds of data and the resulting CCA weights were employed to derive predicted values of the relevant variates in the left-out (test) fold. This procedure was repeated until each of the ten folds served as test data once. The ten test folds were concatenated and the correlation between the predicted variates across the full sample was evaluated using a permutation test with 100,000 samples.

Relationships between measured variables and their corresponding CCA-derived variate are described via correlations between the observed value of a given variable and the predicted value of its corresponding variate. We did not include standardized coefficients since the unique association between a given variable and its corresponding variate was of limited value in the case of our present analyses (with the exception of Analysis 1, as noted in the main text). 99% confidence intervals (CI) for each correlation coefficient were obtained by using the 'bootci' function in Matlab (with default settings and 100,000 bootstrap samples). The aforementioned correlation coefficients are a more conservative estimate of the traditional canonical loadings[41] as they are estimated in the test, rather than discovery, folds.

## Partial least squares (PLS) analysis

In the ABCD, patterns of sexual differentiation in brain function linked to psychopathology, physiological age and pubertal development were identified in a data-driven manner with PLS, which was selected for these analyses due to its better performance in datasets containing highly correlated within-set variables, for which CCA yields less reliable and reproducible solutions[130,131]. PLS was implemented using a series of Matlab scripts, which are available for download at https://github.com/McIntosh-Lab/PLS/. In this analysis, the behavioral set included the psychopathology and physiology variates extracted in Analysis 1. The brain matrix contained the ROI-specific correlation coefficients with the sexual differentiation gradient characterized in the HCP sample.

**Significance and generalizability testing.** The significance of the extracted brain-behavioral LV pair was estimated through permutation testing (i.e., 100000 permutations, 200 times larger than the guidelines provided by ref. 132) in the discovery PLS analysis conducted in the ABCD sample. The generalisability of our PLS models was tested using a 10-fold cross-validation procedure, similar to the one implemented for the CCAs. Specifically, the PLS analysis was run on nine folds of data and the weights associated with the identified brain LVs were used to compute the predicted value of the brain and behavioral LVs in the test fold. This procedure was repeated until each of the ten folds served as test data once. The significance of the extracted brain-behavioral LV correlation was estimated in the test fold using a procedure similar to the one described in ref. 133. Specifically, in the test folds,

(1) we multiplied each ROI-specific index of sexual differentiation in each condition (i.e., T1 vs T2, Fig. 4A, B) by its corresponding weight (as extracted in the training folds);

(2) we multiplied the physiology/psychopathology scores by their corresponding condition-wise weights (as extracted in the training folds);

(3) we applied singular value decomposition (SVD) to the condition-wise correlation between the ROI and physiology/psychopathology matrix, computed as described at (1) and (2) (cf[129]);

(4) we generated 100,000 null distributions by within-condition permutation of the physiology/psychopathology scores (i.e., we used the same permutation order in both condition 1 and condition 2) which preserved the interdependence of the T1/T2 brain scores (e.g., in the permuted sample, participant 10's physiology/psychopathology scores were paired with participant 89's T1 and T2 brain data) and applied SVD to each pair of brain- physiology/psychopathology matrices.

**Reliability testing.** In the discovery PLS analyses, we used 100,000 bootstrap samples (i.e., 1000 times larger than the guidelines provided by ref. 132) to (a) identify ROIs making a reliable contribution to the extracted LV (i.e., bootstrap ratios [BSRs], ROI weight/standard deviation, greater than 2.75 in absolute value, similar to a 99% CI), and (b) estimate 95% confidence intervals (CIs) for the correlation between the behavioral variables and the extracted brain LV[132]. In the cross-validation procedure, we used the "bootci" function in MATLAB (with default settings and 100,000 bootstraps) to characterize the (a) 99% CIs of the correlation coefficient corresponding to each ROI and the predicted brain LV (estimated in the test fold); (b) 95% CIs for the correlations between the predicted value of the brain LV and each variable in the behavioral set.

## Serial mediation model

To probe the temporal sequence in which patterns of sexual differentiation in brain function together with physiological age and pubertal development mediate the link between genetic risk and psychopathology in the ABCD sample, we conducted one serial mediation analysis using model 81 in Hayes' PROCESS 4.2 macro for SPSS[47]. PROCESS is an ordinary least squares (OLS) and logistic regression path analysis modeling tool based on observable variables. Serial mediation is used to test the viability of the causal chain (mediator 1-to-mediator 2…-to mediator *n*) through which an independent variable impacts the outcome variable. The specificity of our mediation sequence of interest was established by testing for alternate paths involving parallel or serial mediation (see Fig. 2, Analysis 3). The mediation model was tested with 95% CIs estimated with 50000 bootstrapping samples. In line with extant guidelines on balancing Type I and Type II errors in mediation analyses[134], the CIs for indirect effects was estimated using percentile bootstrap, which is the default option in PROCESS 4.2. As recommended by ref. 135, a heteroscedasticity-consistent standard error and covariance matrix estimator were used. Bootstrapping-based 95% CIs for the indirect effects (cf.[136]), as outputted by PROCESS, were used as effect size estimates.

## Control variables

In order to maximize the interpretability of the model estimates, the discovery CCA and PLS analyses were conducted on data that had not been residualized for the confounders listed below. To demonstrate the robustness of our results, the CCA and PLS cross-validation, as well as the mediation analyses controlled though for the following variables which were of no interest in the present study: (1) chronological age (for ABCD, at each wave contributing to the respective analysis) to ensure the reported inter-relationships hold irrespective of the participants' chronological age; (2) handedness (coded as "0" for right-handedness and "1" for non-right-handedness) to control for potential differences in the lateralization of the observed effects, e.g.[137,138]; (3) scanner site to account for scanner-related differences, as well as broad differences in family education and socio-economic status across site[139]; (4) race because the genetic architecture of some risk loci may show some racial variations[140–142] and to ensure that any associations with psychopathology hold irrespective of potential experiences of discrimination, (5) adoption status (dummy coded "0"/"1" for the 3 ABCD adoptees and the 5 HCP-D adoptees), (6) (ABCD sample only) sex assigned at birth not corroborated by menstruation history at ages 11-12 or SNP analysis (dummy-coded 0"/"1" for 2 youths) and (7) average framewise displacement (FD) per participant as a proxy of individual differences in head motion (cf[46,143]) which can adversely impact functional connectivity metrics[46]. For the ABCD participants, average FD was computed in reference to the 3 runs contributing to the analyses at each time point (6 runs in total across Time 1 and Time 2). Thus, via linear regression, we removed average framewise displacement from all neural outcomes of interest (i.e., ROI-to-ROI indices of sexual differentiation and the sexual differentiation latent variable [Analyses 2-5]) to address any residual motion-related confounds not addressed by our pre-processing pipeline[119]. As an additional control, we re-ran the discovery PLS analyses after residualizing the behavioral set for the same confounds entered in the cross-validation analyses (since motion may interact with these other confounds to bias the ROI-to-ROI indices of sexual differentiation in FC). We only residualized the behavioral set because residualization of both variable sets for multivariate analyses, such as PLS, which use permutation-based significance testing, has been found to bias results[144]. These additional analyses replicated those reported in the main text. The unthresholded brain LV from the discovery PLS analysis described in the main text diverged only slightly from the one extracted from the residualized data (*r*s of 0.93

and 0.92 with the Schaefer and Gordon atlas, respectively, Fig. S4A, D). The brain LV identified in these additional analyses (Fig. S3, 4) was robustly correlated, based on 95% CIs, with the physiology variate at Time 1 (*r*s from to 0.27 to 0.49 across both atlases) and the psychopathology variate at both time points (*r*s from to 0.18 to 0.62 across both atlases). The relevant brain maps are presented in Fig. S3, 4.

As an additional control, we re-ran the PLS analysis in a subsample of 89 participants passing the stringent motion elimination criteria outlined in reference[120]. Specifically, these criteria were: average participant FD <0.20 mm, 80% of all collected volumes having an FD <0.20 mm, and maximum motion under 5 mm. Reference[120] found that more aggressive motion controls (e.g., censoring) had no additional benefit beyond the exclusion of such high motion participants. We therefore favored this approach to avoid analysizing data with different numbers of timepoints in each participant, which can arise if some form of censoring/scrubbing is applied. Similar to the full sample of 199 youths, these 89 low-motion youths showed a distribution of QC-FC correlations and associated medians that matched the metrics of the best performing denoising pipelines depicted in reference[119] (see Fig. S10). However, the low motion sample showed QC-FC/distance correlations lower than those observed in the full sample of 199 youths. Of note, the QC-FC/distance correlations in the low motion sample matched the top 3 or top 6 (depending on the atlas[119]) best performing pipelines out of the 14 pipelines tested in reference[119]. The PLS brain LV identified in the full sample of 199 youths was replicated in the low-motion sample (see Fig. S3, 4). In sum, the brain LV identified with the discovery PLS analysis (Fig. 4C) was replicated in the low-motion sample and after residualizing for confounds, including motion (Fig. S3, 4). All the other reported analyses that featured the sexual differentiation indices controlled for average participant FD. Given all these controls, we think it is unlikely that our reported results are substantially contaminated by motion artifacts.

## Reporting summary

Further information on research design is available in the Nature Portfolio Reporting Summary linked to this article.

## Data availability

The raw data used in this report is available to researchers working at institutions recognized by NIMH at https://nda.nih.gov/abcd (ABCD) and at https://nda.nih.gov/ccf/lifespan-studies (HCP-D) following completion of the relevant data-use agreements. Researchers with different affiliations need to complete separate data-use agreements, signed by an authorized signing official from their respective institutions prior to submission to the NDA. Researchers need to create an account on https://nda.nih.gov/. Access is typically granted within a month from submitting the data-access request via the NDA site. Access to these data is controlled due to the highly sensitive nature of the data. Researchers need to apply for data access renewal on a yearly basis. Renewal requests need to include a progress report and are reviewed by the data access certification team. The ABCD data repository grows and changes over time. The ABCD data used in this report came from Data Release 5.1 (https://doi.org/10.15154/z563-zd24). The HCP-D data used in this report came from Data Release 2.0 (https://doi.org/10.15154/1520708). With regard to brain atlases, the Schaefer atlas can be downloaded from https://github.com/ThomasYeoLab/CBIG and the Gordon atlas can be downloaded from https://wustl.app.box.com/v/parcels-release. The specific dataset (ABCD, HCP-D) used in this report can be accessed via the NDA site at https://doi.org/10.15154/te4p-qr97.

## Code availability

We used already existing code, as specified in the main text with links for free download.

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

## Acknowledgements

Data used in the preparation of this article were obtained from the Human Connectome Project, WU-Minn Consortium (Principal Investigators: David Van Essen and Kamil Ugurbil; 1U54MH091657; funders: the 16 NIH Institutes and Centers that support the NIH Blueprint for Neuroscience Research and the McDonnell Center for Systems Neuroscience at Washington University), as well as from the Adolescent Brain Cognitive Development (ABCD) Study (https://abcdstudy.org) and the Human Connectome Project-Development (HCP-D) Study (https://humanconnectome.org/study/hcp-lifespan-development), both of which are held in the NIMH Data Archive (NDA). The ABCD study is a multisite, longitudinal study designed to recruit more than 10,000 children ages 9-10 and follow them over 10 years into early adulthood. The ABCD Study is supported by the National Institutes of Health and additional federal partners under award numbers U01DA041048, U01DA050989, U01DA051016, U01DA041022, U01DA051018, U01DA 051037, U01DA050987, U01DA041174, U01DA041106, U01DA041117, U01DA041028, U01DA041134, U01DA050988, U01DA051039, U01DA 041156, U01DA041025, U01DA041120, U01DA051038, U01DA041148, U01DA041093, U01DA041089, U24DA041123, U24DA041147. A full list of supporters is available at https://abcdstudy.org/federal-partners. html. A listing of participating sites and a complete listing of the study investigators can be found at https://abcdstudy.org/consortium_ members/. ABCD consortium investigators designed and implemented the study and/or provided data but did not necessarily participate in analysis or writing of this report. The HCP-D study is part of the HCP-Lifespan research project, which is is supported by grants U01MH109589, U01MH109589-S1, U01AG052564, and U01AG052564-S1 and by the 14 NIH Institutes and Centers that support the NIH Blueprint for Neuroscience Research, by the McDonnell Center for Systems Neuroscience at Washington University, by the Office of the Provost at Washington University, by the University of Minnesota Medical School, by the University of Massachusetts Medical School, and by the University of California Los Angeles Medical School. AF was funded by the National Health and Medical Research Council (ID: 1197431) and Australian Research Council (ID: FL220100184). This manuscript reflects the views of the authors and may not reflect the opinions or views of the NIH, HCP, or ABCD consortium investigators.

## Author contributions

R.P. conceptualized the research aims and designed the analyses with input from A.F. R.P. performed the analyses, created the figures, and wrote the original draft. C.M., E.B., and C.A.H. contributed to the interpretation of the results. All the authors contributed to the editing/revision of the original draft.

## Competing interests

CAH reports Research funding from American Beverage Association (paid to institution); Member of Mars Global Nutrition Advisory Council (honoraria paid to institution); Primary supervisor on PhD studentship funded by Coca-Cola; Personal honoraria from International Sweeteners Association and International Food Information Council for invited talks, All for work outside of the present report. R.P., A.F., C.M., and E.B. declare no competing interests.
