## [Transparent Peer Review file · Nature Communications]

Genetic Risk Predicts Adolescent Mood Pathology via Sexual Differentiation of Brain Function and Physiological Aging

Corresponding Author: Dr Raluca Petrican

Version 0:

Reviewer comments:

Reviewer #1

(Remarks to the Author)

In this manuscript, Petrican and colleagues leveraged longitudinal data from the ABCD Study and cross-sectional data from the HCP-D dataset to examine how sexual differentiation of the brain is linked to physiology and psychiatric vulnerability. Major strengths of this study include the use of two large datasets, a robust analytical pipeline, and examination of physiological, psychopathological, neurobiological, and genetic factors. Overall, this study is likely to be of broad interest to neuroimaging and psychiatry researchers. However, I do have a few questions/concerns about the manuscript. Please find them below.

I am signing my review and am happy to be contacted by the authors if they have questions.
- Elvisha Dhamala

Major comments:

1. How closely related are the T1 and T2 regional sexual differentiation patterns within subjects? If these are highly correlated, could it be possible that Analysis 2 is also showing that T2 sexual differentiation is linked to T2 physiology? Relatedly, in analysis 2 (and analysis 1), the psychopathology and physiology variates were only computed and considered at T2. If these are highly correlated with psychopathology and physiology variates at T1, could it be possible that Analysis 2 is showing the sexual differentiation at each time point is linked to physiology and psychopathology at that same time point, rather than predicting it in the future? I am not sure if these findings support the conclusions drawn without controlling for the T1 psychopathology and physiology variates.
2. How might gender influence the relationships studied here? Both ABCD and HCP-D datasets were collected in the US and are likely subject to the same set of sociocultural gender norms that the youth are exposed to throughout the course of development. Is it possible that these environmental influences of gender are driving sexual differentiation and/or specific patterns of physiology or psychopathology? A brief discussion of how gender may play a role would be helpful for interpreting these findings. Relatedly, sex is not binary, although it is treated as such here. Although this is not necessarily an issue, it would be helpful to include a discussion of what this assumption means. This is particularly important as sex and gender minority individuals are at a heightened risk for psychopathology.
3. What is the reasoning for using a young adult dataset (HCP) as the reference dataset to compare adolescents to (ABCD, ages 11-12) with regards to sexual differentiation? Seeing as the brain continues to mature throughout adolescence, would it make more sense to use adolescent data to derive a reference? Please provide a justification for this and discuss the potential implications of this choice.

Minor comments:

1. Why were only 199 adolescents included in these analyses from the ABCD study (out of 11,000+ participants in total)?
2. Line 378: Authors mention that sex was controlled for in their partial correlation analyses. Given that they're specifically looking to identify whether sexual differentiation is related to different phenotypes, I would assume that they wouldn't want to control for sex. Please provide a justification for this choice.
3. Line 74: It's a little unclear what the term 'biological females' is intended to mean here. If referring to individuals assigned

female at birth, please use that term instead.

4. The organization of the panels in Figure 1 makes it a bit difficult to follow. Consider separating these into 2 figures or re-organizing it so that each of the panels within the top and bottom sections follow a logical sequence (i.e., A → B → C → D, and 1 → 2 → 3 → 4 → 5).

5. What value exactly is being shown in Table 1?

Reviewer #2

(Remarks to the Author)

Review of "Genetic Risk Predicts Adolescent Mood Pathology via Sexual Differentiation of Brain Function and Physiological Aging" (NCOMMS-24-74315)

Brief summary:

This study investigates complex relationships between sexually-differentiated brain organization, physiological markers of well-being, pubertal development, and psychopathology by using both cross-sectional (HCP-D) and longitudinal data (ABCD Study). The main findings link brain feminization, or the similarity of resting-state functional connectivity patterns to those observed in group averages of adult females, in adolescents to more rapid pubertal and physiological changes. Further, brain feminization and physiological aging show a joint mediation of the relationship between anxiety-related brain organization and anxiety-related psychopathology. The authors replicate these initial findings from the ABCD Study in the HCP-D dataset.

Assessment of the paper:

This study takes on the impressive task of linking many features (i.e., brain organization, physiological well-being, pubertal development, psychopathology) and holds the potential to inform complex relationships important for understanding and improving mental health in adolescents. However, there are several large concerns that need to be addressed before the impacts of this paper can be realized. In particular, one critical analysis appears to be circular, the methodological basis of brain feminization (a primary topic of the paper) is unclear, and the processing of several measures (resting state functional connectivity, DHEA) appears inadequate. In addition, there is a general lack of precise language and clarity of writing that makes it difficult to understand the analyses and findings.

Major comments:

1. One of the primary analyses (Analysis 3) in the paper appears circular. The authors state that they computed "polygenic risk scores (PRS) for three neuropsychiatric conditions assessed in the ABCD sample via parental responses on the CBCL, specifically, MDD, anxiety disorders, ... and ADHD". The authors then use the anxiety PRS to predict psychopathology (a CCA variate including CBCL scores on anxiety and related psychopathology domains such as somatic complaints and OCD) in a mediation analysis. It is not surprising then that a PRS calculated from CBCL anxiety scores would relate to CBCL anxiety as they both involve the same data (i.e., CBCL anxiety). Prior analyses (1 & 2) already establish relationships between the outcome (psychopathology CCA variate) and the mediators (T1 PLS-extracted brain LV & T2 physiology CCA variate). As such, it is unclear what knowledge this analysis provides and the paper heavily relies on these findings in the Discussion (also in the Title). Also of note, the use of the SCZ PRS as the predictor in a control mediation model is not a true control comparison as it was not related to a SCZ-inclusive CCA variant; yet this too would be circular.

2. The validity of the methods used to define brain feminization and masculinization are unclear. The template of masculine vs feminine brains was established by regressing FC matrices of young adults on sex. Then, adolescent brains are compared to the adult template. However, it is unclear why averages of adult brains by sex is a valid template by which adolescent brains can be compared. If adolescents exhibit brain organization along a spectrum, shouldn't adults, and all individuals, as well? It is not clear why we could assume adolescents to have an axis of masculine to feminine brain organization but adults would not. Additionally, this analytical approach and the findings are not contextualized with prior literature, so it is unclear if this method has been validated in some manner. Further, and importantly, was psychopathology assessed in the young adults? Could it be that a higher prevalence of anxiety and depression among female young adults would give rise to the link between brain feminization and anxiety?

3. Throughout the Results section there is a general lack of explanation of each construct, analysis approach, and findings, as well as the motivations/goals of each (in particular Analysis 1). Additionally, linkages between specific constructs and the measures used to probe them are generally lacking clarity. For example, in Analysis 2 it is not clear if "physiology" in the first sentence is referring to PhenoAge or DHEA or both. Additionally, it is unclear what is meant by "ROI-specific FC patterns"; "LV" is undefined; and "relating T1 and T2 regional sexual differentiation" could mean change between these timepoints or in each of these timepoints. In another example, the phrase "and the confounders listed under 'Control variables'" is not precise enough for the Results section. A more precise description of the inputs and outputs of the CCA, PLS, and PRS methods are needed to interpret the results. Additionally, it is unclear if Analysis 5 is using HCP-D (as in the section title) or ABCD (as in the text). In another instance, it is unclear if brain masculinization and feminization were related to both sexes or only males and females respectively, or if this differed between the analyses. These methodological details become more clear in the Methods (or Discussion) sections. However, since the Discussion and Methods come after the Results, the Results section should contain sufficient information to allow the reader to understand the analyses and findings.

4. Despite substantial evidence that head motion causes widespread functional connectivity alterations (Power et al., 2012; Satterthwaite et al., 2012, 2019; Van Dijk et al., 2012; Ciric et al., 2017), in particularly in youth, the paper is lacking a discussion of the removal of head motion artifacts. While the HCP-D data was processed with ICA-FIX, the ABCD Study processing contains no mention of motion scrubbing (or other method of removal). Please assess relationships between

head motion and functional connectivity estimates in both samples to report if head motion artifacts were sufficiently removed from the brain data. Additionally, if motion scrubbing was implemented in the ABCD Study, please describe the procedure used and list the amount of data remaining (mean, standard deviation, range) after motion scrubbing.

5. The Methods section lacks any details on how the DHEA information reported in the ABCD Study was processed to ensure accuracy. At minimum, other ABCD Study papers describing how the hormone samples were collected and initially processed should be cited (e.g., Uban et al. (2018) Biospecimens and the ABCD study: Rationale, methods of collection, measurement and early data) and the steps you took to remove problematic samples should be described (e.g., considerations of sample color, timing). Please also report the mean, standard deviation, and range of (a) DHEA samples for each sex and (b) parent and youth pubertal development scores in the ABCD Study. Additionally, the correlation between DHEA and pubertal development score in the HCP-D study is missing.

6. It is unclear how the serial mediation analysis uncovered the “temporal sequence” of relationships since only one model, with T1 brain LV then T2 physio CCA, was tested. Would T1 physio then T2 brain also mediate the relationship? What about concurrent measures as the serial mediators (e.g., T1 brain then T1 physio)?

Minor comments:

1. Why was such a small portion of subjects in the ABCD Study used when more than 700 subjects have resting state data from both the baseline year and follow-up-year-two? Please explain why ABCD subjects were excluded from this project.

2. Generally, the abstract lacks clarity necessary to understand the study from the abstract alone. For example, it is not explained how “sexual differentiation” and “physiological aging” are operationalized. Additionally, it is unclear from the abstract if the study focused on biological females or included both sexes. More precise wording with clear definition of terms should be employed.

3. The use of PhenoAge to capture physiological aging is mentioned in the Introduction, but an explanation of what information is captured by PhenoAge is lacking. Similarly PhenoAge is mentioned in the Results section, but is not defined so an understanding of the results is difficult without reading the Methods first.

4. In the Methods section, it is not stated if the PhenoAge algorithm trained in adults can be reasonably applied to youth. Additionally, it is not stated if the validity of the use of the PhenoAge algorithm in HCP-D data, which is lacking all blood chemistry measures, produces valid results. It seems that this could be explored in the ABCD data comparing PhenoAge estimates with and without the blood chemistry measures.

5. The use of DHEA to characterize pubertal development is underexplained. Please explain the stage/aspects of pubertal development captured by DHEA (e.g., Adrenarche) and if/how differences between sexes were treated.

6. There are a couple issues regarding relationships with the sensorimotor-association axis. First, the S-A axis is not described, including in terms of its relevance for development or the relevance for the findings in this paper (e.g., what does the effect relating to the S-A axis tell us about brain development). Second, when the S-A axis is mentioned in the Results the direction of relationship is not mentioned. For example, does “greater feminization along the S-A axis” mean that association regions are more feminine or that individuals with more feminine brains look more like adult females on both ends of the S-A axis?

7. In the ABCD Study, resting state scans are collected in 5-minute runs. The Methods section reports that 3 resting state runs were used in each year (6 runs total). How does this add up to 25 minutes of resting state in each year? It seems there could only be 15 minutes total resting state collected per year.

8. The Methods section is missing several data processing citations (e.g., fMRIPrep).

9. The Methods section states that polygenic risk scores were identified for MDD and ADHD in addition to Anxiety; but a discussion of the use of these in mediation analyses is lacking from the Results.

10. Generally the Methods section is lacking precision. It seems that topics (e.g., PLS) are brought up out of order or without prior context. For example, in the “Sensorimotor-Association (SM-A) axis” section, the sentence “Spearman’s rank correlation between the cross-validated ROI loadings extracted with PLS” mentions PLS but it is not clear what is in the PLS at this point.

Reviewer #3

(Remarks to the Author)

This paper sought to examine the mediating role of pubertal factors and physiological aging on the link between polygenic risk scores and psychopathology symptoms. One strength of the current study is the use of two separate samples to test and replicate the effects observed in the sequential mediation. In addition, examining sexual differentiation in a continuous fashion in the context of neural development is an innovative premise to tackling sex-dependent risk for psychopathology. However, there are crucial aspects of the conceptual motivation and methods used in the study that limit my enthusiasm for the potential impact the results may have.

First, the use of DHEA as a pubertal index is concerning in this context. There are other pubertal indices available in ABCD (i.e., Pubertal Development Scale (PDS), and more relevant hormones) that would be more suited to this question and would make the comparisons with HCP-D more equivalent. To the first point, DHEA is not traditionally linked to sex differences neurally or behaviorally per se. It is also more directly linked to adrenarche, not gonadarche. In other words, the authors claim to be interested in examining the HPG axis, which regulates gonadarche, while the HPA axis regulates the production of DHEA (in the adrenal gland) and adrenarcheal processes. In cases where there are sex differences in DHEA or as a function of adrenarche, they tend to be less extreme than those attributable to gonadal hormones (e.g., testosterone, estradiol). Therefore, if hormonal assays are of interest, it seems more suitable to examine testosterone and estradiol to address this question, both of which are available in ABCD. In terms of comparability across ABCD and HCP-D, it would be more straightforward to examine PDS since it is available in both samples. Relatedly, the authors mention using DHEA to “stage” participants. The gold standard for staging is through the use of Tanner staging. Hormone levels are relative to the individual and cannot necessarily speak to what pubertal stage individuals are in through the currently available methods. Using the PDS (which can be converted to Tanner stages; Shirtcliff et al., 2009) would be a more reasonable and conventional approach if pubertal stages are of interest.

Second, the details regarding how the authors arrived at their sample sizes is very elusive, particularly for ABCD, which would presumably have thousands of available data points for this particular analysis.

Finally, the general conceptualization presented in the introduction is not an accurate characterization of the literature. Females may not inherently have increased stress reactivity per se. There may be unique social stressors for females that intersect with neurobiology. Women and girls have historically been underrepresented in studies of acute stress physiology, so it feels like a stretch to make a blanket statement about this in the abstract and to some extent in the introduction. Moreover, the introduction does not provide clear rationale for why puberty could be a catalyst for sexual dimorphism in psychiatric risk. For example, the first two paragraphs are missing the points that a) puberty leads to sex differences in the brain and behavior, and b) the prevailing idea that neuroplasticity observed during adolescence is likely driven by hormonal and psychosocial processes occurring during puberty, which dynamically interact to create sex-dependent risk for psychopathology.

Taken together, these aspects of the manuscript made it challenging to fully evaluate each section and the manuscript as a whole, given that some of the variables and critical details are either unsound despite the availability of data, or missing altogether.

Reviewer #4

(Remarks to the Author)

This paper aims to test how sex differences in brain functioning, reproductive maturation, and physiological aging may contribute to the vulnerability to developing mood disorders. This is an important question that is hardly addressed in the literature. The authors included two large open-access datasets (ABCD and HCP) to conduct an extensive set of impressive analyses to tackle their research questions. The outcomes showed that greater physiological wear and tear and higher DHEA levels were associated with higher somatic disorders. And sex-specific associations were found for changes in internalizing and externalizing symptoms. They additionally showed that sexual differentiation of brain function is associated differently with physiological age, reproductive maturity, and internalizing and externalizing symptoms in males and females. They replicated their results in the HCP sample.

The authors mention that “...are likely to underpin the emerging differential susceptibility of males and females to mood disorders during this life stage.” Although mood disorders in part could be explained by these mechanisms, they also interact with other factors including social cultural gendered experiences and expectations. Although these are not studied in this paper, and do not have to be per se, it should at least be mentioned that they also play a role in the gendered imbalance in diagnosis and treatment. For example, gendered expectations differentially affect self-esteem in males and females, which in turn affects mood disorders; there is an underdiagnosis of externalising disorders in females leading to higher rates of comorbidity in e.g. mood disorders, and there is underdiagnosis of mood disorders in males, just to name a few examples. Gendered experiences are particularly important to take into account when addressing the stress system, as e.g. the chances of experiencing a certain type of trauma are not the same for males and females. Moreover, twin studies show that mood disorders are, to a larger extent than other disorders, influenced by experiences compared to genetics. Again, these gendered experiences do not have to be taken into account in the present study per se, but it at least should be discussed.

The authors refer to the paper of Zhang et al., 2021 stating that: “...the human brain is best described along a continuum of sexual differentiation, with brains interposed between the male-like and female-like poles of this continuum being the least susceptible to internalizing symptoms.” I think it is important to note that this is also a case of framing, and a nuanced discussion of such outlier analysis could be included: for example the outlier analysis are framed as a male-female continuum but not directly related to male and female traits and mechanisms. An alternative explanation is for example that males show greater variance in brain structure than females (Forde et al., 2020, Ritchie et al., 2018, Wierenga et al., 2018, 2022). And extreme ends of brain patterns (in both males and females) have been associated with greater risk on mental health problems. Next to the conceptualization there are other methodological problems with the prediction modelling in the

paper of Zhang see a discussion in for example Sanchis-Segura (2022).

There are many factors that influence neurodevelopment and moreover mental health, therefore the following sentence should be toned down: "Our investigation was guided by evidence that adolescent neurodevelopment and mental health are shaped by the interaction between the immune system and the hypothalamic-pituitary-gonadal (HPG) axis"

It would help to include the age ranges of the different datasets in addition to the time points in the introduction. Also, it should be justified how the older sample of HCP can be used for estimates in the younger ABCD sample.

The authors assume that the brain stems from two sex-typical profiles, which has been shown not to be the case. This is both a matter of algorithm choice as well as labeling which should be addressed and justified (Sanchis-Segura (2019, 2021).

On a related note, it has been highly debated if the brain is "sexually dimorphic", which states that the brain is developing in two distinct male and female forms. Using this concept should be discussed and justified, as the effects of differential biological sex effects (the Y-chromosome, hormones, X vs XX chromosomes) are not directly tested in the present paper and therefore no conclusions can be drawn on whether these are associated with the results.

The analysis included both PDS as well as BMI as predictor variables, is BMI not also influenced by PDS? How would this affect the interpretation of the results?

The authors state that physiological age is highly correlated with environmental factors (e.g. poverty, nutrition). How would that affect the interpretation of the results.

In general, the results are hard to read and complex to follow. This maybe due to the word limit, but it would help to clarify at some points. For example, LV is not explained at all before the results section. Or the measure of 'feminization along the S-A axis' is rather unclear.

The results of analysis 1 are confusing, as it seems like the directionality of the association between levels of physiological wear-and-tear and DHEA and somatic scores are the same in males and females. To clarify, the different findings on cross-sectional and change data could be described separately.

Version 1:

Reviewer comments:

Reviewer #1

(Remarks to the Author)

Authors have addressed most of my concerns. Please find below a few remaining/additional comments.

1. The authors state that they examined the "relative patterns of masculinization/feminization within the same brain, not whether a brain, as a whole, is more feminine or masculine". This, along with the idea of sexual dimorphism (discussed below), is a highly debated topic and has been extensively studied in different neurobiological measures (e.g., brain structure, function, connectivity). The discussion could be enhanced by considering how these findings relate to existing literature and perspectives on this topic (e.g., Ingahlhalikar et al., PNAS 2013, along with the commentary and letter in response to the paper [Cahill et al., PNAS 2014; Joel et al., PNAS 2014], and the response to the letter; Joel et al., PNAS 2015, along with the letters in response to the paper [Guidice et al., PNAS 2016; Glezerman et al., PNAS 2016; Rosenblatt et al., 2016; Chekroud et al., 2016], and the replies to those letters; Satterthwaite et al., Cerebral Cortex 2014; Sanchis-Segura et al., BOSD 2019; Sanchis-Segura et al., NeuroImage 2022; among many others). Overall, the manuscript could benefit from a broader engagement with the existing literature.

2. Authors repeatedly use the term sexually dimorphic throughout the manuscript to refer to sex differences. As an example, the first sentence introduces the idea of 'sexually dimorphic mental health trajectories'. However, mental health trajectories are not sexually dimorphic. 'Sexual dimorphism' implies that males exhibit distinct traits compared to females. That is not the case for mental health traits. Males may be more likely to exhibit certain traits than females, and vice versa, but both males and females can exhibit the same mental health traits. Please revise the terminology used (see McCarthy et al., 2012 J Neurosci for a detailed discussion on appropriate usage). Please also consider whether the term sexually dimorphic is appropriate in other instances throughout the manuscript.

3. I recommend avoiding the term 'biological females' in the first sentence of the abstract. In general, internalising disorders are more prevalent in females and in women (regardless of their assigned sex), but it has not yet been established whether this is a sex effect or a gender effect. Therefore, the term 'females' or 'women' (or both) is more appropriate here.

4. Line 159: please describe what "PROCESS 4.2" refers to.

5. Line 187: section title suggests authors are referring to global “feminization” of the brain - consider using “regional FC feminization” instead.

Reviewer #2

(Remarks to the Author)

I appreciate the authors' thoughtful and thorough consideration of my feedback. I find most of their changes to be satisfactory. However, on one particular front I think there is additional work needed.

1. I previously suggested “Despite substantial evidence that head motion causes widespread functional connectivity alterations (Power et al., 2012; Satterthwaite et al., 2012, 2019; Van Dijk et al., 2012; Ciric et al., 2017), in particularly in youth, the paper is lacking a discussion of the removal of head motion artifacts. While the HCP-D data was processed with ICA-FIX, the ABCD Study processing contains no mention of motion scrubbing (or other method of removal). Please assess relationships between

head motion and functional connectivity estimates in both samples to report if head motion artifacts were sufficiently removed from the brain data. Additionally, if motion scrubbing was implemented in the ABCD Study, please describe the procedure used and list the amount of data remaining (mean, standard deviation, range) after motion scrubbing.” The authors replied with a description of the motion correction techniques that were implemented (e.g., nuisance regression with estimates of head motion and respiratory notch filtering, inclusion of average FD as a covariate in models). The authors also stated that “These motion correction steps are in line with those recommended by the ABCD study team (cf. Hagler et al., 2019).” However, the Hagler et al. (2019) paper does discuss motion censoring (AKA scrubbing), or the removal of high motion frames from the timeseries prior to the construction of correlation matrices. Here is the corresponding excerpt from Hagler et al. (2019):

“Motion censoring: Motion censoring is used to reduce residual effects of head motion that may survive the regression (Power et al., 2012, 2014). Time points with FD greater than 0.2 mm are excluded from the variance and correlation calculations. Note that this is a slightly more conservative threshold than that used for the regression step. Time periods with fewer than five contiguous, sub-threshold time points are also excluded. The effects of head motion can potentially linger for several seconds after an abrupt head motion, for example due to spin-history or T1 relaxation effects (Friston et al., 1996), so an additional round of censoring is applied based on detecting time points that are outliers with respect to spatial variation across the brain. SD across ROIs is calculated for each time point, and outlier time points, defined as having an SD value more than three times the median absolute deviation (MAD) below or above the median SD value, are excluded from variance and correlation calculations.”

The authors state that the ABCD resting-state data is of “good quality”, but this should be demonstrated as (1) it is unusual to not include motion censoring in the data analysis of ABCD resting-state data (e.g., Marek, Tervo-Clemmens, et al. 2022), (2) it deviates from the processing steps outlined in Hagler et al. (2019) and, thus, also the released resting state data on the NDA, and (3) the prior literature robustly demonstrates that the steps the authors took here are not sufficient to remove the impacts of head motion on the data and any resulting inferences (Power et al., 2012; Ciric et al., 2017).

If the authors would prefer to not implement motion censoring, they can demonstrate that the impacts of head motion have been sufficiently removed using the benchmarks outlined by Ciric et al. (2017) “Benchmarking of participant-level confound regression strategies for the control of motion artifact in studies of functional connectivity”. Specifically, demonstration of the (a) residual QC-FC correlations after de-noising and (b) distance-dependence of motion artifacts after de-noising. Comparing these results to those published by Ciric et al. (2017) will help to clarify if there are remaining motion impacts in the data. The scripts to create these benchmarks are made publicly available in <http://github.com/PennBBL/xcpEngine> and are further described in Ciric et al., (2018) “Mitigating head motion artefact in functional connectivity MRI.”

Reviewer #3

(Remarks to the Author)

I appreciate the authors responding to all of my initial comments. I think that aspects of the manuscript are improved, including the introduction and the addition of methodological/analytic details. I have several remaining concerns and comments for the authors to address.

I caution the authors against using the term “stages” when referring to adrenarche and gonadarche. They are comprised of stages (up to 5, if you are using the Tanner convention). It could be confusing to readers to refer to them as stages. It also makes it sound like they are contingent upon one another, which they are not. The authors could consider referring to them as processes or components.

What is meant by “standardized” in reference to the hormonal assays?

Please put the units for the hormone assays in Table 1. The ranges shown are very wide on all of the hormones – did the authors check for outliers?

It's unclear why the authors chose to partial out age at menarche for the girls. Typically variables accounting for more momentary or monthly fluctuations, like caffeine and days since last menstrual cycle that the authors use, are covaried out of salivary/serum hormone levels. It's true that girls who are post menarche are expected to have higher levels of the hormones

of interest here, but that seems like meaningful variance that the authors may not want to remove.

I am unsure why the authors are including testosterone in the adrenal hormones for girls. It's true that DHEA/s are precursors to testosterone (and estrogen), but testosterone is not typically examined as part of adrenarche. I worry that by including testosterone, the authors may be undermining the specificity of any potential adrenarcheal effects since testosterone is released only in small quantities from the adrenal glands, with much higher levels released as part of gonadarche.

In the results section for analysis 1, please describe how the change scores were computed.

The findings from analysis 2 showing that neural "masculinization" is related to lower phenoAge and delayed puberty and vice versa in girls seems like it could be attributable to the fact that girls, in general, go through puberty earlier than boys. Could it just be that this result is demonstrating a normative sex-independent developmental process and not necessarily sex differences per se, just the ~1 year lag in pubertal timing between boys and girls (which is already well-documented)?

Version 2:

Reviewer comments:

Reviewer #1

(Remarks to the Author)

Authors have addressed all of my concerns.

Reviewer #2

(Remarks to the Author)

I appreciate that the authors included the Ciric benchmarks evaluating the impact of head motion on functional connectivity measures. I agree with the authors that the QC-FC plots demonstrate metrics similar to the best performing models that Ciric et al. tested. However, the correlation values are left off the distance-dependent graphs. It appears that there is a stronger distance-dependent effect in the study sample compared to the low motion sample and it is unclear how the values compare to those reported by Ciric et al. Please add the correlation values to the distance-dependent graphs in Figure S10.

Reviewer #3

(Remarks to the Author)

I thank the authors for addressing all of my comments. I have no further recommendations.

Version 3:

Reviewer comments:

Reviewer #2

(Remarks to the Author)

I thank the authors for adding the correlation values into the distance-dependent graphs. I additionally agree with the authors that given the control analyses implemented their results are likely not substantially impacted by motion artifacts. I do want to contest the authors statement in the manuscript that the distance-dependent correlation matched the metrics of the best performing pipelines in Ciric et al. (2017): "The distribution of the QC-FC correlations and the scatterplots describing the distance-dependence of the motion artifacts are included in Figure S10. The observed metrics paralleled those of the best performing pipelines described in 119 (see also 120 495)." I caution the authors to temper their language in this sentence, as it is currently somewhat misleading. Looking at the distance-dependence numbers for the full sample now included in Supplement Figure 10 (from -.35 to -.48), this suggests the data used in the manuscript is not among the best performing pipelines tested by Ciric (from .003 and -.116) but is rather among the worst performing pipelines tested by Ciric. I would suggest the authors rephrase this sentence to better align with the data reported in Supplement Figure 10.

Reviewer #1 (Remarks to the Author):

In this manuscript, Petrican and colleagues leveraged longitudinal data from the ABCD Study and cross-sectional data from the HCP-D dataset to examine how sexual differentiation of the brain is linked to physiology and psychiatric vulnerability. Major strengths of this study include the use of two large datasets, a robust analytical pipeline, and examination of physiological, psychopathological, neurobiological, and genetic factors. Overall, this study is likely to be of broad interest to neuroimaging and psychiatry researchers. However, I do have a few questions/concerns about the manuscript. Please find them below.

I am signing my review and am happy to be contacted by the authors if they have questions.

- Elvisha Dhamala

Major comments:

1a. How closely related are the T1 and T2 regional sexual differentiation patterns within subjects? If these are highly correlated, could it be possible that Analysis 2 is also showing that T2 sexual differentiation is linked to T2 physiology?

There was a moderate correlation between the cross-validated T1 and T2 sexual differentiation latent variables from Analysis 2 (r s of .49 and .54 based on Schaefer and Gordon atlas data, respectively).

Analysis 2 tested the correlation of T1 and T2 sexual differentiation with T2 physiology and T2/ $\Delta_{T2,T3}$ Psychopathology. It was only the correlation between T1 sexual differentiation and T2 physiology that was successfully cross-validated (Figure 4-B). The correlation between T2 sexual differentiation and T2 physiology was not successfully cross-validated. In our revised manuscript (please see lines 138-146), we now draw attention to the cross-validated correlation between T2 Physiology and T1 (but not T2) sexual differentiation:

“The discovery PLS analysis uncovered a single brain-behavior latent variable (LV) pair ($p = 2 \times 10^{-5}$, shared variance of 65.35%) relating T1 and T2 regional sexual differentiation to T2 physiology and T2/ $\Delta_{T2,T3}$ psychopathology scores (Figure 4-A). Cross-validation tests, which controlled for participant motion (i.e., average frame-wise displacement, ⁴⁵), age, race, testing site, handedness, adoption status, ambiguous biological sex, suggested that the brain-behavior LV relationship is robust (permutation-based $p = .001$, shared variance of 65.30%) and mostly captures the link between neural sexual differentiation at ages 9-10 and physiology at ages 11-12 ($r_{cv} = .19$, 95% CI = [.05; .34]; see Figure 4-B). Of note, the association between concurrent neural

sexual differentiation and physiology at ages 11-12 failed to reach conventional reliability levels ($r_{cv} = .05$, 95% CI = [-.08; .20]; see Figure 4-B). Testifying to its specificity, the correlation between T1 neural sexual differentiation and T2 Physiology was left virtually unchanged after additionally controlling for the available T1 Physiology measures (i.e., pubertal hormones, self-reported adrenarche and gonadarche) ($r_{cv} = .18$, 95% CI = [.04; .34]).”.

1b. Relatedly, in analysis 2 (and analysis 1), the psychopathology and physiology variates were only computed and considered at T2. If these are highly correlated with psychopathology and physiology variates at T1, could it be possible that Analysis 2 is showing the sexual differentiation at each time point is linked to physiology and psychopathology at that same time point, rather than predicting it in the future? I am not sure if these findings support the conclusions drawn without controlling for the T1 psychopathology and physiology variates.

In Analysis 1, Physiology was indeed only considered at T2 due to availability of blood chemistry data necessary for estimating PhenoAge (as specified on lines 92-93). However, psychopathology (assessed via CBCL) was considered both at T2 (ages 11-12) and in terms of change from T2 (ages 11-12) to T3 (ages 12-13). Controlling for T1 Psychopathology and/or T2 Physiology in Analysis 1 would change the focus of our tests. By controlling for T1 Psychopathology, we would be asking whether Physiology at T2 predicts change in Psychopathology from T1 to T2 and/or change in Psychopathology from T2 to T3 (irrespective of T1 Psychopathology). By controlling for T1 Physiology, we would be asking whether change in Physiology from T1 to T2 is linked to T2 Psychopathology and/or change in Psychopathology from T2 to T3. These are not our questions though. Analysis 1 was intended to probe cross-sectional associations between T2 Physiology and T2 Psychopathology, as well as longitudinal relationships between T2 Physiology and change in Psychopathology from T2 to T3.

As for Analysis 2, the cross-validation results (please see Figure 4-B) suggest that the $T2/\Delta_{T2,T3}$ Psychopathology CCA variate from Analysis 1 is not robustly linked to either T1 or T2 sexual differentiation brain LV (hence further controls would yield limited insights).

Complementarily, as we outlined in response to comment 1.a, the T2 Physiology variate is not robustly linked to the sexual differentiation brain LV at the same time point, but is instead reliably correlated with the T1 sexual differentiation brain LV (please see Figure 4-B). Put differently, Physiology at ages 11-12 is not robustly related to neural sexual differentiation at the same time point, but is instead related to earlier neural sexual differentiation.

The link between Physiology at T2/ages 11-12 and neural sexual differentiation at T1/ages 9-10 may be (partly) explained by T1 Physiology. This is something we sought to address by controlling for the available T1 Physiology measures in the cross-validation of the T1 neural sexual differentiation-T2 Physiology correlation (please see our reply to comment 1.a) and in our revised Analysis 3

(please see schematic representation in Figure 2, Analysis 3). Regarding the latter, prior to running Analysis 3, all the contributing variables were residualized by the available T1 Physiology measures, specifically, pubertal hormone levels and self-reported adrenarche/gonadarche (blood biomarkers for PhenoAge were unavailable at T1). To test the specificity of our proposed mediation sequence (PRS-T1 neural sexual differentiation-T2 Physiology- T2/ $\Delta_{T2,T3}$ Psychopathology), we simultaneously controlled for alternate paths, specifically, serial mediation via T1 neural sexual differentiation-T2 neural sexual differentiation, as well as parallel mediation via T2 Physiology and T2 neural sexual differentiation. None of these alternate paths yielded robust indirect effects (please see Figures 5, S3 and S6). Hence, we interpret the results as providing broad support our proposed temporal sequence.

Similar to our argument regarding Analysis 1, controlling for T1 psychopathology in the mediation analysis would not be appropriate since it would imply focus on change in psychopathology from T2/ages 11-12 to T3/ages 12-13, irrespective of psychopathology at T1/ages 9-10. We see no reason to impose this constraint, as our proposed serial mediation model may well explain why adolescents who were higher on psychopathology at T1/ages 9-10 may show a declining mental health trajectory from T2/ages 11-12 to T3/ages 12-13. Accordingly, our Analyses 4 and 5 were intended to test and have demonstrated a concurrent relationship between the sexual differentiation brain LV from Analysis 2 (projected onto the HCP-D sample) and a concurrent profile of psychopathology similar to the one identified in Analysis 1.

The revised analysis 3 is described on

(lines 155-169) “Our first two analyses provided evidence that physiology at ages 11-12 years shows longitudinal associations with earlier (but not concurrent) neural sexual differentiation (Figure 4-B) and increases in psychopathology, rather than concurrent psychiatric symptoms (Figure 3-A). Building on these findings, we next examined whether sexual differentiation in brain function and physiology sequentially mediate the relationship between genetic risk and psychopathology in the ABCD sample. To this end, we specified the serial mediation model depicted in Figure 2 (Analysis 3), tested with PROCESS 4.2. In line with existing guidelines for probing causal chains⁴⁶, the serial mediators were sampled from different time points. Based on the results from our first two analyses, we were primarily interested in whether the T1 neural sexual differentiation LV from Analysis 2 (mediator 1) and the sex-dependent physiology CCA variate from Analysis 1 (mediator 2) would constitute a viable causal chain linking polygenic risk scores (PRS) for anxiety and, potentially, depression to rising internalizing symptoms (cf Figure 3-A). In testing this sequence, we simultaneously accounted for alternate models involving serial (T1 and T2 neural sexual differentiation from Analysis 2) and parallel (T2 Physiology from Analysis 1 and T2 neural sexual differentiation from Analysis 2) mediation, respectively (see Figure 2-Analysis 3). The absence of blood chemistry data at T1 prevented us from computing PhenoAge and, thus, estimating the alternate serial and parallel mediation models involving T1 Physiology. However, prior to mediation analysis, all variables were adjusted for T1 pubertal hormone levels and self-reported adrenarche/gonadarche based on the PDS (in addition to participant motion, age, race,

testing site, handedness, adoption status, ambiguous biological sex). Additionally, inclusion of the PRS-T2 Physiology-Psychopathology path allowed us to estimate indirect effects of physiology that are independent of T1/T2 neural sexual differentiation and therefore uniquely relate to T1 PhenoAge.”

(lines 176-185) “As expected, we observed a robust indirect effect of the anxiety PRS on the T2/ $\Delta_{T2,T3}$ psychopathology variate via the cross-validated T1 brain LV (mediator 1) and the cross-validated T2 physiology CCA variate (mediator 2) (total standardized indirect effect: .007, SE = .005, 95%CI = [.001; .019]) (Figure 5-A). No additional indirect effects were detected, which implies that any association between T1 PhenoAge and later psychopathology, independent of T1 pubertal development (for which we controlled), would most likely be explained by its overlap with T1 neural sexual differentiation. The mediational analysis anchored in the MDD PRS yielded no robust indirect effects, but instead yielded a reliable direct effect (Figure 5-B). This pattern of results implies that genetic risk for MDD predicts worsening depression and anxiety disorder-related symptoms from ages 11-12 to ages 12-13. However, this link between the MDD PRS and affective problems is not mediated by the neural sexual differentiation and physiology patterns herein identified. Finally, we confirmed the specificity of the sequence anchored in the anxiety PRS by re-running the serial mediation analysis with ADHD PRS as the predictor, which revealed no significant indirect effect (Figure S5).”

The link between Analysis 3 and Analysis 4 with regards to the concurrent link between neural sexual differentiation and psychopathology is discussed on lines 258-263 (added text in red font)

“The longitudinal ABCD results were replicated cross-sectionally in the HCP-D sample, where sexual differentiation along the S-A axis was observed among faster-developing females, with greater exposure to poverty, at risk for faster physiological aging (higher BMI) and showing more severe mood (particularly, anxiety) symptoms. Considering these findings in light of the Analysis 3 outputs, there emerges the possibility that neural sexual differentiation may constitute a marker of concurrent psychiatric vulnerability (cf. Analyses 4 and 5) which may also identify individuals most likely to show a trajectory of worsening depression and anxiety disorder-related symptoms. This line of inquiry is beyond the scope of the present analysis, but certainly warranting further study.”

2. How might gender influence the relationships studied here? Both ABCD and HCP-D datasets were collected in the US and are likely subject to the same set of sociocultural gender norms that the youth are exposed to throughout the course of development. Is it possible that these environmental influences of gender are driving sexual differentiation and/or specific patterns of physiology or psychopathology? A brief discussion of how gender may play a role would be helpful for interpreting these findings. Relatedly, sex is not binary, although it is treated as such here. Although this is not

necessarily an issue, it would be helpful to include a discussion of what this assumption means. This is particularly important as sex and gender minority individuals are at a heightened risk for psychopathology.

In our “Limitations and Future Directions” section we included a brief discussion of how gender may impact our present results (lines 284-289):

“Lastly, our three samples rated their sex in a binary manner and only one participant in each of the two adolescent cohorts indicated that their gender is different from the sex assigned at birth (no information on gender—as distinct from biological sex-- was collected in the HCP). Therefore, some of the sex differences herein reported may stem from gendered experiences, including those resulting from gender differences in societal expectations that may extend to potential biases in psychiatric diagnosis. Future studies featuring samples that allow for meaningful dissociations between sex and gender ⁹¹ and which probe the interactive impact of multiple social identities ⁹² would provide further insights into the effects herein documented.”

3. What is the reasoning for using a young adult dataset (HCP) as the reference dataset to compare adolescents to (ABCD, ages 11-12) with regards to sexual differentiation? Seeing as the brain continues to mature throughout adolescence, would it make more sense to use adolescent data to derive a reference? Please provide a justification for this and discuss the potential implications of this choice.

Sexual differentiation in brain and behavior have been shown to depend on sexual maturation processes (as specified on lines 29-30 of our revised manuscript). Hence, neural sexual differentiation was assumed to be maximal for individuals who had reached reproductive maturity. This is why the healthy young adult participants in the Human Connectome project were used as a representation of the most accurate reference for estimating sex differences in brain function. This justification has been added to lines 132-136 of our revised manuscript:

“Sexual differentiation in the ABCD sample was characterized in reference to the functional connectivity (FC) patterns observed among the young healthy adult participants (ages 22-30) in the Human Connectome Project (HCP). These individuals have reached reproductive maturity and are thus expected to show maximal differentiation of brain function related to their biological sex (as assigned at birth [self-reported] and confirmed through menstruation history^{12, 16, 17, 18, 41, 42}..”

Please note though that the ROI-to-ROI FC estimates based on the Schaefer and Gordon atlases were quantified separately within the ABCD, HCP-D and HCP samples (as specified on lines 136-137). It was the independently extracted ROI-to-ROI FC matrices

that were subsequently compared between the HCP and each of the two developmental samples (ABCD, HCP-D) to compute continuous indices of sexual differentiation in adolescent brain function.

Minor comments:

4. Why were only 199 adolescents were included in these analyses from the ABCD study (out of 11,000+ participants in total)?

The ABCD data release 5.1, which includes all the available data from the two-year follow-up assessment, contained only 199 ABCD study participants who were biologically unrelated to one another and had good quality data on all the variables included in our analyses. Since our analyses were closely inter-related, inclusion of different participant samples in each test would have made interpretation of the results much more difficult. In our revised manuscript, our participant selection steps for the ABCD and HCP-D samples are illustrated in a STROBE diagram (please see Figure 7). Please note that the HCP-D sample now contains 277 (rather than 308) participants because in our initial submission we failed to eliminate participants sharing the same family ID (and, thus, likely to be biologically related). Our results nonetheless hold after this exclusion.

5. Line 378: Authors mention that sex was controlled for in their partial correlation analyses. Given that they're specifically looking to identify whether sexual differentiation is related to different phenotypes, I would assume that they wouldn't want to control for sex. Please provide a justification for this choice.

We controlled for sex in the partial correlation analyses linking neural sexual differentiation to psychopathology, pubertal development, body mass index and financial deprivation in the HCP-D sample. Analysis 5 linked these partial correlation maps to the neural sexual differentiation latent variable identified in the ABCD sample through Analysis 2. As stated on lines 213-216 of our revised manuscript, the purpose of Analysis 5 was to test “**whether the ABCD profile of neural sexual differentiation could predict psychiatric risk not only in a sex-dependent manner (cf. Analysis 4), but also independently of the effects of sex. This analysis thus tests the utility of a continuous approach to conceptualizing sex differences in brain function by shedding light on whether individual variability in relative brain feminization (or masculinization) can predict the same psychiatric symptoms across sexes.**”

6. Line 74: It's a little unclear what the term 'biological females' is intended to mean here. If referring to individuals assigned female at birth, please use that term instead.

We kept the term “biological females” when discussing cross-species studies in which sex was established through genetic analyses and/or hormonal measures. For our analyses, we specified that we referred to sex assigned at birth (based on self- or parental reports corroborated through single-nucleotide polymorphism analyses and/or menstruation history) (please see lines 311-314, 317-318, 325-326).

7. The organization of the panels in Figure 1 makes it a bit difficult to follow. Consider separating these into 2 figures or re-organizing it so that each of the panels within the top and bottom sections follow a logical sequence (i.e., A → B → C → D, and 1 → 2 → 3 → 4 → 5).

In the revised manuscript, Figure 1 from the initial submission has been separated into two figures. The revised Figure 1 depicts our conceptual framework whereas the revised Figure 2 illustrates our analytic workflow.

8. What value exactly is being shown in Table 1?

This Table is now numbered Table 2 in the revised manuscript. It shows the means and standard deviation for each subscale of the Child Behavior Checklist (CBCL) in the analysed ABCD (N = 199 at ages 11-12 [Time 2] and 12-13 [Time 3]) and HCP-D (N = 277) samples. We apologize that, in the prior version of our manuscript, Time 2 and Time 3 were mis-labelled as Time 1 and Time 2. In addition to correcting this labelling error, we have included the relevant participant ages for each ABCD wave.

Reviewer #2 (Remarks to the Author):

Review of “Genetic Risk Predicts Adolescent Mood Pathology via Sexual Differentiation of Brain Function and Physiological Aging” (NCOMMS-24-74315)

Brief summary:

This study investigates complex relationships between sexually-differentiated brain organization, physiological markers of well-being, pubertal development, and psychopathology by using both cross-sectional (HCP-D) and longitudinal data (ABCD Study). The main findings link brain feminization, or the similarity of resting-state functional connectivity patterns to those observed in group averages of adult females, in adolescents to more rapid pubertal and physiological changes. Further, brain feminization and physiological aging show a joint mediation of the relationship between anxiety-related brain organization and anxiety-related psychopathology. The authors replicate these initial findings from the ABCD Study in the HCP-D dataset.

Assessment of the paper:

This study takes on the impressive task of linking many features (i.e., brain organization, physiological well-being, pubertal development, psychopathology) and holds the potential to inform complex relationships important for understanding and improving mental health in adolescents. However, there are several large concerns that need to be addressed before the impacts of this paper can be realized. In particular, one critical analysis appears to be circular, the methodological basis of brain feminization (a primary topic of the paper) is unclear, and the processing of several measures (resting state functional connectivity, DHEA) appears inadequate. In addition, there is a general lack of precise language and clarity of writing that makes it difficult to understand the analyses and findings.

Major comments:

1. One of the primary analyses (Analysis 3) in the paper appears circular. The authors state that they computed “polygenic risk scores (PRS) for three neuropsychiatric conditions assessed in the ABCD sample via parental responses on the CBCL, specifically, MDD, anxiety disorders, ... and ADHD”. The authors then use the anxiety PRS to predict psychopathology (a CCA variate including CBCL scores on anxiety and related psychopathology domains such as somatic complaints and OCD) in a mediation analysis. It is not surprising then that a PRS calculated from CBCL anxiety scores would relate to CBCL anxiety as they both involve the same data (i.e., CBCL anxiety). Prior analyses (1 & 2) already establish relationships between the outcome (psychopathology CCA variate) and the mediators (T1 PLS-extracted brain LV & T2 physiology CCA variate). As such, it is unclear what knowledge this analysis provides and the paper heavily relies on these findings in the Discussion (also in the Title). Also of note, the use of the SCZ PRS as the predictor in a control mediation model is not a true control comparison as it was not related to a SCZ-inclusive CCA variant; yet this too would be circular.

We apologize for the misunderstanding. The polygenic risk scores (PRSs) for anxiety, MDD and ADHD were computed based on summary statistics from large genome-wide association studies focused on each disorder and using case/patient-(healthy) control comparisons (all the Ns > 17,000 individuals for each disorder, with greater than 7,000 cases for each disorder). Hence, the weights for the risk alleles contributing to each PRS were estimated in completely independent samples (several orders larger than the ABCD subsample used in our analyses). We now clarify this in the Results section of the main text (please see lines 170-171):

“PRSs for anxiety, depression and ADHD were computed using summary statistics from large and independent disorder-focused genome-wide association studies (GWASs), featuring case/patient-control comparisons (i.e., MDD ⁴⁷, anxiety disorders ⁴⁸, ADHD ⁴⁹).”

A more detailed description of the PRS computation procedure is included in the Methods (please see lines 526-547).

In the prior version of our manuscript, we ran the serial mediation model with the SCZ PRS to test the specificity of serial mediation model linking the anxiety PRS to our identified T2/ Δ T_{2,T3} Psychopathology variate (on which changes in anxiety disorder scores loaded most robustly). We agree though with the reviewer that testing the mediation model using ADHD PRS would constitute a stronger specificity test since ADHD was included in Analysis 1, but was found to not load robustly on the extracted Psychopathology variate (Figure 3-A). We ran the mediation model anchored in the ADHD PRS and, as expected, found no evidence of robust indirect effects to the Psychopathology variate (please see Figure S5). These findings are now reported on lines 170-185 of the revised manuscript:

“PRSs for anxiety, depression and ADHD were computed using summary statistics from large and independent disorder-focused genome-wide association studies (GWASs), featuring case/patient-control comparisons (i.e., MDD⁴⁷, anxiety disorders⁴⁸, ADHD⁴⁹). We did not include PRSs for Oppositional Defiant or Conduct Disorder because we could not locate relevant case-control GWASs. Based on the results from Analyses 1 and 2, robust indirect effects via neural sexual differentiation and physiology were expected for mediational models anchored in the anxiety PRS and, potentially, the depression PRS, whereas models anchored in the ADHD PRS were included for specificity analysis since ADHD did not have a robust loading on the psychopathology variate in Analysis 1 (Figure 3-A).

As expected, we observed a robust indirect effect of the anxiety PRS on the T2/ Δ T_{2,T3} psychopathology variate via the cross-validated T1 brain LV (mediator 1) and the cross-validated T2 physiology CCA variate (mediator 2) (total standardized indirect effect: .007, SE = .005, 95%CI = [.001; .019]) (Figure 5-A). No additional indirect effects were detected, which implies that any association between T1 PhenoAge and later psychopathology, independent of T1 pubertal development (for which we controlled), would most likely be explained by its overlap with T1 neural sexual differentiation.

The mediational analysis anchored in the MDD PRS yielded no robust indirect effects, but instead yielded a reliable direct effect (Figure 5-B). This pattern of results implies that genetic risk for MDD predicts worsening depression and anxiety disorder-related symptoms from ages 11-12 to ages 12-13. However, this link between the MDD PRS and affective problems is not mediated by the neural sexual differentiation and physiology patterns herein identified. Finally, we confirmed the specificity of the sequence anchored in the anxiety PRS by re-running the serial mediation analysis with ADHD PRS as the predictor, which revealed no significant indirect effect (Figure S5). “

2a. The validity of the methods used to define brain feminization and masculinization are unclear. The template of masculine vs feminine brains was established by regressing FC matrices of young adults on sex. Then, adolescent brains are compared to the adult template. However, it is unclear why averages of adult brains by sex is a valid template by

which adolescent brains can be compared. If adolescents exhibit brain organization along a spectrum, shouldn't adults, and all individuals, as well? It is not clear why we could assume adolescents to have an axis of masculine to feminine brain organization but adults would not.

We agree with the reviewer that young adults also show a spectrum of neural sexual differentiation. However, the healthy young adults from the HCP were used as a reference group because sexual differentiation in brain and behavior have been shown to depend on sexual maturation processes. Hence, we assumed that neural sexual differentiation is maximal for individuals who had reached reproductive maturity, such as the HCP young adult participants. In our revised manuscript, the above information on reproductive maturation and neural sexual differentiation has been included on lines 29-30

“Pubertal hormones have been implicated in the development of sexually dimorphic patterns of behavior and neurophysiology¹³⁻¹⁵”

and lines 132-136

“Sexual differentiation in the ABCD sample was characterized in reference to the functional connectivity (FC) patterns observed among the young healthy adult participants (ages 22-30) in the Human Connectome Project (HCP). These individuals have reached reproductive maturity and are thus expected to show maximal differentiation of brain function related to their biological sex (as assigned at birth [self-reported] and confirmed through menstruation history^{12, 16, 17, 18, 41, 42})”.

As an aside, please note that the ROI-to-ROI FC estimates based on the Schaefer and Gordon atlases were quantified separately within the ABCD, HCP-D and HCP samples (as specified on lines 136-137). It was the independently extracted ROI-to-ROI FC matrices that were subsequently compared between the HCP and each of the two developmental samples (ABCD, HCP-D) to compute continuous indices of sexual differentiation in adolescent brain function.

Please also note that our method for estimating neural sexual differentiation does account for individual differences in FC within each sex. This is explained on lines 488-499 (the relevant text is in red font, whereas the remaining text is supplied for context):

“In the HCP sample, we regressed each FC estimate against sex (coded for 1 for male, -1 for female), age in years, income, years of education, employment status, race (coded 0 for White, 1 for non-White), handedness (coded 0 for predominantly right-handed, 1 for not predominantly right-handed), **psychopathology (i.e., scores on the DSM-oriented scales of depression, anxiety, ADHD, antisocial personality disorder, somatic disorder, avoidant personality disorder from the Adult Self-Report [ASR]³²), average**

framewise displacement during the resting state scans as a proxy for individual differences in head motion together with scores on fluid and crystallized/verbal intelligence (i.e., Progressive Matrices and Picture Vocabulary Test, respectively, ¹¹⁴). This analysis resulted in a t-statistic quantifying the degree to which individual differences in each FC estimate are associated biological sex, such that positive values represent stronger FC in males and negative values represent stronger FC in females. The absolute value of the t-statistic quantifies the magnitude of sexual differentiation for each FC index and, as such, it implicitly accounts for individual differences in sexual differentiation in the young adult group. A low absolute t-value denotes an FC index that shows greater between-individual, rather than between-sex, variability, whereas a high absolute t-value denotes an FC index that is consistently higher (positive t-value) or lower (negative t-value) across most males relative to females. The resulting matrix of t-values thus represents a reference of the degree to which FC estimates are sexually differentiated in healthy young adults.”

2b. Additionally, this analytical approach and the findings are not contextualized with prior literature, so it is unclear if this method has been validated in some manner.

We have revised our manuscript to better integrate our findings on regionally specific neural sexual differentiation with the prior literature on the somatomotor-association axis and neurodevelopment.

The revised sections (changed text in red font) are on

Lines 59-66

“Second, we characterized regionally specific patterns of sexual differentiation in brain function at ages 9-10 and 11-12, for each individual adolescent in the ABCD sample, using functional magnetic resonance imaging (fMRI). Subsequently, in the ABCD sample, we tested the relevance of neural sexual differentiation for concurrent and future pubertal development/physiological aging and associated patterns of psychiatric susceptibility (as identified at point 1) (Figure 2: Analysis 2). To contextualize our findings within the broader literature, we also probed the extent to which the regionally specific patterns of neural sexual differentiation thus identified tracked the sensorimotor-association (S-A) axis defined by prior work as a key organizing principle of neurodevelopment (i.e., association areas mature later than sensorimotor areas)^{33, 34}, including associated sex differences in patterns of functional brain network organization (i.e., association network regions are most effective at classifying youths based on sex) ³⁵.”

Lines 147-152

“Cross-validated ROI loadings on the brain LV tracked the canonical S-A axis (Spearman’s rho = -.44, pspin = 2 x 10⁻⁵), along which typical neurodevelopmental processes, including those associated with sex differences in functional brain architecture, have

been shown to unfold^{33, 34, 35}. Thus, greater feminization along the S-A axis (i.e., greater feminization of association, relative to sensorimotor, regions) at T1 was observed among female adolescents showing more advanced PhenoAge and pubertal development (particularly, gonadarche) at T2 (Figure 4-C-F), whereas greater masculinization along the S-A axis (i.e., greater masculinization of association, relative to sensorimotor, regions) typified male adolescents characterized by a younger PhenoAge and delayed pubertal development at T2.“

Lines 229-233

“Building on evidence implicating pubertal hormones and metabolic dysregulation in structural neurodevelopment^{54, 55} and mental health, particularly internalizing symptoms among female youth^{21, 22, 27, 28, 56, 57, 58, 59, 60}, we demonstrate that sexual differentiation along a previously identified sensorimotor-to-association (S-A) functional hierarchy⁶¹, an alleged key organizing principle of neurodevelopment and sex differences in brain organization^{33, 34, 35}, shows distinct relationships with physiological age, reproductive maturation and affective vs behavioral symptoms among male vs female adolescents.”

2c. Further, and importantly, was psychopathology assessed in the young adults? Could it be that a higher prevalence of anxiety and depression among female young adults would give rise to the link between brain feminization and anxiety?

Thank you for drawing this possibility to our attention. In the revised manuscript, we re-computed the matrix of t-values reflecting sexual differentiation of FC in the HCP sample by controlling not only for age, race, handedness, education, income, employment status (as we had done previously), but also for scores on the DSM-oriented scales from the Adult Self-Report (Achenbach, 2009), which match the CBCL scales (i.e., depression, anxiety, ADHD, antisocial personality disorder, somatic disorder, avoidant personality disorder) in addition to average framewise displacement during the resting state scans as a proxy for individual differences in head motion together with scores on fluid and crystallized/verbal intelligence (i.e., Progressive Matrices and Picture Vocabulary Test, respectively, Barch et al., 2013) as measures of overall cognitive function. All the results reported in our prior manuscript version remained unchanged and, thus, are unlikely to be explained by sex differences in psychopathology, head motion, intelligence or demographic variables. The above information has been included on lines 488-495 of our revised manuscript:

“In the HCP sample, we regressed each FC estimate against sex (coded for 1 for male, -1 for female), age in years, income, years of education, employment status, race (coded 0 for White, 1 for non-White), handedness (coded 0 for predominantly right-handed, 1 for not predominantly right-handed), psychopathology (i.e., scores on the DSM-oriented scales of depression, anxiety, ADHD, antisocial personality disorder, somatic disorder, avoidant personality disorder from the Adult Self-Report [ASR]³²), average

framework displacement during the resting state scans as a proxy for individual differences in head motion together with scores on fluid and crystallized/verbal intelligence (i.e., Progressive Matrices and Picture Vocabulary Test, respectively, ¹¹⁴). This analysis resulted in a t-statistic quantifying the degree to which individual differences in each FC estimate are associated biological sex, such that positive values represent stronger FC in males and negative values represent stronger FC in females.”

3. Throughout the Results section there is a general lack of explanation of each construct, analysis approach, and findings, as well as the motivations/goals of each (in particular Analysis 1). Additionally, linkages between specific constructs and the measures used to probe them are generally lacking clarity. For example, in Analysis 2 it is not clear if “physiology” in the first sentence is referring to PhenoAge or DHEA or both. Additionally, it is unclear what is meant by “ROI-specific FC patterns”; “LV” is undefined; and “relating T1 and T2 regional sexual differentiation” could mean change between these timepoints or in each of these timepoints. In another example, the phrase “and the confounders listed under ‘Control variables’” is not precise enough for the Results section. A more precise description of the inputs and outputs of the CCA, PLS, and PRS methods are needed to interpret the results. Additionally, it is unclear if Analysis 5 is using HCP-D (as in the section title) or ABCD (as in the text). In another instance, it is unclear if brain masculinization and feminization were related to both sexes or only males and females respectively, or if this differed between the analyses. These methodological details become more clear in the Methods (or Discussion) sections. However, since the Discussion and Methods come after the Results, the Results section should contain sufficient information to allow the reader to understand the analyses and findings.

The Results section has been revised extensively based on the reviewer’s comment (please see sections in red font on lines 80-226). With regards to specific comments, in the revised manuscript, we only refer to the “physiology CCA variate” with the contributing variables being identified on lines 112-114. We eliminated the terminology “ROI-specific FC patterns” from our revised manuscript. “LV” is written in full as “latent variable” (LV) on its first appearance (line 138). For each analysis, we list all the confounders for which we controlled (lines 121-122, 140-141, 167-168, 207, 219-220).

The inputs and outputs of the CCA and PLS analyses are described in detail (with added text in red font) on

Lines 109-114

(Analysis 1) “To this end, we conducted a CCA in our target ABCD sample (N = 199) to identify maximally correlated latent factors (i.e., variates) linking the 18 CBCL scores (i.e., 9 scores from the two-year follow-up [ages 11-12] and 9 change scores estimated between the two- and three-year follow-ups : ages 11-12 to 12-13]; hereafter called the T2/ $\Delta_{T2,T3}$ psychopathology CCA variate) to

sex assigned at birth (corroborated through single nucleotide polymorphism (SNP) analysis and/or menstrual history), PhenoAge, pubertal hormone levels, and self-reported adrenarche/gonadarche assessed at T2 (hereafter called the T2 physiology CCA variate).”

Lines 129-137

(Analysis 2) “Having established the sexually dimorphic and primarily longitudinal relationship between physiology and psychopathology in the ABCD sample, we next sought to shed light on its relevance to patterns of neural sexual differentiation. We thus entered the $T2/\Delta_{T2,T3}$ psychopathology and T2 physiology CCA variates from analysis 1 into a partial least squares (PLS) correlation (Analysis 2) to examine their associations with ROI-specific measures of sexual differentiation in FC, as identified at T1 (ages 9-10) and T2 (ages 11-12). Sexual differentiation in the ABCD sample was characterized in reference to the functional connectivity (FC) patterns observed among the young healthy adult participants (ages 22-30) in the Human Connectome Project (HCP). These individuals have reached reproductive maturity and are thus expected to show maximal differentiation of brain function related to their biological sex (as assigned at birth [self-reported] and confirmed through menstruation history^{12, 16, 17, 18, 41, 42}). ROI-to-ROI FC estimates based on two widely used functional atlases^{43, 44} were extracted independently from the ABCD and HCP samples, respectively.”

Lines 155-172

(Analysis 3) “Our first two analyses provided evidence that physiology at ages 11-12 years shows longitudinal associations with earlier (but not concurrent) neural sexual differentiation (Figure 4-B) and increases in psychopathology, rather than concurrent psychiatric symptoms (Figure 3-A). Building on these findings, we next examined whether sexual differentiation in brain function and physiology sequentially mediate the relationship between genetic risk and psychopathology in the ABCD sample. To this end, we specified the serial mediation model depicted in Figure 2 (Analysis 3), tested with PROCESS 4.2. In line with existing guidelines for probing causal chains⁴⁶, the serial mediators were sampled from different time points. Based on the results from our first two analyses, we were primarily interested in whether the T1 neural sexual differentiation LV from Analysis 2 (mediator 1) and the sex-dependent physiology CCA variate from Analysis 1 (mediator 2) would constitute a viable causal chain linking polygenic risk scores (PRS) for anxiety and, potentially, depression to rising internalizing symptoms (cf Figure 3-A). In testing this sequence, we simultaneously accounted for alternate models involving serial (T1 and T2 neural sexual differentiation from Analysis 2) and parallel (T2 Physiology from Analysis 1 and T2 neural sexual differentiation from Analysis 2) mediation, respectively (see Figure 2-Analysis 3). The absence of blood chemistry data at T1 prevented us from computing PhenoAge and, thus, estimating the alternate serial and parallel mediation models involving T1 Physiology. However, prior to mediation analysis, all variables were adjusted for T1

pubertal hormone levels and self-reported adrenarche/gonadarche based on the PDS (in addition to participant motion, age, race, testing site, handedness, adoption status, ambiguous biological sex). Additionally, inclusion of the PRS-T2 Physiology-Psychopathology path allowed us to estimate indirect effects of physiology that are independent of T1/T2 neural sexual differentiation and therefore uniquely relate to T1 PhenoAge.

PRSs for anxiety, depression and ADHD were computed using summary statistics from large and independent disorder-focused genome-wide association studies (GWASs), featuring case/patient-control comparisons (i.e., MDD ⁴⁷, anxiety disorders ⁴⁸, ADHD ⁴⁹). We did not include PRSs for Oppositional Defiant or Conduct Disorder because we could not locate relevant case-control GWASs.”

Lines 189-193

(Analysis 4) “In our final set of analyses, we tested whether the sex-dependent links between brain function, psychiatric risk, and physiological maturation, as established in the ABCD sample (Analyses 1-3) would extend cross-sectionally in the larger HCP-D sample. In Analysis 4, we therefore used CCA to probe whether individual differences in sexual differentiation of regional FC profiles (as defined using the PLS-extracted latent variable from Analysis 2) (variate 1) relate to scores on the CBCL scales, pubertal development, and markers of physiological aging (BMI, exposure to poverty) in a sex-dependent manner (variate 2).”

lines 217-220

(Analysis 5) “ Our final analysis linked the cross-validated sexual differentiation brain LV from the ABCD (Analysis 2, Figure 4-D [but unthresholded]) to the partial correlation maps quantifying the relationship between neural sexual differentiation and the DSM-oriented CBCL scores, pubertal development (i.e., adrenarche, gonadarche), and markers of physiological aging as identified in the ABCD (BMI, financial deprivation) (adjusted for sex, age, handedness, testing site, participant motion and adoption status).”

A brief description of the PRS methods is included in the Analysis 3 Results section on lines 170-172 which read as follows

“PRSs for anxiety, depression and ADHD were computed using summary statistics from large and independent disorder-focused genome-wide association studies (GWASs), featuring case/patient-control comparisons (i.e., MDD ⁴⁷, anxiety disorders ⁴⁸, ADHD ⁴⁹). We did not include PRSs for Oppositional Defiant or Conduct Disorder because we could not locate relevant case-control GWASs.”

Analysis 5 linked the sexual differentiation LV identified in the ABCD sample in Analysis 3 to the partial correlation maps linking neural sexual differentiation to psychopathology, BMI, deprivation and pubertal development in the HCP-D sample. Hence, to avoid confusion, given the cross-sample (ABCD-HCP-D) comparison, we dropped “HCP-D” from the Analysis 5 sub-heading.

The sex-dependence/sex-dependence of our analyses is underscored in our revised Figure 2 (which was previously part of Figure 1). The sex-(in)dependence of our reported results is also specified in the Results section corresponding to each analysis, please lines 123-126 (Analysis 1), lines 149-152 (Analysis 2), line 161 (Analysis 3), lines 207-210 (Analysis 4) and lines 224-225 (Analysis 5).

4. Despite substantial evidence that head motion causes widespread functional connectivity alterations (Power et al., 2012; Satterthwaite et al., 2012, 2019; Van Dijk et al., 2012; Ciric et al., 2017), in particularly in youth, the paper is lacking a discussion of the removal of head motion artifacts. While the HCP-D data was processed with ICA-FIX, the ABCD Study processing contains no mention of motion scrubbing (or other method of removal). Please assess relationships between head motion and functional connectivity estimates in both samples to report if head motion artifacts were sufficiently removed from the brain data. Additionally, if motion scrubbing was implemented in the ABCD Study, please describe the procedure used and list the amount of data remaining (mean, standard deviation, range) after motion scrubbing.

As detailed on lines 471-476 of our revised manuscript (also present in our initial submission), prior to computing the ROI-to-ROI FC matrices in the ABCD data, we used linear regression to apply the following corrections to each ROI's time series (separately within each run): (1) removal of the quadratic trends and 24 motion terms (i.e., the six motion parameters, their first derivatives, and squares from the time course of each parcel (to correct for head motion artifacts and scanner noise [quadratic trend]); and (2) the mean time courses of cerebral white matter (WM), gray matter (GM), cerebrospinal fluid (CSF) (to correct for head motion and physiological motion artifacts). Please note that, prior to being regressed (as part of Step 1), the motion terms had been filtered to eliminate signals within the respiratory effect range, a step that had been shown to lead to more effective removal of head motion-related artifacts (i.e., .31-.43 Hz, cf. Fair et al., 2020, NeuroImage). These motion correction steps are in line with those recommended by the ABCD study team (cf. Hagler et al., 2019).

Additionally, for all 3 samples (ABCD, HCP-D, HCP), we followed existing guidelines in the literature (Yan et al., 2013, NeuroImage; Power et al., 2015, NeuroImage) to ensure that residual head motion artifacts do not impact our group-level effect estimates. Thus, in all the analyses using the ABCD and/or HCP-D neural differentiation data (Analyses 2-5) and in the computation of the neural sexual differentiation matrix in the HCP sample, we controlled for participant-level motion (i.e., average frame-wise displacement [FD], cf. Power et al., 2015) (please see lines 140, 167, 207, 220, 491-492, 521-522, 617-619). For the ABCD participants, average

FD was computed in reference to the 3 runs contributing to the analyses at each time point (6 runs in total across Time 1 and Time 2).

Given the above controls and the “good quality” of the analyzed ABCD resting state data, we saw no reason to do any motion scrubbing and introduce the confound of between-participant differences in time series length in the ABCD study. “Good quality” was defined based on the “iqc_rsfmri_ok_ser “ and the run-specific “_pc_score variables from the “mri_y_qc_raw_rsfmr.csv” file in the ABCD data release 5.1 as participants who had at least 3 completed protocol compliant time series who passed quality control at both Time 1/ages 9-10 and Time 2/ages 11-12. This information is included in the legend for Figure 7, which contains the STROBE flowchart.

5. The Methods section lacks any details on how the DHEA information reported in the ABCD Study was processed to ensure accuracy. At minimum, other ABCD Study papers describing how the hormone samples were collected and initially processed should be cited (e.g., Uban et al. (2018) Biospecimens and the ABCD study: Rationale, methods of collection, measurement and early data) and the steps you took to remove problematic samples should be described (e.g., considerations of sample color, timing). Please also report the mean, standard deviation, and range of (a) DHEA samples for each sex and (b) parent and youth pubertal development scores in the ABCD Study. Additionally, the correlation between DHEA and pubertal development score in the HCP-D study is missing.

We thank the reviewer for their suggestion. In our revised manuscript, we indexed pubertal maturation via self-reports of adrenarche and gonadarche (collected with the PDS) as well as corresponding sex-specific hormonal indices. Details on the processing of saliva samples are now included on lines 344-363

“Pubertal hormone levels were extracted from saliva samples and processed as described in ⁹⁹. Hormonal markers of adrenarche and gonadarche were estimated separately for boys and girls. In girls, adrenarche was estimated as the average of the standardized DHEA and testosterone levels, whereas standardized estradiol levels were used as an index of gonadarche (cf. ³⁷). In boys, standardized DHEA levels were used as a marker of adrenarche, while standardized testosterone levels were used as an indicator of gonadarche given its role in reproductive maturation/genital development in male youth (for a review of supporting findings, see ³⁸). Using the information available in the “ph_y_sal_horm.csv” file from the ABCD data release 5.1, hormonal indices of adrenarche and gonadarche were residualized separately within each sex for saliva quality-related concerns (i.e., low quantity, discoloration, excessive bubbles, potential contamination, other concerns), activity levels and caffeine quantity in the 12 hours preceding saliva collection and the delay between waking time and saliva collection (cf. ⁹⁹). The saliva quality variable was dummy-coded as “1” (“no concerns”) or “0” (“concerns present”) since only 4 youths at T1 and 2 youths at T2 showed more than one saliva

quality-related issue. Across the full sample, concerns for the saliva samples were noted for 20 youths at T1 and 14 youths at T2. For female youths, the hormonal indices were further residualized by self-reported age at menarche, days since the start of the last menstrual cycle and whether the cycles were regular or not (cf. ⁹⁹). None of the female participants reported taking birth control pills. The hormonal indices of adrenarche and gonadarche, residualized and standardized separately within each sex, were combined across the full sample and entered in the reported analyses. Table 1 below contains summary statistics for hormonal indices, as well as youth and parental ratings of pubertal development. At Time 2, partial correlation analyses controlling for sex revealed a robust association between self-reported adrenarche and the adrenal hormonal index (DHEA[boys]/DHEA/testosterone [girls], r of .24, $p < .001$), as well as the gonadal hormonal index (testosterone [boys]/estradiol [girls], r of .34, $p < .001$). At the same time point, a robust correlation (adjusted for sex) was detected between self-reported gonadarche and the gonadal hormone index (r of .25, $p < .001$) as well as the adrenal hormone index (r of .19, $p = .008$). In contrast, at Time 1, we observed a robust correlation (adjusted for sex) only between self-reported adrenarche and the adrenal hormone index (r of .16, $p = .022$.)”

Table 1
Summary Statistics for the Pubertal Development Measures Collected at Time 1 and Time 2

Variable	ABCD (N = 199)	
	Time 1 (Ages 9-10)	Time 2 (Ages 11-12)
	Female M (SD, range)/Male M (SD, range)	Female M (SD, range)/Male M (SD, range)
DHEA*	77.20 (43.14, 5.09-188.09)/61.08 (41.62, 8.00-215.65)	115.53 (67.74, 13.50-370.10)/84.44 (69.32, 3.12-314.47)
Testosterone*	37.26 (16.06, 5.22-73.00)/36.97 (23.58, 8.41-224.03)	51.71 (21.31, 14.68-112.35)/54.63 (33.51, 11.73-168.58)
Estradiol*	1.01 (.45, .13-2.53)	1.15 (.63, .34-5.29)
Adrenarche (self-report)	1.61 (.62, 1-3.50)/1.60 (.61, 1-4)	2.37 (.73, 1-4)/1.97 (.72, 1-4)
Gonadarche (self-report)	1.62 (.55, 1-3)/1.66 (.53, 1-4)	2.26 (.75, 1-4)/1.83 (.52, 1-3)
Adrenarche (parental report)	1.56 (.60, 1-3)/1.36 (.53, 1-3)	2.35 (.76, 1-4)/2.04 (.77, 1-4)
Gonadarche (parental report)	1.78 (.39, 1-3)/1.58 (.42, 1-3)	2.47 (.68, 1-3.67)/1.84 (.61, 1-4)

Note. M = mean. SD = standard deviation.

As for the HCP-D sample, as we specify on lines 377-380 of the Methods (added text in red font), “**HCP-D. Hormonal measures**

of pubertal development are unavailable in the latest existing data release (2.0) from the HCP-D. Hence, to index reproductive maturation, we relied on self-reports of adrenarche (Female youth: $M = 2.87$, $SD = .70$; Male youth: $M = 2.47$, $SD = .67$; score range: 1-4) and gonadarche (Female youth: $M = 3.09$, $SD = .87$; Male youth: $M = 2.19$, $SD = .72$; score range: 1-4) which were strongly correlated ($r(143) = .70$ [female youth] and $r(130) = .54$ [male youth], both $ps < .001$) (parental reports were not available for the full sample)."

6. It is unclear how the serial mediation analysis uncovered the “temporal sequence” of relationships since only one model, with T1 brain LV then T2 physio CCA, was tested. Would T1 physio then T2 brain also mediate the relationship? What about concurrent measures as the serial mediators (e.g., T1 brain then T1 physio)?

In response to this comment, we have revised Analysis 3 to control for alternate paths, such as parallel mediation via T2 Physiology and T2 sexual differentiation, as well as the alternate serial mediation path involving T1 and T2 sexual differentiation (please see Figure 2, Analysis 3). The absence of blood chemistry data at T1 prevents us from computing PhenoAge and, thus, estimating the alternate serial and parallel mediation models involving T1 Physiology. However, prior to mediation analysis, all variables were adjusted for T1 pubertal hormone levels and self-reported adrenarche/gonadarche based on the Pubertal Development Scale. Additionally, inclusion of the genetic risk-T2 Physiology-Psychopathology path (Figure 2, Analysis 3) allowed us to estimate indirect effects of physiology which are independent of T1/T2 neural sexual differentiation and, thus, potentially, uniquely related to T1 PhenoAge. Please note that in the mediation model anchored in the anxiety PRS (i.e., the only model that yielded robust indirect effects), the T1 sexual differentiation-T2 Physiology was the only reliable indirect effect path (please see Figure 5-A).

As we note on lines 159-160 of our Results for Analysis 3, serial mediation probes the viability of potential causal chains. As such, it is theoretically meaningful only if the mediators are sampled from different time points. In the case of T1 sexual differentiation/T1 Physiology or T2 sexual differentiation/T2 physiology, temporal precedence cannot be established, hence a serial mediation analysis would not yield meaningful results. Instead, investigation of parallel mediation (as we implemented in our revised manuscript) would be advised.

Minor comments:

1. Why was such a small portion of subjects in the ABCD Study used when more than 700 subjects have resting state data from both the baseline year and follow-up-year-two? Please explain why ABCD subjects were excluded from this project.

The ABCD data release 5.1, which includes all the available data from the two-year follow-up assessment, contained only 199 ABCD study participants who were biologically unrelated to one another and had good quality data on all the variables included in our analyses. Since our analyses were closely inter-related, inclusion of different participant samples in each test would have made interpretation of the results much more difficult. In our revised manuscript, our participant selection steps for the ABCD and HCP-D samples are illustrated in a STROBE diagram (please see Figure 7). Please note that the HCP-D sample now contains 277 (rather than 308) participants because in our initial submission we failed to eliminate participants sharing the same family ID (and, thus, likely to be biologically related).

2. Generally, the abstract lacks clarity necessary to understand the study from the abstract alone. For example, it is not explained how “sexual differentiation” and “physiological aging” are operationalized. Additionally, it is unclear from the abstract if the study focused on biological females or included both sexes. More precise wording with clear definition of terms should be employed.

In the revised Abstract, we have specified that we included both female and male youths. We replaced “physiological aging” with “greater pubertal development and immune/metabolic dysregulation”. We also specified that “sexual differentiation” was operationalized as “relative similarity to the neural functional coupling (FC) patterns that vary between adult male and female participants in the Human Connectome Project (N = 336)”.

3. The use of PhenoAge to capture physiological aging is mentioned in the Introduction, but an explanation of what information is captured by PhenoAge is lacking. Similarly PhenoAge is mentioned in the Results section, but is not defined so an understanding of the results is difficult without reading the Methods first.

In the Introduction (please see lines 56-57), we specified that PhenoAge assesses immune/metabolic dysregulation. A brief overview of the PhenoAge algorithm is provided in the revised Figure 1 and its legend.

In the revised Results for Analysis 1 we added further details on the PhenoAge algorithm. The revised section on lines 88-107 reads as follows:

“Physiological wear-and-tear was estimated with the PhenoAge algorithm³⁹ which uses an exponential growth formula to predict mortality from blood chemistry measures of metabolic and immune system functioning in a reference group. The PhenoAge algorithm yields a biological age prediction which corresponds to the chronological age at which an individual’s mortality risk would be normal in a reference group. A more advanced PhenoAge (relative to chronological age) indicates greater physiological wear-

and-tear and, thus, greater than expected (by chronological age) risk for disease and mortality^{39, 40}. Using the blood chemistry data available in the ABCD sample at T2/ages 11-12 (i.e., the earliest available time point), we trained and validated the PhenoAge algorithm in two young adult (ages 20-40) samples from the National Health and Nutrition Examination Survey (NHANES) (<https://wwwn.cdc.gov/nchs/nhanes/Default.aspx>) (NHANES III: N = 6,084; 2,892 males; NHANES IV: N = 14,782; 7,421 males). We focused on young adults for whom, where applicable, deaths were related to intrinsic causes (e.g., cardiac, cancers, kidney/lung-related, respiratory infections, diabetes) rather than accidents, because we reasoned that they would best reflect physiological wear and tear processes that are distinguishable from those linked to typical aging and, thus, most likely to be observed among the adolescents in the ABCD cohort. In both NHANES samples, more advanced PhenoAge than expected based on chronological age, was observed among poorer individuals (both *r*s of .16, *p* < .001), was linked to higher body mass index (BMI) (*r*s of .22 [NHANES III] and .29 [NHANES IV], both *p*s < .001) and poorer overall self-reported health (*r*s of .16 [NHANES III] and .20 [NHANES IV], both *p*s < .001). We replicated these associations in the larger ABCD sample of biologically unrelated youths with available blood chemistry, BMI, financial deprivation and medical history data at the two- or three-year follow-up (N = 922). Specifically, in these youths, a cross-validated canonical correlation analysis (CCA) linked more advanced PhenoAge (adjusted for chronological age and sex) to parental reports of financial deprivation, higher BMI and more serious medical problems, as indicated by the number of emergency room visits and unplanned medical visits, particularly those related to very severe headaches, episodes of high fever, asthma, bronchitis, allergies and diabetes. Importantly, the association between PhenoAge and the physical health/deprivation variate was replicated in our target sample of 199 ABCD participants (see Methods for detailed analyses and Figure S1 for a representation of the pertinent CCA results)."

4. In the Methods section, it is not stated if the PhenoAge algorithm trained in adults can be reasonably applied to youth. Additionally, it is not stated if the validity of the use of the PhenoAge algorithm in HCP-D data, which is lacking all blood chemistry measures, produces valid results. It seems that this could be explored in the ABCD data comparing PhenoAge estimates with and without the blood chemistry measures.

In the revised Results for Analysis 1 (please see our prior reply), the revised methods (please see lines 403-422) and Figure S1, we provide evidence that, similarly to the results observed in the two NHANES samples, PhenoAge predicts higher body mass index, greater exposure to poverty and more severe medical problems in the ABCD sample.

As for the HCP-D sample, unavailability of blood chemistry data relevant to PhenoAge prevented us from including this variable in the HCP-D analyses (PhenoAge as an algorithm has been validated on blood chemistry data). However, as we specify on lines 201-203 of our revised manuscript, the main purpose of the cross-sectional HCP-D analyses was to probe the viability of neural sexual differentiation as a potential psychiatric vulnerability marker linked to other observable individual characteristics, such as

body mass index, exposure to poverty, which had been linked to PhenoAge in the ABCD sample.

5. The use of DHEA to characterize pubertal development is underexplained. Please explain the stage/aspects of pubertal development captured by DHEA (e.g., Adrenarche) and if/how differences between sexes were treated.

In our revised manuscript, we indexed pubertal maturation via self-reports of adrenarche and gonadarche (collected with the PDS), as well as corresponding sex-specific hormonal indices.

The revised pubertal maturation index, as well as the relationship between self-reported and hormonal indices of pubertal maturation are briefly described in the Results on lines 82-87

“Self-ratings on the Pubertal Development Scale [PDS] ³⁶ and hormonal assays were used to estimate progression through the two stages of puberty which are linked to the emergence of secondary sex characteristics (i.e., adrenarche), as well as reproductive maturation and development of sex-specific characteristics (i.e., gonadarche) ^{37, 38}. Hormonal markers of adrenarche and gonadarche were computed separately for boys and girls. In girls, average standardized DHEA and testosterone levels indexed adrenarche, whereas standardized estradiol levels gauged gonadarche (cf. ³⁷). In boys, standardized DHEA levels assessed adrenarche, while standardized testosterone levels were used as an indicator for gonadarche (for a review of supporting findings, see ³⁸).“

More detailed descriptions of the revised pubertal development index, the rationale behind its computation and the sex-specific relevance of DHEA, testosterone and estradiol are included in the Methods on lines 332-348

“**ABCD.** In line with prior work ^{37, 97}, pubertal maturation was assessed via self-ratings on the Pubertal Development Scale (PDS, ³⁶) and hormonal assays, specifically, dehydroepiandrosterone (DHEA), testosterone and, for female youth only, estradiol. Based on self-reports and hormonal measures, we independently estimated progression through the two stages of puberty, adrenarche and gonadarche. Adrenarche is the earlier stage of puberty, which starts around ages 5-7 and is characterized by release of adrenal hormones, which, in turn, support the development of secondary sex characteristics, such as pubic hair growth and acne, in both male and female youth (for review of relevant findings, see ^{37, 38}). Maturation of the hypothalamic-pituitary-gonadal (HPG) axis marks the second stage of puberty, gonadarche, in which the release of sex hormones (i.e., estradiol in girls, testosterone in boys) leads to reproductive maturation and the development of sex-specific characteristics ³⁸. Following prior studies ^{37, 38, 98}, in the PDS, adrenarche was estimated via self-reported skin changes and body hair growth for both boys and girls,

whereas gonadarche was estimated via self-reported growth spurt, breast development and menarche in girls, and self-reported growth spurt, deepening of the voice and facial hair growth in boys. At Time 1 only (ages 9-10), 9 youths failed to complete any of the adrenarche-relevant items and 5 youths failed to complete any of the gonadarche-relevant items on the PDS. For these participants, at Time 1 only, we used the corresponding parental ratings.

Pubertal hormone levels were extracted from saliva samples and processed as described in ⁹⁹. Hormonal markers of adrenarche and gonadarche were estimated separately for boys and girls. In girls, adrenarche was estimated as the average of the standardized DHEA and testosterone levels, whereas standardized estradiol levels were used as an index of gonadarche (cf. ³⁷). In boys, standardized DHEA levels were used as a marker of adrenarche, while standardized testosterone levels were used as an indicator of gonadarche given its role in reproductive maturation/genital development in male youth (for a review of supporting findings, see ³⁸).”

Further details on the processing of hormonal assays and relevant sex-specific covariates used in estimating hormonal indices of adrenarche vs gonadarche are included in response to this reviewer’s Major Point 5.

6. There are a couple issues regarding relationships with the sensorimotor-association axis. First, the S-A axis is not described, including in terms of its relevance for development or the relevance for the findings in this paper (e.g., what does the effect relating to the S-A axis tell us about brain development). Second, when the S-A axis is mentioned in the Results the direction of relationship is not mentioned. For example, does “greater feminization along the S-A axis” mean that association regions are more feminine or that individuals with more feminine brains look more like adult females on both ends of the S-A axis?

We now explain the motivation for looking at the S-A axis in the Introduction (please see lines 59-66, changed text is in red font):

“Second, we characterized regionally specific patterns of sexual differentiation in brain function at ages 9-10 and 11-12, for each individual adolescent in the ABCD sample, using functional magnetic resonance imaging (fMRI). Subsequently, in the ABCD sample, we tested the relevance of neural sexual differentiation for concurrent and future pubertal development/physiological aging and associated patterns of psychiatric susceptibility (as identified at point 1) (Figure 2: Analysis 2). To contextualize our findings within the broader literature, we also probed the extent to which the regionally specific patterns of neural sexual differentiation thus identified tracked the sensorimotor-association (S-A) axis defined by prior work as a key organizing principle of neurodevelopment (i.e., association areas mature later than sensorimotor areas)^{33, 34}, including associated sex differences in patterns of functional brain network organization (i.e., association network regions are most effective at classifying youths based on sex) ³⁵.”

Furthermore, the Analysis 2 results related to the S-A axis are explained in greater detail on lines 147-152

“Cross-validated ROI loadings on the brain LV tracked the canonical S-A axis (Spearman’s rho = -.44, pspin = 2×10^{-5}), along which typical neurodevelopmental processes, including those associated with sex differences in functional brain architecture, have been shown to unfold^{33, 34, 35}. Thus, greater feminization along the S-A axis (i.e., greater feminization of association, relative to sensorimotor, regions) at T1 was observed among female adolescents showing more advanced PhenoAge and pubertal development (particularly, gonadarche) at T2 (Figure 4-C-F), whereas greater masculinization along the S-A axis (i.e., greater masculinization of association, relative to sensorimotor, regions) typified male adolescents characterized by a younger PhenoAge and delayed pubertal development at T2.”

7. In the ABCD Study, resting state scans are collected in 5-minute runs. The Methods section reports that 3 resting state runs were used in each year (6 runs total). How does this add up to 25 minutes of resting state in each year? It seems there could only be 15 minutes total resting state collected per year.

We apologize for the confusion: the 4 resting state runs in the HCP-D amount to 25 minutes, whereas the 4 resting state runs collected in the ABCD amount indeed to 20 minutes. In the Methods, we approximated duration to 25 minutes for developmental samples. We realize that this could be confusing, hence in the revised manuscript (lines 450-453), we state that “Four resting state fMRI scans (eyes open with passive crosshair viewing), each lasting approximately 15 minutes (HCP), 6 minutes (HCP-D) and 5 minutes (ABCD), respectively, were collected from all three samples (ABCD: baseline/2-year follow-up; HCP/HCP-D: 1 timepoint). For the ABCD participants, due to data availability, we used only 3 scans from each time point (6 resting state scans in total) for a total of approximately 15 minutes at each timepoint.”

On lines 483-486, we also specify that “In each sample, the correlations were run separately for each run using a duration of approximately 5 minutes (417 volumes for HCP; 375 volumes for ABCD/HCP-D). The resulting ROI-to-ROI (300 x 300) matrices, each estimated over a roughly 5-minute interval for all three samples, were averaged across the 3 (ABCD) or 4 (HCP/HCP-D) runs available at each time point.”

8. The Methods section is missing several data processing citations (e.g., fMRIPrep).

In our revised manuscript (lines 469-470), we added that “minimal processing of the ABCD data had been performed using a Multi-Modal Processing Stream¹⁰⁶, which primarily combines MATLAB, Freesurfer¹⁰⁷, FSL¹⁰⁸ and AFNI¹⁰⁹ functions”. We also specified

that we used FSL and MATLAB to further preprocess the ABCD data (please see line 471). For the HCP-D and HCP samples, we provided a brief overview of the HCP Preprocessing Pipelines applied to these data. Word limits do not allow us to go in further detail.

9. The Methods section states that polygenic risk scores were identified for MDD and ADHD in addition to Anxiety; but a discussion of the use of these in mediation analyses is lacking from the Results.

In our revised manuscript, we present the results of the mediational analyses anchored in the MDD and ADHD PRS, respectively (please see lines 181-185 and Figures 5-B, S4, S5). The ADHD PRS mediation model was included as a specificity test: no robust indirect effects were expected since ADHD did not have a reliable loading on the CCA Psychopathology variate from Analysis 1 (please see Figure 3-A). The MDD PRS mediation analyses were exploratory: the lack of robust indirect effects was not particularly surprising since the CCA Psychopathology variate from Analysis 1 seemed to capture predominantly change in anxiety disorder-related scores (Figure 3-A). The revised section on lines 181-185 read as follows:

“The mediational analysis anchored in the MDD PRS yielded no robust indirect effects, but instead yielded a reliable direct effect (Figure 5-B). This pattern of results implies that genetic risk for MDD predicts worsening depression and anxiety disorder-related symptoms from ages 11-12 to ages 12-13. However, this link between the MDD PRS and affective problems is not mediated by the neural sexual differentiation and physiology patterns herein identified. Finally, we confirmed the specificity of the sequence anchored in the anxiety PRS by re-running the serial mediation analysis with ADHD PRS as the predictor, which revealed no significant indirect effect (Figure S5).”

10. Generally the Methods section is lacking precision. It seems that topics (e.g., PLS) are brought up out of order or without prior context. For example, in the “Sensorimotor-Association (SM-A) axis” section, the sentence “Spearman’s rank correlation between the cross-validated ROI loadings extracted with PLS” mentions PLS but it is not clear what is in the PLS at this point.

As outlined in our response to this Reviewer’s comment 3, we have revised our Results section to include details on the inputs and outputs of each analysis, including PLS/Analysis 3 (please see lines 130-137). As such, description of the PLS analysis precedes the Methods section flagged by the reviewer. To further enhance clarity, we have revised the above-mentioned Methods as follows (please see lines 508-509) (added text in red font):

“Subsequently, for each of the two functional atlases, we conducted a Spearman’s rank correlation between the cross-validated ROI loadings extracted with PLS from the ABCD sample in Analysis 2 (for details, see Figure 2-Analysis 2 and the Analysis 2 section in the Results) and the S-A axis ranks.”

Reviewer #3 (Remarks to the Author):

This paper sought to examine the mediating role of pubertal factors and physiological aging on the link between polygenic risk scores and psychopathology symptoms. One strength of the current study is the use of two separate samples to test and replicate the effects observed in the sequential mediation. In addition, examining sexual differentiation in a continuous fashion in the context of neural development is an innovative premise to tackling sex-dependent risk for psychopathology. However, there are crucial aspects of the conceptual motivation and methods used in the study that limit my enthusiasm for the potential impact the results may have.

1. First, the use of DHEA as a pubertal index is concerning in this context. There are other pubertal indices available in ABCD (i.e., Pubertal Development Scale (PDS), and more relevant hormones) that would be more suited to this question and would make the comparisons with HCP-D more equivalent. To the first point, DHEA is not traditionally linked to sex differences neurally or behaviorally per se. It is also more directly linked to adrenarche, not gonadarche. In other words, the authors claim to be interested in examining the HPG axis, which regulates gonadarche, while the HPA axis regulates the production of DHEA (in the adrenal gland) and adrenarcheal processes. In cases where there are sex differences in DHEA or as a function of adrenarche, they tend to be less extreme than those attributable to gonadal hormones (e.g., testosterone, estradiol). Therefore, if hormonal assays are of interest, it seems more suitable to examine testosterone and estradiol to address this question, both of which are available in ABCD. In terms of comparability across ABCD and HCP-D, it would be more straightforward to examine PDS since it is available in both samples. Relatedly, the authors mention using DHEA to “stage” participants. The gold standard for staging is through the use of Tanner staging. Hormone levels are relative to the individual and cannot necessarily speak to what pubertal stage individuals are in through the currently available methods. Using the PDS (which can be converted to Tanner stages; Shirtcliff et al., 2009) would be a more reasonable and conventional approach if pubertal stages are of interest.

We thank the reviewer for their suggestion. In our revised manuscript, we changed the pubertal maturation index so that it would be based on self-reports of adrenarche and gonadarche (collected with the PDS), as well as corresponding sex-specific hormonal indices. All the results reported in the initial submission hold in the revised manuscript.

The revised pubertal maturation index is briefly described in the Results on lines 82-87

“Self-ratings on the Pubertal Development Scale [PDS] ³⁶ and hormonal assays were used to estimate progression through the two stages of puberty which are linked to the emergence of secondary sex characteristics (i.e., adrenarche), as well as reproductive maturation and development of sex-specific characteristics (i.e., gonadarche) ^{37, 38}. Hormonal markers of adrenarche and gonadarche were computed separately for boys and girls. In girls, average standardized DHEA and testosterone levels indexed adrenarche, whereas standardized estradiol levels gauged gonadarche (cf. ³⁷). In boys, standardized DHEA levels assessed adrenarche, while standardized testosterone levels were used as an indicator for gonadarche (for a review of supporting findings, see ³⁸). “

A more detailed description of the revised pubertal development index and the rationale behind its computation are included in the Methods on lines 344-363

“Pubertal hormone levels were extracted from saliva samples and processed as described in ⁹⁹. Hormonal markers of adrenarche and gonadarche were estimated separately for boys and girls. In girls, adrenarche was estimated as the average of the standardized DHEA and testosterone levels, whereas standardized estradiol levels were used as an index of gonadarche (cf. ³⁷). In boys, standardized DHEA levels were used as a marker of adrenarche, while standardized testosterone levels were used as an indicator of gonadarche given its role in reproductive maturation/genital development in male youth (for a review of supporting findings, see ³⁸). Using the information available in the “ph_y_sal_horm.csv” file from the ABCD data release 5.1, hormonal indices of adrenarche and gonadarche were residualized separately within each sex for saliva quality-related concerns (i.e., low quantity, discoloration, excessive bubbles, potential contamination, other concerns), activity levels and caffeine quantity in the 12 hours preceding saliva collection and the delay between waking time and saliva collection (cf. ⁹⁹). The saliva quality variable was dummy-coded as “1” (“no concerns”) or “0” (“concerns present”) since only 4 youths at T1 and 2 youths at T2 showed more than one saliva quality-related issue. Across the full sample, concerns for the saliva samples were noted for 20 youths at T1 and 14 youths at T2. For female youths, the hormonal indices were further residualized by self-reported age at menarche, days since the start of the last menstrual cycle and whether the cycles were regular or not (cf. ⁹⁹). None of the female participants reported taking birth control pills. The hormonal indices of adrenarche and gonadarche, residualized and standardized separately within each sex, were combined across the full sample and entered in the reported analyses. Table 1 below contains summary statistics for hormonal indices, as well as youth and parental ratings of pubertal development. At Time 2, partial correlation analyses controlling for sex revealed a robust association between self-reported adrenarche and the adrenal hormonal index (DHEA[boys]/DHEA/testosterone [girls], r of .24, $p < .001$), as well as the gonadal hormonal index (testosterone [boys]/estradiol [girls], r of .34, $p < .001$). At the same

time point, a robust correlation (adjusted for sex) was detected between self-reported gonadarche and the gonadal hormone index (r of .25, $p < .001$) as well as the adrenal hormone index (r of .19, $p = .008$). In contrast, at Time 1, we observed a robust correlation (adjusted for sex) only between self-reported adrenarche and the adrenal hormone index (r of .16, $p = .022$).”

And lines 377-380

HCP-D. Hormonal measures of pubertal development are unavailable in the latest existing data release (2.0) from the HCP-D. Hence, to index reproductive maturation, we relied on self-reports of adrenarche (Female youth: $M = 2.87$, $SD = .70$; Male youth: $M = 2.47$, $SD = .67$; score range: 1-4) and gonadarche (Female youth: $M = 3.09$, $SD = .87$; Male youth: $M = 2.19$, $SD = .72$; score range: 1-4) which were strongly correlated ($r(143) = .70$ [female youth] and $r(130) = .54$ [male youth], both $ps < .001$) (parental reports were not available for the full sample).

2. Second, the details regarding how the authors arrived at their sample sizes is very elusive, particularly for ABCD, which would presumably have thousands of available data points for this particular analysis.

The ABCD data release 5.1, which includes all the available data from the two-year follow-up assessment, contained only 199 ABCD study participants who were biologically unrelated to one another and had good quality data on all the variables included in our analyses. Since our analyses were closely inter-related, inclusion of different participant samples in each test would have made interpretation of the results much more difficult. In our revised manuscript, our participant selection steps for the ABCD and HCP-D samples are illustrated in a STROBE diagram (please see Figure 7). Please note that the HCP-D sample now contains 277 (rather than 308) participants because in our initial submission we failed to eliminate participants sharing the same family ID (and, thus, likely to be biologically related).

3. Finally, the general conceptualization presented in the introduction is not an accurate characterization of the literature. Females may not inherently have increased stress reactivity per se. There may be unique social stressors for females that intersect with neurobiology. Women and girls have historically been underrepresented in studies of acute stress physiology, so it feels like a stretch to make a blanket statement about this in the abstract and to some extent in the introduction. Moreover, the introduction does not provide clear rationale for why puberty could be a catalyst for sexual dimorphism in psychiatric risk. For example, the first two paragraphs are missing the points that a) puberty leads to sex differences in the brain and behavior, and b) the prevailing idea that neuroplasticity observed during adolescence is likely driven by hormonal and psychosocial processes occurring during puberty, which dynamically interact to create sex-

dependent risk for psychopathology.

We agree. In our revised Abstract, we eliminate the reference to increased stress reactivity among females. In our revised Introduction we contextualize findings of sex differences in adolescent psychopathology in reference to social environmental factors. We also unpack our rationale for examining the inter-relationships among neural sexual differentiation, puberty and psychopathology. The revised section on lines 24-33 reads as follows:

“Interactions between social environmental factors and interconnected neurophysiological processes spanning resolutions that range from the molecular (e.g., gene expression^{3, 4}) to the systemic (e.g., the maturation of large-scale brain networks⁵ and the developmental impact of inflammatory processes^{6, 7}) are thought to underpin the rising differential susceptibility of males and females to mood disorders during this life stage^{8, 9}. Because the emergence of sex-dependent risk for psychopathology broadly coincides with the onset of puberty⁹, hormonal mechanisms have been interrogated as potential drivers of the greater cross-species prevalence of internalizing disorders in biological females^{10, 11, 12}. Pubertal hormones have been implicated in the development of sexually dimorphic patterns of behavior and neurophysiology¹³⁻¹⁵, including those relevant to internalizing disorders^{12, 16, 17, 18}. Specifically, with advancing sexual maturation, female adolescents show increasingly stronger neuroimmune responses to stress relative to male youths, which dovetails with evidence that inflammation and metabolic dysregulation, likely indicative of greater physiological wear-and-tear^{19, 20}, make a particularly strong contribution to the pathology of depression and anxiety disorders among adolescent and young adult females^{21, 22}.”

In the “Limitations and Future Directions” sections we also mentioned the potential role of gendered experiences in the reported effects. The revised section on lines 278-289 (added text in red font) reads as follows:

“Relatedly, understanding the mechanisms underpinning physiological wear-and-tear among youth raised in adverse environments, including sexually dimorphic intergenerational transmission of adversity sequelae (e.g., via parental gut microbiota⁸⁶ or maternal placenta in the final stages of pregnancy⁸⁷) and the potential moderating effect of psychosocial factors, such as parenting⁸⁸ are important goals for future investigations. Indeed, Analysis 5 suggested that our identified patterns of neural sexual differentiation covary with financial deprivation across both male and female individuals. Consequently, the sex-dependent associations among brain function, physiology, and psychopathology described in this report may partly reflect sex differences in adversity exposure, including intergenerational transmission of adversity sequelae (cf. ^{89, 90}), a possibility that would certainly warrant further study. Lastly, our three samples rated their sex in a binary manner and only one participant in each of the two adolescent cohorts indicated that their gender as different from the sex assigned at birth (no information on gender—as distinct from biological sex--was collected in the HCP). Therefore, some of the sex differences herein reported may stem from gendered experiences, including

those resulting from gender differences in societal expectations that may extend to potential biases in psychiatric diagnosis. Future studies featuring samples that allow for meaningful dissociations between sex and gender⁹¹ and which probe the interactive impact of multiple social identities⁹² would provide further insights into the effects herein documented.”

Taken together, these aspects of the manuscript made it challenging to fully evaluate each section and the manuscript as a whole, given that some of the variables and critical details are either unsound despite the availability of data, or missing altogether.

We hope that our revisions addressed the reviewer’s concerns.

Reviewer #4 (Remarks to the Author):

This paper aims to test how sex differences in brain functioning, reproductive maturation, and physiological aging may contribute to the vulnerability to developing mood disorders. This is an important question that is hardly addressed in the literature. The authors included two large open-access datasets (ABCD and HCP) to conduct an extensive set of impressive analyses to tackle their research questions. The outcomes showed that greater physiological wear and tear and higher DHEA levels were associated with higher somatic disorders. And sex-specific associations were found for changes in internalizing and externalizing symptoms. They additionally showed that sexual differentiation of brain function is associated differently with physiological age, reproductive maturity, and internalizing and externalizing symptoms in males and females. They replicated their results in the HPC sample.

1. The authors mention that “...are likely to underpin the emerging differential susceptibility of males and females to mood disorders during this life stage.” Although mood disorders in part could be explained by these mechanisms, they also interact with other factors including social cultural gendered experiences and expectations. Although these are not studied in this paper, and do not have to be per se, it should at least be mentioned that they also play a role in the gendered imbalance in diagnosis and treatment. For example, gendered expectations differentially affect self-esteem in males and females, which in turn affects mood disorders; there is an underdiagnosis of externalising disorders in females leading to higher rates of comorbidity in e.g. mood disorders, and there is underdiagnosis of mood disorders in males, just to name a few examples. Gendered experiences are particularly important to take into account when addressing the stress system, as e.g. the chances of experiencing a certain type of trauma are not be the same for males and females. Moreover, twin studies show that mood disorders are, to a larger extent than other disorders, influenced by experiences compared to genetics. Again, these gendered experiences do not have to be taken into account in the present study per

se, but it at least should be discussed.

We agree. Although gender effects are not within the scope of the present manuscript, we have made the following changes in order to acknowledge the role of social environmental and gendered experiences may play in the herein effects reported.

First, in our revised Introduction, we contextualize findings of sex differences in adolescent psychopathology in reference to social environmental factors. We also unpack our rationale for examining the inter-relationships among neural sexual differentiation, puberty and psychopathology. The revised section on lines 24-33 reads as follows:

“Interactions between social environmental factors and interconnected neurophysiological processes spanning resolutions that range from the molecular (e.g., gene expression^{3,4}) to the systemic (e.g., the maturation of large-scale brain networks⁵ and the developmental impact of inflammatory processes^{6,7}) are thought to underpin the rising differential susceptibility of males and females to mood disorders during this life stage^{8,9}. Because the emergence of sex-dependent risk for psychopathology broadly coincides with the onset of puberty⁹, hormonal mechanisms have been interrogated as potential drivers of the greater cross-species prevalence of internalizing disorders in biological females^{10,11,12}. Pubertal hormones have been implicated in the development of sexually dimorphic patterns of behavior and neurophysiology¹³⁻¹⁵, including those relevant to internalizing disorders^{12,16,17,18}. Specifically, with advancing sexual maturation, female adolescents show increasingly stronger neuroimmune responses to stress relative to male youths, which dovetails with evidence that inflammation and metabolic dysregulation, likely indicative of greater physiological wear-and-tear^{19,20}, make a particularly strong contribution to the pathology of depression and anxiety disorders among adolescent and young adult females^{21,22}. “

Second, in the “Limitations and Future Directions” sections, we mention the potential role of gendered experiences in the reported effects. The revised section is on lines 278-289 (added text in red font):

“Relatedly, understanding the mechanisms underpinning physiological wear-and-tear among youth raised in adverse environments, including sexually dimorphic intergenerational transmission of adversity sequelae (e.g., via parental gut microbiota⁸⁶ or maternal placenta in the final stages of pregnancy⁸⁷) and the potential moderating effect of psychosocial factors, such as parenting⁸⁸ are important goals for future investigations. Indeed, Analysis 5 suggested that our identified patterns of neural sexual differentiation covary with financial deprivation across both male and female individuals. Consequently, the sex-dependent associations among brain function, physiology, and psychopathology described in this report may partly reflect sex differences in adversity exposure, including intergenerational transmission of adversity sequelae (cf. ^{89,90}), a possibility that would certainly warrant further study. Lastly, our three samples rated their sex in a binary manner and only one participant in each of the two adolescent cohorts

indicated that their gender as different from the sex assigned at birth (no information on gender—as distinct from biological sex-- was collected in the HCP). Therefore, some of the sex differences herein reported may stem from gendered experiences, including those resulting from gender differences in societal expectations that may extend to potential biases in psychiatric diagnosis. Future studies featuring samples that allow for meaningful dissociations between sex and gender ⁹¹ and which probe the interactive impact of multiple social identities ⁹² would provide further insights into the effects herein documented.”

2. The authors refer to the paper of Zhang et al., 2021 stating that: “...the human brain is best described along a continuum of sexual differentiation, with brains interposed between the male-like and female-like poles of this continuum being the least susceptible to internalizing symptoms.” I think it is important to note that this is also a case of framing, and a nuanced discussion of such outlier analysis could be included: for example the outlier analysis are framed as a male-female continuum but not directly related to male and female traits and mechanisms. An alternative explanation is for example that males show greater variance in brain structure than females (Forde et al., 2020, Ritchie et al., 2018, Wierenga et al., 2018, 2022). And extreme ends of brain patterns (in both males and females) have been associated with greater risk on mental health problems. Next to the conceptualization there are other methodological problems with the prediction modelling in the paper of Zhang see a discussion in for example Sanchis-Segura (2022).

The paper by Zhang et al. (2021) was only mentioned only as an example of research raising the possibility that the human brain may be better described along a continuum of sexual differentiation. However, there is no overlap in our methodologies, hence further discussion of Zhang et al.’s methodology would not be justified in our present report. We thus revised the relevant Introduction sections as follows (changed text in red font)

Lines 34-39

“Extant research on sex-based differences in risk for mood disorders and their link to immune and metabolic dysregulation has traditionally focused on differences between biological males and females. **However, recent evidence suggests that categorizing individuals based on their biological sex ignores considerable within-group variability, since the human brain may be better described along a continuum of sexual differentiation** (for an example, see ²³). Thus, continuous assessments of the degree to which an individual’s brain is sexually differentiated (i.e., **the degree to which it shows brain features expressed more strongly among healthy male, rather than female, adults, or vice versa**) may offer a more sensitive approach to personalizing psychiatric detection and intervention paradigms.”

Lines 59-66 (link to the relevant prior literature, added text in red font)

“Second, we characterized regionally specific patterns of sexual differentiation in brain function at ages 9-10 and 11-12, for each individual adolescent in the ABCD sample, using functional magnetic resonance imaging (fMRI). Subsequently, in the ABCD sample, we tested the relevance of neural sexual differentiation for concurrent and future pubertal development/physiological aging and associated patterns of psychiatric susceptibility (as identified at point 1) (Figure 2: Analysis 2). To contextualize our findings within the broader literature, we also probed the extent to which the regionally specific patterns of neural sexual differentiation thus identified tracked the sensorimotor-association (S-A) axis defined by prior work as a key organizing principle of neurodevelopment (i.e., association areas mature later than sensorimotor areas)^{33, 34}, including associated sex differences in patterns of functional brain network organization (i.e., association network regions are most effective at classifying youths based on sex)³⁵.”

3. There are many factors that influence neurodevelopment and moreover mental health, therefore the following sentence should be toned down: “Our investigation was guided by evidence that adolescent neurodevelopment and mental health are shaped by the interaction between the immune system and the hypothalamic-pituitary-gonadal (HPG) axis”

We revised the sentence to read as follows: “Our investigation was guided by evidence that adolescent neurodevelopment and mental health are influenced by the interaction between the immune system and the hypothalamic-pituitary-gonadal (HPG) axis”.

4. It would help to include the age ranges of the different datasets in addition to the time points in the introduction. Also, it should be justified how the older sample of HCP can be used for estimates in the younger ABCD sample.

In the Introduction (please see lines 57-60, 69), Results (please see lines 80, 92, 118, 119, 124, 132-134, 155) and Figures 2 and 3, we added age ranges for all 3 samples (as applicable).

The use of the young adult HCP sample is justified on

lines 29-30

“Pubertal hormones have been implicated in the development of sexually dimorphic patterns of behavior and neurophysiology¹³⁻¹⁵,... “

and lines 132-136

“Sexual differentiation in the ABCD sample was characterized in reference to the functional connectivity (FC) patterns observed among the young healthy adult participants (ages 22-30) in the Human Connectome Project (HCP). These individuals have reached reproductive maturity and are thus expected to show maximal differentiation of brain function related to their biological sex (as assigned at birth [self-reported] and confirmed through menstruation history^{12, 16, 17, 18, 41, 42}).”

As an aside, please note that the ROI-to-ROI FC estimates based on the Schaefer and Gordon atlases were quantified separately within the ABCD, HCP-D and HCP samples (as specified on lines 136-137). It was the independently extracted ROI-to-ROI FC matrices that were subsequently compared between the HCP and each of the two developmental samples (ABCD, HCP-D) to compute continuous indices of sexual differentiation in adolescent brain function.

5. The authors assume that the brain stems from two sex-typical profiles, which has been shown not to be the case. This is both a matter of algorithm choice as well as labeling which should be addressed and justified (Sanchis-Segura (2019, 2021)).

We would like to underscore the fact that our measure of neural sexual differentiation reflects regionally specific patterns (i.e., greater feminization of association relative to somatomotor regions or vice versa within the same brain) rather than categorization of a whole brain as being more feminine or more masculine. Additionally, we used functional, rather than structural, brain indices. Therefore, we think that the methodological issues outlined by the reviewer, including the assumption that an individual brain on the whole is set to mature into a female or male brain, does not really apply to our present study.

Please also note that our method for estimating regionally specific neural sexual differentiation does account for individual differences in FC within each sex, as explained on lines 488-499 which read as follows (added text is in red font):

“In the HCP sample, we regressed each FC estimate against sex (coded for 1 for male, -1 for female), age in years, income, years of education, employment status, race (coded 0 for White, 1 for non-White), handedness (coded 0 for predominantly right-handed, 1 for not predominantly right-handed), psychopathology (i.e., scores on the DSM-oriented scales of depression, anxiety, ADHD, antisocial personality disorder, somatic disorder, avoidant personality disorder from the Adult Self-Report [ASR]³²), average framewise displacement during the resting state scans as a proxy for individual differences in head motion together with scores on fluid and crystallized/verbal intelligence (i.e., Progressive Matrices and Picture Vocabulary Test, respectively, ¹¹⁴). This analysis resulted in a t-statistic quantifying the degree to which individual differences in each FC estimate are associated biological sex,

such that positive values represent stronger FC in males and negative values represent stronger FC in females. The absolute value of the t-statistic quantifies the magnitude of sexual differentiation for each FC index and, as such, it implicitly accounts for individual differences in sexual differentiation in the young adult group. A low absolute t-value denotes an FC index that shows greater between-individual, rather than between-sex, variability, whereas a high absolute t-value denotes an FC index that is consistently higher (positive t-value) or lower (negative t-value) across most males relative to females. The resulting matrix of t-values thus represents a reference of the degree to which FC estimates are sexually differentiated in healthy young adults.”

6. On a related note, it has been highly debated if the brain is “sexually dimorphic”, which states that the brain is developing in two distinct male and female forms. Using this concept should be discussed and justified, as the effects of differential biological sex effects (the Y-chromosome, hormones, X vs XX chromosomes) are not directly tested in the present paper and therefore no conclusions can be drawn on whether these are associated with the results.

As we mentioned in our reply to this reviewer’s prior point, our study looked at relative patterns of masculinization/feminization within the same brain, not whether a brain, as a whole, is more feminine or masculine. Regarding biological sex, in our revised manuscript, we corroborated parental or self-reports of sex assigned at birth with SNP analysis and/or menstruation history. The relevant sections in the revised manuscript are on

Lines 311-314

“Biological sex was based on parental reports of sex assigned at birth, which were corroborated with menstruation history data (youth and parental reports) from the two-year follow-up and SNP analysis (yielding the SNPSEX variable) using the PLINK command “plink --bfile data --check-sex”. A dummy-coded variable was created to account for the two youths for which sex assigned at birth was not corroborated through menstruation history or SNP analysis.”

Lines 317-318

“Sex was determined from parental reports of sex assigned at birth and corroborated through menstruation history for 108 of the 137 female participants aged 12 and older.”

Lines 325-326

“Sex assignments were based on self-reported sex assigned at birth which was also corroborated through self-reported menstruation history”

However, as we outlined in response to this Reviewer's comment 1, sex and gender were confounded in the ABCD and HCP-D samples, hence some of the sex effects herein reported may be related to gendered experiences. This limitation is outlined in the revised section on lines 278-289

“Relatedly, understanding the mechanisms underpinning physiological wear-and-tear among youth raised in adverse environments, including sexually dimorphic intergenerational transmission of adversity sequelae (e.g., via parental gut microbiota⁸⁶ or maternal placenta in the final stages of pregnancy⁸⁷) and the potential moderating effect of psychosocial factors, such as parenting⁸⁸ are important goals for future investigations. Indeed, Analysis 5 suggested that our identified patterns of neural sexual differentiation covary with financial deprivation across both male and female individuals. Consequently, the sex-dependent associations among brain function, physiology, and psychopathology described in this report may partly reflect sex differences in adversity exposure, including intergenerational transmission of adversity sequelae (cf. ^{89, 90}), a possibility that would certainly warrant further study. Lastly, our three samples rated their sex in a binary manner and only one participant in each of the two adolescent cohorts indicated that their gender as different from the sex assigned at birth (no information on gender—as distinct from biological sex-- was collected in the HCP). Therefore, some of the sex differences herein reported may stem from gendered experiences, including those resulting from gender differences in societal expectations that may extend to potential biases in psychiatric diagnosis. Future studies featuring samples that allow for meaningful dissociations between sex and gender⁹¹ and which probe the interactive impact of multiple social identities⁹² would provide further insights into the effects herein documented.”

7. The analysis included both PDS as well as BMI as predictor variables, is BMI not also influenced by PDS? How would this affect the interpretation of the results?

We briefly discussed the implications of the robust BMI-PDS correlation on lines 190-199 of our revised manuscript:

“In Analysis 4, we therefore used CCA to probe whether individual differences in sexual differentiation of regional FC profiles (as defined using the PLS-extracted latent variable from Analysis 2) (variate 1) relate to scores on the CBCL scales, pubertal development, and markers of physiological aging (BMI, exposure to poverty) in a sex-dependent manner (variate 2). Of note, BMI, which had been robustly linked to PhenoAge in the ABCD sample, was also independently correlated with more advanced adrenarche/gonadarche in the ABCD (r s of .24 and .25, both p s < .001), as well as the HCP-D (r s of .15 [p = .014] and .28 [p < .001]) (correlations adjusted for sex and chronological age, similar to the CCA cross-validation analyses described below). Conversely, financial deprivation, which had been linked to PhenoAge in the ABCD sample was not robustly correlated with

adrenarche/gonadarche in the HCP-D (both $ps > .10$) and the correlation with adrenarche (r of .14, $p = .04$) in the ABCD sample became non-significant after controlling for PhenoAge ($p = .08$). Hence, while BMI is a likely correlate of global aging processes, financial deprivation seems to be more specifically associated with the immune/metabolic dysregulation indexed by PhenoAge.”

8. The authors state that physiological age is highly correlated with environmental factors (e.g. poverty, nutrition). How would that affect the interpretation of the results.

Environmental factors, such as poverty and nutrition, are directly linked to our main results involving neural sexual differentiation in Analyses 4 and 5. Consequently, in our revised Discussion (added text in red font), these effects are interpreted in the context of Analyses 4 and 5 to these analyses on

lines 258-269 (added text in red font)

“The longitudinal ABCD results were replicated cross-sectionally in the HCP-D sample, where sexual differentiation along the S-A axis was observed among faster-developing females, with greater exposure to poverty, at risk for faster physiological aging (higher BMI) and showing more severe mood (particularly, anxiety) symptoms. Considering these findings in light of the Analysis 3 outputs, there emerges the possibility that neural sexual differentiation may constitute a marker of concurrent psychiatric vulnerability (cf. Analyses 4 and 5) which may also identify individuals most likely to show a trajectory of worsening depression and anxiety disorder-related symptoms. This line of inquiry is beyond the scope of the present analysis, but certainly warranting further study.

Underscoring the importance of a continuous approach to neural sexually differentiation²³, we further found that, regardless of biological sex, greater feminization along the S-A axis correlated with prior exposure to poverty, anxiety and, to a lesser extent, depression symptoms, whereas greater masculinization was related to more advanced adrenarche, as well as more severe somatic and behavioral problems. These results fit well with the broader literature on the associations among physiological aging, deprivation, BMI (e.g., via maladaptive stress coping mechanisms, such emotional eating^{73, 74}), and health⁷⁵, including the bidirectional relationship between BMI and internalizing symptoms that emerges in middle childhood⁷⁶, as well as the stronger link between BMI and deprivation observed in girls (relative to boys)⁷⁷. ”

lines 278-289 (added text in red font)

“Relatedly, understanding the mechanisms underpinning physiological wear-and-tear among youth raised in adverse environments, including sexually dimorphic intergenerational transmission of adversity sequelae (e.g., via parental gut microbiota⁸⁶ or maternal placenta in the final stages of pregnancy⁸⁷) and the potential moderating effect of psychosocial factors, such as parenting⁸⁸ are

important goals for future investigations. Indeed, Analysis 5 suggested that our identified patterns of neural sexual differentiation covary with financial deprivation across both male and female individuals. Consequently, the sex-dependent associations among brain function, physiology, and psychopathology described in this report may partly reflect sex differences in adversity exposure, including intergenerational transmission of adversity sequelae (cf. ^{89, 90}), a possibility that would certainly warrant further study. Lastly, our three samples rated their sex in a binary manner and only one participant in each of the two adolescent cohorts indicated that their gender as different from the sex assigned at birth (no information on gender—as distinct from biological sex—was collected in the HCP). Therefore, some of the sex differences herein reported may stem from gendered experiences, including those resulting from gender differences in societal expectations that may extend to potential biases in psychiatric diagnosis. Future studies featuring samples that allow for meaningful dissociations between sex and gender ⁹¹ and which probe the interactive impact of multiple social identities ⁹² would provide further insights into the effects herein documented. ”

9. In general, the results are hard to read and complex to follow. This maybe due to the word limit, but if would help to clarify at some points. For example, LV is not explained at all before the results section. Or the measure of ‘feminization along the S-A axis’ is rather unclear.

We hope that the revisions made in response to all the reviewers’ comments improved the readability of the Results. With regards to specific comments, “LV” is spelled out as ‘latent variable’ on its mention in the Results. It is not mentioned in the Introduction because it pertains to methodology.

The motivation for looking at the S-A axis is unpacked in the Introduction (please see lines 59-66, changed text is in red font):

“Second, we characterized regionally specific patterns of sexual differentiation in brain function at ages 9-10 and 11-12, for each individual adolescent in the ABCD sample, using functional magnetic resonance imaging (fMRI). Subsequently, in the ABCD sample, we tested the relevance of neural sexual differentiation for concurrent and future pubertal development/physiological aging and associated patterns of psychiatric susceptibility (as identified at point 1) (Figure 2: Analysis 2). To contextualize our findings within the broader literature, we also probed the extent to which the regionally specific patterns of neural sexual differentiation thus identified tracked the sensorimotor-association (S-A) axis defined by prior work as a key organizing principle of neurodevelopment (i.e., association areas mature later than sensorimotor areas)^{33, 34}, including associated sex differences in patterns of functional brain network organization (i.e., association network regions are most effective at classifying youths based on sex) ³⁵.”

Furthermore, the Analysis 2 results related to the S-A axis are explained in greater detail on lines 147-152

“Cross-validated ROI loadings on the brain LV tracked the canonical S-A axis (Spearman’s $\rho = -.44$, $p_{\text{spin}} = 2 \times 10^{-5}$), along which typical neurodevelopmental processes, including those associated with sex differences in functional brain architecture, have been shown to unfold^{33, 34, 35}. Thus, greater feminization along the S-A axis (i.e., greater feminization of association, relative to sensorimotor, regions) at T1 was observed among female adolescents showing more advanced PhenoAge and pubertal development (particularly, gonadarche) at T2 (Figure 4-C-F), whereas greater masculinization along the S-A axis (i.e., greater masculinization of association, relative to sensorimotor, regions) typified male adolescents characterized by a younger PhenoAge and delayed pubertal development at T2.”

Additionally, to make the presentation of the results clearer, the inputs and outputs of the CCA and PLS analyses are described in detail (with added text in red font) on

Lines 109-114

(Analysis 1) “To this end, we conducted a CCA in our target ABCD sample ($N = 199$) to identify maximally correlated latent factors (i.e., variates) linking the 18 CBCL scores (i.e., 9 scores from the two-year follow-up [ages 11-12] and 9 change scores estimated between the two- and three-year follow-ups : ages 11-12 to 12-13]; hereafter called the T2/ $\Delta_{T2,T3}$ psychopathology CCA variate) to sex assigned at birth (corroborated through single nucleotide polymorphism (SNP) analysis and/or menstrual history), PhenoAge, pubertal hormone levels, and self-reported adrenarche/gonadarche assessed at T2 (hereafter called the T2 physiology CCA variate).”

Lines 129-137

(Analysis 2) “Having established the sexually dimorphic and primarily longitudinal relationship between physiology and psychopathology in the ABCD sample, we next sought to shed light on its relevance to patterns of neural sexual differentiation. We thus entered the T2/ $\Delta_{T2,T3}$ psychopathology and T2 physiology CCA variates from analysis 1 into a partial least squares (PLS) correlation (Analysis 2) to examine their associations with ROI-specific measures of sexual differentiation in FC, as identified at T1 (ages 9-10) and T2 (ages 11-12). Sexual differentiation in the ABCD sample was characterized in reference to the functional connectivity (FC) patterns observed among the young healthy adult participants (ages 22-30) in the Human Connectome Project (HCP). These individuals have reached reproductive maturity and are thus expected to show maximal differentiation of brain function related to their biological sex (as assigned at birth [self-reported] and confirmed through menstruation history^{12, 16, 17, 18, 41, 42}). ROI-to-ROI FC estimates based on two widely used functional atlases^{43, 44} were extracted independently from the ABCD and HCP samples, respectively.”

Lines 155-172

(Analysis 3) “Our first two analyses provided evidence that physiology at ages 11-12 years shows longitudinal associations with earlier (but not concurrent) neural sexual differentiation (Figure 4-B) and increases in psychopathology, rather than concurrent psychiatric symptoms (Figure 3-A). Building on these findings, we next examined whether sexual differentiation in brain function and physiology sequentially mediate the relationship between genetic risk and psychopathology in the ABCD sample. To this end, we specified the serial mediation model depicted in Figure 2 (Analysis 3), tested with PROCESS 4.2. In line with existing guidelines for probing causal chains ⁴⁶, the serial mediators were sampled from different time points. Based on the results from our first two analyses, we were primarily interested in whether the T1 neural sexual differentiation LV from Analysis 2 (mediator 1) and the sex-dependent physiology CCA variate from Analysis 1 (mediator 2) would constitute a viable causal chain linking polygenic risk scores (PRS) for anxiety and, potentially, depression to rising internalizing symptoms (cf Figure 3-A). In testing this sequence, we simultaneously accounted for alternate models involving serial (T1 and T2 neural sexual differentiation from Analysis 2) and parallel (T2 Physiology from Analysis 1 and T2 neural sexual differentiation from Analysis 2) mediation, respectively (see Figure 2-Analysis 3). The absence of blood chemistry data at T1 prevented us from computing PhenoAge and, thus, estimating the alternate serial and parallel mediation models involving T1 Physiology. However, prior to mediation analysis, all variables were adjusted for T1 pubertal hormone levels and self-reported adrenarche/gonadarche based on the PDS (in addition to participant motion, age, race, testing site, handedness, adoption status, ambiguous biological sex). Additionally, inclusion of the PRS-T2 Physiology-Psychopathology path allowed us to estimate indirect effects of physiology that are independent of T1/T2 neural sexual differentiation and therefore uniquely relate to T1 PhenoAge. PRSs for anxiety, depression and ADHD were computed using summary statistics from large and independent disorder-focused genome-wide association studies (GWASs), featuring case/patient-control comparisons (i.e., MDD ⁴⁷, anxiety disorders ⁴⁸, ADHD ⁴⁹). We did not include PRSs for Oppositional Defiant or Conduct Disorder because we could not locate relevant case-control GWASs.”

Lines 189-193

(Analysis 4) “In our final set of analyses, we tested whether the sex-dependent links between brain function, psychiatric risk, and physiological maturation, as established in the ABCD sample (Analyses 1-3) would extend cross-sectionally in the larger HCP-D sample. In Analysis 4, we therefore used CCA to probe whether individual differences in sexual differentiation of regional FC profiles (as defined using the PLS-extracted latent variable from Analysis 2) (variate 1) relate to scores on the CBCL scales, pubertal development, and markers of physiological aging (BMI, exposure to poverty) in a sex-dependent manner (variate 2).”

lines 217-220

(Analysis 5) “ Our final analysis linked the cross-validated sexual differentiation brain LV from the ABCD (Analysis 2, Figure 4-D [but unthresholded]) to the partial correlation maps quantifying the relationship between neural sexual differentiation and the DSM-oriented CBCL scores, pubertal development (i.e., adrenarche, gonadarche), and markers of physiological aging as identified in the ABCD (BMI, financial deprivation) (adjusted for sex, age, handedness, testing site, participant motion and adoption status).”

10. The results of analysis 1 are confusing, as it seems like the directionality of the association between levels of physiological wear-and-tear and DHEA and somatic scores are the same in males and females. To clarify, the different findings on cross-sectional and change data could be described separately.

Somatic, anxiety disorder, OCD, PTSD and depression scores have the same sign loading on the Analysis 1 Psychopathology variate (Figure 3-A) and, thus, are all related in the same direction with the Physiology variate (Figure 3-B). We hope that the revised description of the Analysis 1 results on lines 120-126 clarifies this matter:

“Discovery CCAs unveiled a statistically significant physiology-psychopathology variate pair which was successfully cross-validated in analyses adjusted for age, race, testing site, handedness, adoption status (coded “1” for the 3 adoptees and “0” for the remaining sample), ambiguous biological sex (coded “1” for the 2 participants with ambiguous sex and “0” for the remaining sample) ($r_{CV} = .18$, permutation-based $p = .019$, Figure 3-C). Thus, female adolescents characterized by more advanced PhenoAge and pubertal development, particularly, gonadarche, showed persistently higher somatic disorder scores and increasing depression, anxiety, OCD and PTSD scores from ages 11-12 to 12-13 (Figure 3-A, B). Conversely, consistently lower somatic disorder scores but rising conduct and oppositional defiant disorder scores were observed among male adolescents characterized by a younger PhenoAge and delayed pubertal development (Figure 3-A, B).”

REVIEWER COMMENTS

Reviewer #1 (Remarks to the Author):

Authors have addressed most of my concerns. Please find below a few remaining/additional comments.

1. The authors state that they examined the "relative patterns of masculinization/feminization within the same brain, not whether a brain, as a whole, is more feminine or masculine". This, along with the idea of sexual dimorphism (discussed below), is a highly debated topic and has been extensively studied in different neurobiological measures (e.g., brain structure, function, connectivity). The discussion could be enhanced by considering how these findings relate to existing literature and perspectives on this topic (e.g., Ingahlakar et al., PNAS 2013, along with the commentary and letter in response to the paper [Cahill et al., PNAS 2014; Joel et al., PNAS 2014], and the response to the letter; Joel et al., PNAS 2015, along with the letters in response to the paper [Guidice et al., PNAS 2016; Glezerman et al., PNAS 2016; Rosenblatt et al., 2016; Chekroud et al., 2016], and the replies to those letters; Satterthwaite et al., Cerebral Cortex 2014; Sanchis-Segura et al., BOSD 2019; Sanchis-Segura et al., NeuroImage 2022; among many others). Overall, the manuscript could benefit from a broader engagement with the existing literature.

We thank the reviewer for drawing our attention to these papers. In the Discussion (lines 286-293), we added a paragraph intended to situate our work within the relevant literature (word limit-related concerns did not allow us to elaborate on this point any further):

“It is worth noting that substantial efforts have been channelled towards characterizing the prototypical “female” vs “male” brain and developing sex-based classification algorithms using structural or functional neural features^{80, 81, 82, 83}. A complementary strand of research views human brains as comprising a “mosaic” of characteristics, some of which are more common in females, some in males, and some equally between the sexes⁸⁴. Our work is more closely aligned with the latter approach, but our analysis diverges from both lines of research by focusing on how the relative feminization/masculinization of functional connectivity patterns observed in the same brain was independently and additively related to sex and aging-relevant variables (i.e., PhenoAge, pubertal development). Put differently, we sought to elucidate whether sex and aging-related processes would be independently related to similar patterns of within-brain sexual differentiation, which, in turn, could explain differential susceptibility to internalizing vs externalizing disorders.”

2. Authors repeatedly use the term sexually dimorphic throughout the manuscript to refer to sex differences. As an

example, the first sentence introduces the idea of 'sexually dimorphic mental health trajectories'. However, mental health trajectories are not sexually dimorphic. 'Sexual dimorphism' implies that males exhibit distinct traits compared to females. That is not the case for mental health traits. Males may be more likely to exhibit certain traits than females, and vice versa, but both males and females can exhibit the same mental health traits. Please revise the terminology used (see McCarthy et al., 2012 J Neurosci for a detailed discussion on appropriate usage). Please also consider whether the term sexually dimorphic is appropriate in other instances throughout the manuscript.

We thank the reviewer for drawing our attention to McCarthy's et al.'s (2012) paper. In light of the terminology proposed by McCarthy et al., the processes described in our report fall under the umbrella of "sex differences". Hence, "sexual dimorphism" and its derivatives have been replaced by "sex differences" (e.g., lines 25, 66, 161, 230, 248, 259, 266, 303, 307, 310), "sex differential" (e.g., lines 31, 80, 141, 253, 264, 320) or "vary by sex" (line 48), with a direct link to the terminology proposed by McCarthy (2012) on line 310.

3. I recommend avoiding the term 'biological females' in the first sentence of the abstract. In general, internalising disorders are more prevalent in females and in women (regardless of their assigned sex), but it has not yet been established whether this is a sex effect or a gender effect. Therefore, the term 'females' or 'women' (or both) is more appropriate here.

We agree and have replaced "biological females" with "females" in the first sentence of the abstract.

"Internalizing disorders are more prevalent among females."

1. Line 159: please describe what "PROCESS 4.2" refers to.

We added the requested clarification to lines 171-173 which read as follows (added text in red font, remaining text included for context):

"To this end, we specified the serial mediation model depicted in Figure 2 (Analysis 3) and tested with PROCESS 4.2, **which is an ordinary least squares (OLS) and logistic regression path analysis modelling tool based on observable variables** ⁴⁷ (SI 7.3)."

1. Line 187: section title suggests authors are referring to global "feminization" of the brain - consider using "regional FC feminization" instead.

The section title for Analysis 4 (lines 202-203) has been modified in line with the reviewer's suggestion

“Analysis 4: The Link between **Regional** FC Feminization, Advanced Phenotype, Pubertal Maturation, and Anxiety Disorders Replicates in An Independent Cross-Sectional Sample”

For the same reason, we also revised the section title for Analysis 5 (lines 226-227) which reads as follows

“Analysis 5: **FC Feminization along the S-A Axis** Distinguishes Exposure to Poverty and Anxiety from Behavioural, Somatic and Attentional Disorders, Regardless of Biological Sex”

Reviewer #2 (Remarks to the Author):

I appreciate the authors' thoughtful and thorough consideration of my feedback. I find most of their changes to be satisfactory. However, on one particular front I think there is additional work needed.

1. I previously suggested “Despite substantial evidence that head motion causes widespread functional connectivity alterations (Power et al., 2012; Satterthwaite et al., 2012, 2019; Van Dijk et al., 2012; Ciric et al., 2017), in particular in youth, the paper is lacking a discussion of the removal of head motion artifacts. While the HCP-D data was processed with ICA-FIX, the ABCD Study processing contains no mention of motion scrubbing (or other method of removal). Please assess relationships between head motion and functional connectivity estimates in both samples to report if head motion artifacts were sufficiently removed from the brain data. Additionally, if motion scrubbing was implemented in the ABCD Study, please describe the procedure used and list the amount of data remaining (mean, standard deviation, range) after motion scrubbing.” The authors replied with a description of the motion correction techniques that were implemented (e.g., nuisance regression with estimates of head motion and respiratory notch filtering, inclusion of average FD as a covariate in models). The authors also stated that “These motion correction steps are in line with those recommended by the ABCD study team (cf. Hagler et al., 2019).” However, the Hagler et al. (2019) paper does discuss motion censoring (AKA scrubbing), or the removal of high motion frames from the timeseries prior to the construction of correlation matrices. Here is the corresponding excerpt from Hagler et al. (2019):

“Motion censoring: Motion censoring is used to reduce residual effects of head motion that may survive the regression

(Power et al., 2012, 2014). Time points with FD greater than 0.2 mm are excluded from the variance and correlation calculations. Note that this is a slightly more conservative threshold than that used for the regression step. Time periods with fewer than five contiguous, sub-threshold time points are also excluded. The effects of head motion can potentially linger for several seconds after an abrupt head motion, for example due to spin-history or T1 relaxation effects (Friston et al., 1996), so an additional round of censoring is applied based on detecting time points that are outliers with respect to spatial variation across the brain. SD across ROIs is calculated for each time point, and outlier time points, defined as having an SD value more than three times the median absolute deviation (MAD) below or above the median SD value, are excluded from variance and correlation calculations.”

The authors state that the ABCD resting-state data is of “good quality”, but this should be demonstrated as (1) it is unusual to not include motion censoring in the data analysis of ABCD resting-state data (e.g., Marek, Tervo-Clemmens, et al. 2022), (2) it deviates from the processing steps outlined in Hagler et al. (2019) and, thus, also the released resting state data on the NDA, and (3) the prior literature robustly demonstrates that the steps the authors took here are not sufficient to remove the impacts of head motion on the data and any resulting inferences (Power et al., 2012; Ciric et al., 2017).

If the authors would prefer to not implement motion censoring, they can demonstrate that the impacts of head motion have been sufficiently removed using the benchmarks outlined by Ciric et al. (2017) “Benchmarking of participant-level confound regression strategies for the control of motion artifact in studies of functional connectivity”. Specifically, demonstration of the (a) residual QC-FC correlations after de-noising and (b) distance-dependence of motion artifacts after de-noising. Comparing these results to those published by Ciric et al. (2017) will help to clarify if there are remaining motion impacts in the data. The scripts to create these benchmarks are made publicly available in <http://github.com/PennBBL/xcpEngine> and are further described in Ciric et al., (2018) “Mitigating head motion artefact in functional connectivity MRI.”

In response to the Reviewer’s comment, we have added Figure S10 which depicts the distribution of the QC-FC correlations and the distance-dependent motion artifacts across the Schaefer and Gordon atlas, both in the full ABCD sample analysed here (N=199), as well as in a subsample of participants passing the stringent motion elimination criteria outlined in Parkes et al. (2018, NeuroImage). Specifically, these criteria were: average participant FD < .20 mm, 80% of all collected volumes having an FD < .20 mm, and maximum motion under 5 mm. Parkes et al. found that censoring had no additional benefit beyond the exclusion of such high motion participants. We therefore favoured this approach to avoid analyzing data with different numbers of timepoints in each participant, which can arise if some form of censoring/scrubbing is applied. The distribution of the QC-FC correlations and

associated medians, as well as the pattern of distance-dependent artifacts matches the metrics of the best performing denoising pipelines depicted in Ciric et al. (2017, NeuroImage, Figures 3, 4), supporting the efficacy of our approach.

Ciric et. Al 2017.Neuroimage fig.3 QC-FC Correlations

Figure S10: Schaefer atlas (main sample [A, B] and low motion sample [C, D]) and Gordon atlas (main sample [I, J] and low motion sample [K, L])

[FIGURE REDACTED]

Ciric et al. (2017, NeuroImage, Figure 4-C): Distance- dependence of the motion artifact

Figure S10: Schaefer atlas (main sample [E, F] and low motion sample [G,H]) and Gordon atlas (main sample [M, N] and low motion sample [O,P])

[FIGURE REDACTED]

To control for any differences between higher and lower FD participants which may have remained even after the application of the participant-level denoising pipeline (evaluated in Figure S10), average FD per participant was partialled out during the cross-validation of the PLS results (Analysis 2), serial mediation analyses (Analysis 3), and the cross-validation of the CCA results in the HCP-D sample (Analyses 4 and 5), as well as in the creation of the HCP-D psychopathology/aging brain maps which were entered in Analysis 5.

To confirm that the discovery PLS analyses (Analysis 2) were not biased by participant-level motion, we reran the discovery PLS analyses after residualizing the behavioral set for the same confounds entered in the cross-validation analyses (since motion may interact with these other confounds to bias connectivity). We only residualized the behavioural set because recent evidence suggests that residualization of both variable sets for multivariate analyses, such as PLS, which use permutation-based significance testing, may bias results (Winkler et al., 2020, NeuroImage). The results of these additional analyses, which replicate those reported in the main text, are presented in Figures S3 and S4 for the full sample of 199 youths and for the 89 low-motion participants. The unthresholded brain LV from the discovery PLS analysis described in the main text diverged only slightly from the one extracted from the residualized data (r s of .93 and .92 with the Schaefer and Gordon atlas, respectively) (Figure S4).

Our response above has been included in the main text of the manuscript on lines 491-496

“To verify the effectiveness of the denoising pipeline, we estimated QC-FC correlations (i.e., correlations between participant-level framewise displacement and each ROI-to-ROI functional connectivity index, see “ROI definition and correlations” below). The distribution of the QC-FC correlations and the scatterplots describing the distance-dependence of the motion artifacts are included in Figure S10. The observed metrics paralleled those of the best performing pipelines described in ¹¹⁹ (see also ¹²⁰). To remove any potential lingering artifacts, all group-level analyses controlled for average framewise displacement per participant (see section titled “Control variables”).”

and lines 640-651

“Thus, via linear regression, we removed average framewise displacement from all neural outcomes of interest (i.e., ROI-to-ROI indices of sexual differentiation and the sexual differentiation latent variable [Analyses 2-5]) to address any residual motion-related confounds not addressed by our pre-processing pipeline¹¹⁹. As an additional control, we re-ran the discovery PLS analyses after residualizing the behavioral set for the same confounds entered in the cross-validation analyses (since motion may interact with these other confounds to bias the ROI-to-ROI indices of sexual differentiation in FC). We only residualized the behavioural set because residualization of both variable sets for multivariate analyses, such as PLS, which use permutation-based significance testing, has been found to bias results ¹⁴⁴. These additional analyses replicated those reported in the main text. The unthresholded brain LV from the discovery PLS analysis described in the main text diverged only slightly from the one extracted from the residualized data (r s of .93 and .92 with the Schaefer and Gordon atlas, respectively, Figure S4-A, D). The brain LV identified in these additional analyses (Figures S3, S4) was robustly correlated, based on 95% CIs, with the physiology variate at Time 1 (r s from to .27 to .49 across both atlases) and the psychopathology variate at both time points (r s from to .18 to .62 across both atlases). The relevant brain maps are presented in Figures S3 and S4 for the full sample of 199 youths and for the 89 low-motion participants.”

As a side note, the reviewer states that the ABCD data available on NDA includes motion censoring. We assume that the reviewer refers to the tabulated imaging data, since the preprocessed resting state nifti files released on NDA (and recommended for analysis) did not include motion censoring (or any other motion correction beyond realignment). One reason for doing so was likely to allow more flexibility for investigations, such as ours, that span multiple samples and benefit from maximal comparability with regards to scan length.

Reviewer #3 (Remarks to the Author):

I appreciate the authors responding to all of my initial comments. I think that aspects of the manuscript are improved, including the introduction and the addition of methodological/analytic details. I have several remaining concerns and comments for the authors to address.

1. I caution the authors against using the term “stages” when referring to adrenarche and gonadarche. They are comprised of stages (up to 5, if you are using the Tanner convention). It could be confusing to readers to refer to them as stages. It also makes it sound like they are contingent upon one another, which they are not. The authors could consider referring to them as processes or components.

In the revised manuscript, we refer to adrenarche and gonadarche as “component processes of puberty” (please see lines 84-85, 358, and 359).

1. What is meant by “standardized” in reference to the hormonal assays?

The hormonal index for adrenarche and gonadarche contained different hormones for girls vs boys. To account for this heterogeneity, the contributing hormone levels were standardized (i.e., z-scored) separately within boys and girls, respectively.

For the adrenal hormone index, DHEA levels were standardized within boys. For girls, we first averaged the standardized DHEA and testosterone levels, then we standardized the resulting average scores (please see our reply to point 5 below for the rationale behind combining DHEA and testosterone in the adrenal hormone index for girls).

For the gonadal hormone index, testosterone levels were z-scored within boys, whereas estradiol levels were z-scored within girls.

The above clarifications are included on lines 368-373 of our revised manuscript (added text in red font, remaining text included for context):

“Hormonal markers of adrenarche and gonadarche were estimated separately for boys and girls. In boys, z-scored DHEA levels were used as a marker of adrenarche, whereas in girls, adrenarche was estimated as the average of the standardized (i.e., z-scored) DHEA and testosterone levels, because existing evidence links testosterone more strongly to adrenarche, rather than

gonadarche, among female youth³⁷. Gonadal hormone scores were computed based on hormones that had been directly implicated in sex-specific reproductive maturation processes. Thus, z-scored estradiol levels indexed female gonadarche, whereas z-scored testosterone levels indexed male gonadarche (for a review of supporting findings, see^{37, 38}).

lines 383-385

“The hormonal indices of adrenarche and gonadarche, residualized and standardized (i.e., z-scored) separately within each sex, were combined across the full sample and entered in the reported analyses.”

1. Please put the units for the hormone assays in Table 1. The ranges shown are very wide on all of the hormones – did the authors check for outliers?

We added the measurement units for the hormonal assays in Table 1. We used non-parametric tests of statistical significance (i.e., permutation- and bootstrap-based testing) for which the presence of outliers is not problematic. The only instance in which we had not used non-parametric tests of significance was the correlation analyses between hormonal and self-reported indices of pubertal development. In the revised manuscript, we have re-run these analyses using Spearman’s correlations and permutation-based testing of significance. The revised section on lines 386-393 reads as follows:

“At Time 2, partial correlation analyses (with 100,000 permutation samples) controlling for sex revealed a robust association between self-reported adrenarche and the adrenal hormonal index (DHEA[boys]/DHEA/testosterone [girls], Spearman’s rho of r of .27, permutation-based $p = 1.5 \times 10^{-4}$), as well as the gonadal hormonal index (testosterone [boys]/estradiol [girls], Spearman’s rho of r of .35, permutation-based $p = 10^{-5}$). At the same time point, a robust correlation (adjusted for sex) was detected between self-reported gonadarche and the gonadal hormone index (Spearman’s rho of r of .29, permutation-based $p = 10^{-4}$) as well as the adrenal hormone index (Spearman’s rho of r of .24, permutation-based $p = .001$). In contrast, at Time 1, while self-reported adrenarche was robustly correlated with both the adrenal hormone index (Spearman’s rho of .17, permutation-based $p = .015$) and the gonadal hormone index (Spearman’s rho of .19, permutation-based $p = .009$), self-reported gonadarche showed a statistically significant correlation only with the adrenal hormone index (Spearman’s rho of .19, permutation-based $p = .006$).

1. It’s unclear why the authors chose to partial out age at menarche for the girls. Typically variables accounting for more momentary or monthly fluctuations, like caffeine and days since last menstrual cycle that the authors use, are covaried out of salivary/serum hormone levels. It’s true that girls who are post menarche are expected to have higher levels of the hormones of interest here, but that seems like meaningful variance that the authors may not want to remove.

We controlled for age at menarche because it is related to cycle (ir)regularity, including associated patterns of hormonal variations. We have clarified this on lines 379-383 which read as follows:

“For female youths, the hormonal indices were further residualized by self-reported age at menarche since, in samples as young as ours who had recently experienced menarche, age at menarche relates to regularity of the menstrual cycle and associated hormonal fluctuations¹⁰⁶, days since the start of the last menstrual cycle, and whether the menstrual cycles are regular (based on self- and parental reports as a subjective complement to the cycle regularity information gauged via self-reported age at menarche) (cf.¹⁰⁵).”

5. I am unsure why the authors are including testosterone in the adrenal hormones for girls. It's true that DHEA/s are precursors to testosterone (and estrogen), but testosterone is not typically examined as part of adrenarche. I worry that by including testosterone, the authors may be undermining the specificity of any potential adrenarcheal effects since testosterone is released only in small quantities from the adrenal glands, with much higher levels released as part of gonadarche.

The female adrenal hormone index included both DHEA and testosterone in light of evidence from Byrne et al. (2023) that, in girls, testosterone levels load more strongly on hormonal and PDS indices of adrenarche than gonadarche.

Conversely, testosterone was not included in the gonadal hormone index for girls because our gonadal hormone index included only hormones that had been shown to be directly related to reproductive maturation and the emergence of sex-specific characteristics. To our knowledge, in girls, this is estradiol, whereas in boys it is testosterone.

These clarifications have been added to lines 368-373 of our revised manuscript (added text in red font, remaining text included for context):

“Hormonal markers of adrenarche and gonadarche were estimated separately for boys and girls. In boys, z-scored DHEA levels were used as a marker of adrenarche, whereas in girls, adrenarche was estimated as the average of the standardized (i.e., z-scored) DHEA and testosterone levels, because existing evidence links testosterone more strongly to adrenarche, rather than gonadarche, among female youth³⁷. Gonadal hormone scores were computed based on hormones that had been directly implicated in sex-specific reproductive maturation processes. Thus, z-scored estradiol levels indexed female gonadarche, whereas z-scored testosterone levels indexed male gonadarche (for a review of supporting findings, see^{37, 38}).”

6. In the results section for analysis 1, please describe how the change scores were computed.

We have added this information on lines 111-114 of our revised manuscript (revised text in red font, remaining text included for context):

“To this end, we conducted a CCA in our target ABCD sample (N = 199) to identify maximally correlated latent factors (i.e., variates) linking the 18 CBCL scores (i.e., 9 scores from the two-year follow-up [ages 11-12] and 9 change scores estimated as the difference between the corresponding standardized scores at the three- and two-year follow-ups : ages 12-13 vs 11-12]...)...”

1. The findings from analysis 2 showing that neural “masculinization” is related to lower phenoAge and delayed puberty and vice versa in girls seems like it could be attributable to the fact that girls, in general, go through puberty earlier than boys. Could it just be that this result is demonstrating a normative sex-independent developmental process and not necessarily sex differences per se, just the ~1 year lag in pubertal timing between boys and girls (which is already well-documented)?

The reviewer raises the possibility that the physiology variate identified in Analysis 1 reflects shared variance between sex and maturation; specifically, that at similar chronological ages, adolescent girls should show more advanced PhenoAge and pubertal maturation relative to boys. To address this possibility, we conducted a regression analysis predicting the z-scored physiology variate from sex, z-scored PhenoAge, z-scored pubertal development variables and all the control variables entered in the cross-validation of the Analysis 1 results (of the control variables, only age at T2 and T3 were z-scored since the others were dummy coded). This analysis is akin to estimating the standardized canonical coefficients which reflect the unique contribution of each observed variable to its corresponding variate. The results indicated that the physiology variate reflects the unique, additive contributions of sex ($b = .88$, 95% CI [.79; .99]), PhenoAge ($b = .55$, 95% CI [.50; .61], and gonadarche, based on sex hormone levels ($b = .14$, 95% CI [.08; .21]) and self-reports ($b = .33$, 95% CI [.27; .39]) (hormonally indexed and self-reported adrenarche did not show robust unique associations with the physiology variate in the multiple linear regression analysis). The results remain unchanged if the z-scored physiology variate is regressed only on its corresponding observed variables (i.e., sex, PhenoAge, self-reported adrenarche, self-reported gonadarche, adrenal hormone index, gonadal hormone index).

This additional analysis is described on lines 129-138 of our revised manuscript:

“To shed further light on the results described above, we evaluated whether sex, PhenoAge and the pubertal development variables, particularly gonadarche, make independent contributions to the physiology variate. To this end, we conducted a regression analysis predicting the z-scored physiology variate from sex, z-scored PhenoAge, z-scored pubertal development variables, and all the control variables entered in the cross-validation of the Analysis 1 results. This analysis is akin to estimating the standardized canonical coefficients, which reflect the unique contribution of each observed variable to its corresponding variate⁴¹. The results confirmed that the physiology variate reflects the unique, additive contributions of sex ($b = .88$, 95% CI [.79; .99]), PhenoAge ($b = .55$, 95% CI [.50; .61]), and gonadarche, based on hormone levels ($b = .14$, 95% CI [.08; .21]) and self-reports ($b = .33$, 95% CI [.27; .39]). Hormonally indexed and self-reported adrenarche did not show robust unique associations with the physiology variate in the multiple linear regression analysis. The results remain unchanged if the z-scored physiology variate is regressed only on its corresponding observed variables. These findings indicate that the physiology variate does not merely reflect earlier reproductive maturation and more advanced PhenoAge in girls relative to boys.”

REVIEWER COMMENTS

Reviewer #2 (Remarks to the Author):

I appreciate that the authors included the Ciric benchmarks evaluating the impact of head motion on functional connectivity measures. I agree with the authors that the QC-FC plots demonstrate metrics similar to the best performing models that Ciric et al. tested. However, the correlation values are left off the distance-dependent graphs. It appears that there is a stronger distance-dependent effect in the study sample compared to the low motion sample and it is unclear how the values compare to those reported by Ciric et al. Please add the correlation values to the distance-dependent graphs in Figure S10.

We apologise for this omission: the correlation values have been added to the distance-dependent graphs in Figure S10. The distance-dependent correlation in the low motion sample (Spearman's rho [absolute value] between .18 and .20) (Figure S10-G, H, O,P) matched the metrics of the best performing pipelines in Ciric et al. (2017, Figure 4).

Distance- Figure S10: Schaefer atlas (main sample [E, F] and low motion sample [G,H]) and Gordon atlas (main sample [M, N] and low motion sample [O?])

[FIGURE REDACTED]

Ciric et al. (2017, NeuroImage, Figure 4-C):dependence of the motion artifact

There is indeed a stronger distance-dependent effect in the study sample (N= 199), although the QC-FC metrics from the study sample match the metrics of the best performing pipelines from Ciric et al. (2017). As we mentioned in our reply during the prior review round, we have taken several measures to control for any differences between higher and lower FD participants which may have remained even after the application of the participant-level denoising pipeline (evaluated in Figure S10).

In short, we took the following steps:

1. We confirmed that the discovery PLS analyses (Analysis 2) were not biased by participant-level motion by re-running the discovery PLS analyses after residualizing the behavioral set for average FD per participant and other confounds which may interact with motion to bias connectivity (as listed under “Control Variables”). The unthresholded brain LV from the discovery PLS analysis described in the main text diverged only slightly from the one extracted from the residualized data (r s of .93 and .92 with the Schaefer and Gordon atlas, respectively) (Figure S4).
2. We ran the same discovery PLS analyses on the low motion sample who had shown QC-FC metrics and distance-dependent effects matching the metrics of the best performing pipelines from Ciric et al. (2017). The unthresholded brain LV extracted from the low motion sample (Figure S4-B, C, E, F), either without or with further residualisation of the behavioural set by average participant FD and the other confounds listed under “Control Variables”, replicated the one extracted from the study sample (Figure 4-C, E [Schaefer atlas]; Figure S2-C, E [Gordon atlas]).
3. (remaining analyses) Average FD per participant (together with the other confounds listed under “Control Variables”) was partialled out during the cross-validation of the PLS results (Analysis 2), serial mediation analyses (Analysis 3), and the cross-validation of the CCA results in the HCP-D sample (Analyses 4 and 5), as well as in the creation of the HCP-D psychopathology/aging brain maps which were entered in Analysis 5.

Given the control analyses we implemented (as described above), we think it is unlikely that our reported results are substantially contaminated by motion artifacts.

REVIEWER COMMENTS

Reviewer #2 (Remarks to the Author):

I thank the authors for adding the correlation values into the distance-dependent graphs. I additionally agree with the authors that given the control analyses implemented their results are likely not substantially impacted by motion artifacts. I do want to contest the authors statement in the manuscript that the distance-dependent correlation matched the metrics of the best performing pipelines in Ciric et al. (2017): "The distribution of the QC-FC correlations and the scatterplots describing the distance-dependence of the motion artifacts are included in Figure S10. The observed metrics paralleled those of the best performing pipelines described in 119 (see also 120 495)." I caution the authors to temper their language in this sentence, as it is currently somewhat misleading. Looking at the distance-dependence numbers for the full sample now included in Supplement Figure 10 (from -.35 to -.48), this suggests the data used in the manuscript is not among the best performing pipelines tested by Ciric (from .003 and -.116) but is rather among the worst performing pipelines tested by Ciric. I would suggest the authors rephrase this sentence to better align with the data reported in Supplement Figure 10.

We agree with the reviewer and revised the sections in which we describe the motion control analyses to align them with the results depicted in Figure S10. Specifically, we clarify that only the distribution of the QC-FC correlations matches the metrics of the best performing pipelines from Ciric et al. (2017) for both the sample of 199 participants used in the main analyses and the low motion sample. However, for the distance-dependent motion metrics, it is only in the low motion sample that we see metrics similar to the top 3 or top 6 best performing pipelines out of the 14 pipelines tested in Ciric et al. (2017).

The revised sections are on lines 776-778

"The distribution of the QC-FC correlations and the scatterplots describing the distance-dependence of the motion artifacts are included in Figure S10. The observed metrics for the QC-FC correlations paralleled those of the best performing pipelines described in ¹¹⁹ (see also ¹²⁰)."

and lines 1015-1034

"As an additional control, we re-ran the PLS analysis in a subsample of 89 participants passing the stringent motion elimination criteria outlined in reference¹²⁰. Specifically, these criteria were: average participant FD < .20 mm, 80% of all collected volumes having an FD < .20 mm, and maximum motion under 5 mm. Reference¹²⁰ found that more aggressive motion controls (e.g., censoring) had no

additional benefit beyond the exclusion of such high motion participants. We therefore favoured this approach to avoid analyzing data with different numbers of timepoints in each participant, which can arise if some form of censoring/scrubbing is applied. Similar to the full sample of 199 youths, these 89 low-motion youths showed a distribution of QC-FC correlations and associated medians that matched the metrics of the best performing denoising pipelines depicted in reference¹¹⁹ (see Figure S10). However, the low motion sample showed QC-FC/distance correlations lower than those observed in the full sample of 199 youths. Of note, the QC-FC/distance correlations in the low motion sample matched the top 3 or top 6 (depending on the atlas¹¹⁹) best performing pipelines out of the 14 pipelines tested in reference¹¹⁹. The PLS brain LV identified in the full sample of 199 youths was replicated in the low-motion sample (see Figures S3, S4). In sum, the brain LV identified with the discovery PLS analysis (Figure 4-C) was replicated in the low-motion sample and after residualizing for confounds, including motion (Figures S3, S4). All the other reported analyses that featured the sexual differentiation indices controlled for average participant FD. Given all these controls, we think it is unlikely that our reported results are substantially contaminated by motion artifacts.”